# A taxonomy of seizure dynamotypes

Maria Luisa Saggio[1†], Dakota Crisp[2†], Jared M Scott[2], Philippa Karoly[3], Levin Kuhlmann[4,5], Mitsuyoshi Nakatani[1], Tomohiko Murai[6], Matthias Dümpelmann[7,8], Andreas Schulze-Bonhage[7,8,9], Akio Ikeda[6], Mark Cook[3,4], Stephen V Gliske[10], Jack Lin[10], Christophe Bernard[1]*, Viktor Jirsa[1]*, William C Stacey[2,10]*

[1]Aix Marseille Univ, Inserm, INS, Institut de Neurosciences des Systèmes, Marseille, France, Marseille, France; [2]Department of Biomedical Engineering, BioInterfaces Institute, University of Michigan, Ann Arbor, United States; [3]Graeme Clark Institute, The University of Melbourne, Melbourne, Australia; [4]Department of Medicine, St. Vincent's Hospital, The University of Melbourne, Melbourne, Australia; [5]Faculty of Information Technology, Monash University, Clayton, Australia; [6]Department of Epilepsy, Movement Disorders and Physiology, Kyoto University Graduate School of Medicine, Kyoto, Japan; [7]Epilepsy Center, Medical Center – University of Freiburg, Freiburg im Breisgau, Germany; [8]Faculty of Medicine, University of Freiburg, Freiburg im Breisgau, Germany; [9]Center for Basics in NeuroModulation (NeuroModul Basics), Epilepsy Center, Medical Center - University of Freiburg, Freiburg im Breisgau, Germany; [10]Department of Neurology, University of Michigan, Ann Arbor, United States

**\*For correspondence:**
christophe.bernard@univ-amu.fr (CB);
viktor.jirsa@univ-amu.fr (VJ);
william.stacey@umich.edu (WCS)

[†]These authors contributed equally to this work

**Abstract** Seizures are a disruption of normal brain activity present across a vast range of species and conditions. We introduce an organizing principle that leads to the first objective Taxonomy of Seizure Dynamics (TSD) based on bifurcation theory. The 'dynamotype' of a seizure is the dynamic composition that defines its observable characteristics, including how it starts, evolves and ends. Analyzing over 2000 focal-onset seizures from multiple centers, we find evidence of all 16 dynamotypes predicted in TSD. We demonstrate that patients' dynamotypes evolve during their lifetime and display complex but systematic variations including hierarchy (certain types are more common), non-bijectivity (a patient may display multiple types) and pairing preference (multiple types may occur during one seizure). TSD provides a way to stratify patients in complement to present clinical classifications, a language to describe the most critical features of seizure dynamics, and a framework to guide future research focused on dynamical properties.

## Introduction

Epilepsy is one of the most common neurological disorders with an estimated prevalence of 50 million worldwide (*World Health Organization, 2020*). It is characterized by spontaneously recurring seizures, which are 'a transient occurrence of signs and/or symptoms due to abnormal excessive or synchronous neuronal activity in the brain' (*Fisher et al., 2005*). However, there are a vast array of signs, symptoms, and underlying causes of seizures. Thus, despite high prevalence and considerable morbidity and mortality, it has been challenging to characterize, treat, and understand seizures, which prevents the development of reasoned, mechanistic approaches to therapy and improved patient care. Seizure classifications to date have been purely descriptive of empirical data: clinical manifestations (e.g. focal vs. generalized) that are based upon the region of brain affected rather than the seizure itself, and visual descriptions of electroencephalogram (EEG) waveforms. These classifications have been subjected to numerous revisions. In its latest position paper, the International

**eLife digest** Epileptic seizures have been recognized for centuries. But it was only in the 1930s that it was realized that seizures are the result of out-of-control electrical activity in the brain. By placing electrodes on the scalp, doctors can identify when and where in the brain a seizure begins. But they cannot tell much about how the seizure behaves, that is, how it starts, stops or spreads to other areas. This makes it difficult to control and prevent seizures. It also helps explain why almost a third of patients with epilepsy continue to have seizures despite being on medication.

Saggio, Crisp et al. have now approached this problem from a new angle using methods adapted from physics and engineering. In these fields, "dynamics research" has been used with great success to predict and control the behavior of complex systems like electrical power grids. Saggio, Crisp et al. reasoned that applying the same approach to the brain would reveal the dynamics of seizures and that such information could then be used to categorize seizures into groups with similar properties. This would in effect create for seizures what the periodic table is for the elements.

Applying the dynamics research method to seizure data from more than a hundred patients from across the world revealed 16 types of seizure dynamics. These "dynamotypes" had distinct characteristics. Some were more common than others, and some tended to occur together. Individual patients showed different dynamotypes over time. By constructing a way to classify seizures based on the relationships between the dynamotypes, Saggio, Crisp et al. provide a new tool for clinicians and researchers studying epilepsy.

Previous clinical tools have focused on the physical symptoms of a seizure (referred to as the phenotype) or its potential genetic causes (genotype). The current approach complements these tools by adding the dynamotype: how seizures start, spread and stop in the brain. This approach has the potential to lead to new branches of research and better understanding and treatment of seizures.

League Against Epilepsy states: 'Because current knowledge is insufficient to form a scientifically based classification, the 2017 Classification is operational (practical)' (*Fisher et al., 2017*). In effect, that classification is based upon the epilepsy phenotype—the clinical symptoms that arise during a seizure. Over the past several years, a separate approach has arisen: investigating the genotype of specific epilepsies (*McGovern et al., 2013*; *Epi4K Consortium et al., 2013*), which may lead to more informed treatment decisions that match deficits with mechanisms. And for decades, intractable epilepsy has been treated with epilepsy surgery, which relies upon the high spatial resolution of imaging and implanted electrodes to find a seizure focus. These approaches are based upon our available tools: clinical expertise, genetics, imaging, pathology, and surgery. However, seizures are by definition dynamic phenomena, and none of these tools characterize the fundamental dynamics of seizures.

In this work, we introduce an organizing principle of seizure dynamics based on nonlinear dynamics and bifurcation theory. Bifurcations are sudden qualitative changes in behavior, including onset and offset of oscillations. Here, we introduce the term 'dynamotype' to describe a seizure's composite, observable, dynamic characteristics in electrophysiological recordings comprising seizure onset and offset. Together, dynamotype, phenotype and genotype provide a rich, multifaceted description of the dynamics, clinical manifestation, and underlying pathology of a seizure. The organization of seizures along dynamotypes leads naturally to a Taxonomy of Seizure Dynamics (TSD) providing practical, objective metrics for classification. As the periodic table of elements is a tabular display of chemical elements arranged according to proton number and electronic configuration, TSD is a tabular arrangement according to bifurcation type of seizure onset and offset. Furthermore, the organization of the periodic table can be used to derive relationships between the various element properties and predict chemical properties and behaviors of undiscovered or newly synthesized elements. Here, we explore the capacity of TSD to fulfill this second functional part of the analogy also and demonstrate the existence of all dynamotypes in human epilepsy. We then discuss TSD in the context of a canonical model in nonlinear dynamics and identify the relations amongst the seizure dynamotypes. TSD is available for immediate transfer to clinical practice, providing a rational

method of characterizing seizures and subsequently a better understanding of the underlying principles governing seizure generation and termination.

The basis of TSD is the observation that seizures are characterized by abrupt changes in the EEG waveform at seizure onset and offset, which we interpreted as bifurcations known from dynamic system theory (*Lopes da Silva et al., 2003a*). As a seizure evolves, the brain moves from a normal state into a seizure and back again. Recent work has focused on describing these transitions with empirically chosen visual patterns and has found interesting relationships with underlying pathology (*Perucca et al., 2014*), surgical outcome, (*Jiménez-Jiménez et al., 2015*; *Lagarde et al., 2016*) and sudden unexpected death in epilepsy (*Rajakulendran and Nashef, 2015*). However, these transitions can also be described more rigorously and mathematically as bifurcations. Bifurcations represent qualitative changes that both define and constrain the system dynamics (*Strogatz, 2015*). The concept has been used to understand neuronal firing: when a neuron goes through a bifurcation, the emergent dynamics often comprise a new set of behaviors such as multiple stable fixed points (rest activity) or limit cycles (oscillatory activity) (*Izhikevich, 2000*). When a neuron oscillates quickly about a limit cycle, it produces fast repetitive activity known generically as bursting, which is a periodic wave form with periods of oscillatory spiking and periods of quiescence. These dynamics can be described by a set of differential equations and accompanying parameters. The concepts about periodic wave forms with alternating periods of oscillations and quiescence have been extended to EEG wave forms in seizures (*Jirsa et al., 2014*), where oscillatory (ictal) states and quiescent (non-ictal) states alternate. Two variables are the minimum necessary to generate oscillations, and in systems with two variables, there are only six types of bifurcations involved in bursting (*Figure 1*, for further details also see *Saggio et al., 2017*). Four can be used to enter the bursting regime, and another four to exit, giving a total of 16 possible dynamotypes (*Izhikevich, 2000*). A key benefit of this organizing principle is that it unambiguously identifies the invariant properties of individual events, which may provide mechanistic insight into the underlying causes and response to specific interventions. It also provides a model that not only accounts for the effects of noise on the system (*Suffczynski et al., 2005*) and multistability (*Lopes da Silva et al., 2003b*; *Milton, 2012*), but also generates a time series. Generalizing this to epilepsy, (*Jirsa et al., 2014*) proposed the existence of 16 theoretically possible dynamotypes (i.e. seizure types), and found one seizure offset bifurcation that was present across multiple species, brain regions, and pathologies, including a small cohort of humans. Based on that initial work, we now expand and present a taxonomy of seizure dynamotypes. In the following, we begin with the definition of the different types of seizures based on dynamics at their onset and offset. Then, we show that seizures recorded from different centers in the world can be rigorously classified, and how classification can be performed in daily clinical practice. Next, we introduce a canonical model in nonlinear dynamics (*Saggio et al., 2017*) with two important properties: 1) the model is canonical, which means that under certain mild conditions (see Materials and methods) the behaviors of other models of arbitrary physiological detail can be represented and explained by the canonical model; 2) the canonical model captures all dynamotypes in a single mathematical representation. Transitions between types can be obtained through an ultra-slow modulation of the model's parameters providing a map of the parameter space, which systematically predicts relations between dynamotypes, including a hierarchy across dynamotypes. We demonstrate from a large repertoire of empirical data that patients navigate this seizure map to express the different types of seizures. Finally, we discuss how TSD can be used in a wide range of novel applications in clinical care and research.

## Materials and methods

### Classification of seizure dynamics

The goal of this work is to characterize seizures by their underlying onset and offset dynamics, which depends upon identifying reliable, canonical dynamic features. While the dynamics of a single neuron have already been described (*Izhikevich, 2000*), linking that behavior to a seizure generated by millions of neurons is complex. We chose to analyze the EEG signal from standard intracranial electrodes, as it is the most clinically relevant and widely studied method to measure brain dynamics. These patterns, which are visible within the EEG waveform, identify the bifurcations that define the invariant properties, and thus the first rigorous classification of seizure dynamics. *Figure 1*

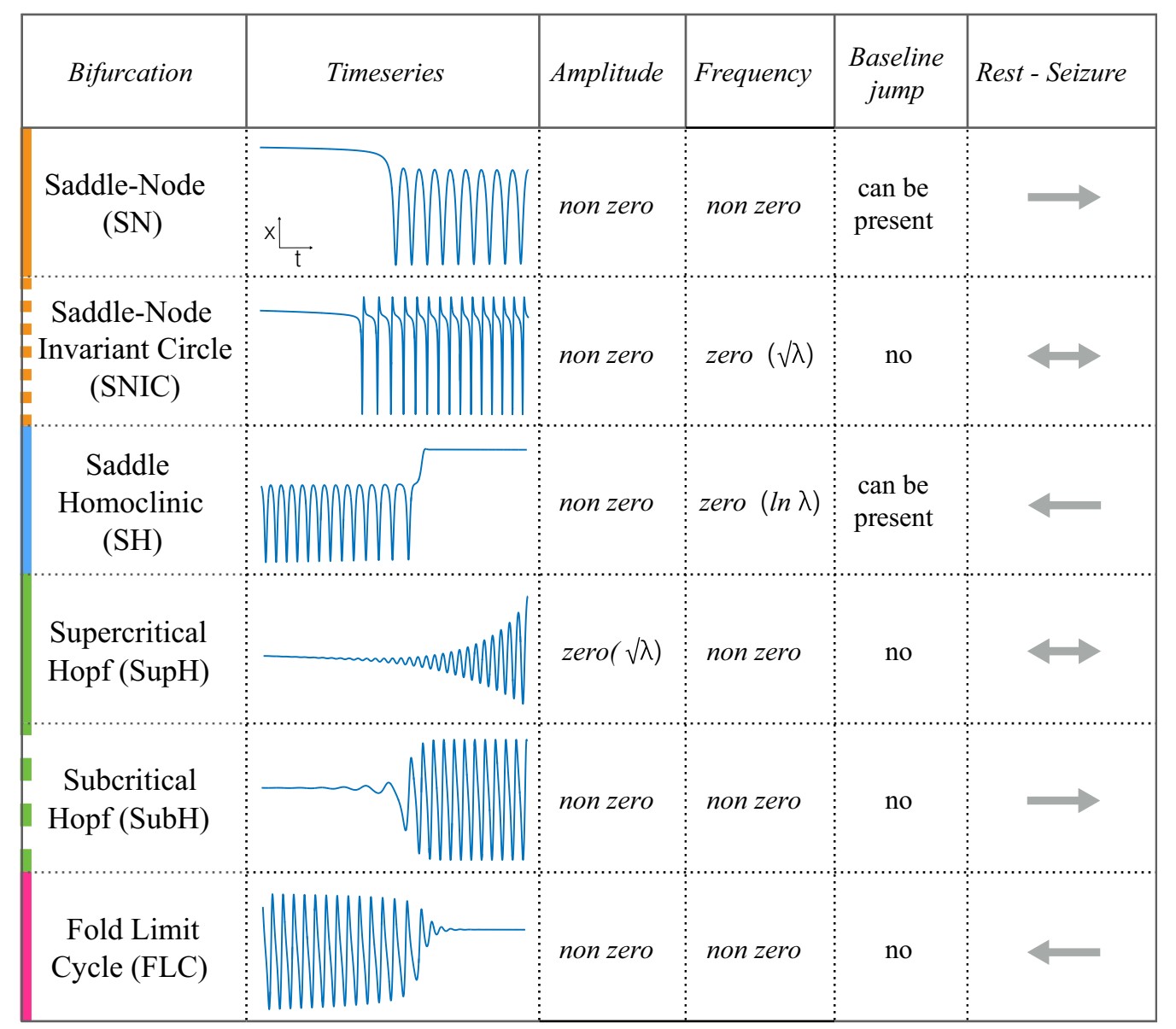

**Figure 1.** Scaling-laws of bifurcations. Six bifurcations are responsible for the transition from rest to seizure and vice-versa. For each bifurcation we report: name and abbreviation; an example of timeseries; whether the amplitude or frequency of the oscillations goes to zero at the bifurcation point, and if they do how they change as a function of the distance to the bifurcation point ($\lambda$); whether the baseline of the signal shows a baseline shift; and if the bifurcation can be used to start ($\rightarrow$) or stop ($\leftarrow$) a seizure or both ($\leftarrow\rightarrow$).

demonstrates these different bifurcations, showing how the signal changes in terms of amplitude and frequency of successive spikes and may contain a shift in the baseline as the seizure starts or stops. Of note, in dynamical terms, a 'spike' is defined as any prominent sharp transient associated with the dynamical process. For human EEG, we assume this includes all fast transients < 200 ms with amplitude that is distinguishable from the background. Of note this dynamical definition also includes the fast, low amplitude spiking seen at the beginning of some seizures.

In this work, we present algorithms to measure the invariant properties, which can then be used to classify the seizure types. While the theory behind this classification has been proven mathematically (*Kuznetsov, 2004*), measurement of these values under real conditions is challenging. This is because 1) EEG recordings of the brain are much more complex than single bursting cells, 2) EEG is

notoriously noisy, and 3) there is limited understanding of the underlying physiology that produces the EEG waveforms (*Einevoll et al., 2013*; *Reimann et al., 2013*). Despite these limitations, we previously found strong evidence that at least one dynamotype exists across multiple species (*Jirsa et al., 2014*). Herein, we present both an automated algorithm and a visual method to analyze these noisy data. We find that visual analysis is quite reliable and often preferable under clinical conditions, as demonstrated by recent work in other noisy neural signals (*Haddad and Marder, 2018*). Therefore, while we do present the algorithm results as validation, the final clinical analysis is based upon the visual classifications.

## Onset types

As described in *Figure 1*, there are four onset bifurcations. Two of these progress gradually from resting state into seizure via specific scaling laws for the amplitude or frequency (*Strogatz, 2015*): in the supercritical Hopf bifurcation (SupH), the amplitude of the oscillations starts at zero and increases proportionally to the square root of the distance from the bifurcation point; the Saddle-Node on an Invariant Circle (SNIC) bifurcation has oscillations that increase in frequency as the square-root of the same distance. The other two, Saddle-Node (SN) and subcritical Hopf (SubH) bifurcations, have abrupt amplitude and frequency changes that do not follow specific scaling laws. The SN can contain a jump in the signal baseline (i.e. direct current (DC) shift), but in the absence of a detectable DC shift these two types can be difficult to distinguish even theoretically. SN without DC shift ('SN (-DC)") and SubH are thus grouped together in this work. Further demonstration of the generation of these time series can be found in *Jirsa et al., 2014*; *Saggio et al., 2017*.

## Offset types

Two of the bifurcations are characterized by decreasing frequency, following square-root scaling for the SNIC and logarithmic scaling for the Saddle-Homoclinic (SH) bifurcation. Both these dynamics manifest as slowing of the seizure down to zero near its end, which is the well-known clinical hallmark of a seizure (*St. Louis and Frey, 2016*). This 'slowing' at seizure termination has been identified as an inherent characteristic of seizures across all spatial scales (*Kramer et al., 2012*), and in multiple species and brain regions (*Jirsa et al., 2014*). Due to the small number of spikes near the end of seizures, it is difficult to distinguish between these two scaling laws when using only frequency of spikes (*Jirsa et al., 2014*); throughout this work, when we refer to 'slowing-down' it refers to logarithmic or square root scaling, implying scaling down to zero. The other two bifurcations do not require slowing at termination: the SupH bifurcation with square-root-scaled decreasing amplitude and no scaling law for the frequency, and the Fold Limit Cycle (FLC) that has no specific scaling law for either frequency or amplitude. Note that these two offset types may appear somewhat atypical to a clinician, as 'slowing down' is expected at the end of most seizures. The only dependence on a DC shift is that only the SH can have a shift, but the absence of a shift the slowing down pattern can be SH (-DC) or SNIC.

## Patient selection and seizure analysis to create human seizure taxonomy

We analyzed seizures from 120 patients recorded on intracranial EEG in seven centers worldwide (Appendix I.1) (*Ihle et al., 2012*; *Cook et al., 2013*; *Kanazawa et al., 2015*; *Wagenaar et al., 2015*) to identify the bifurcations at onset and offset. All patients had focal onset seizures. Results are shown in *Figure 2*.

## Accurate identification of bifurcations

The canonical features necessary to distinguish the bifurcations are the dynamics of the amplitude and interspike intervals (ISI) and the presence/absence of a DC shift (*Figure 2A*). We identified the spike timing and amplitude to allow for both visual and mathematical analysis (Appendix I.2). In order to validate the analysis, we first compared the results of three human reviewers (Appendix I.3) and an automated algorithm (Appendix I.4) in a gold-standard computational model that generated 60 seizures of each type (*Saggio et al., 2017*). These methods were very successful in distinguishing the different bifurcations in the model data, although we had to consider SN (-DC) and SubH as a single group for seizure onset, and SH (-DC) and SNIC as a single group for offsets because there is

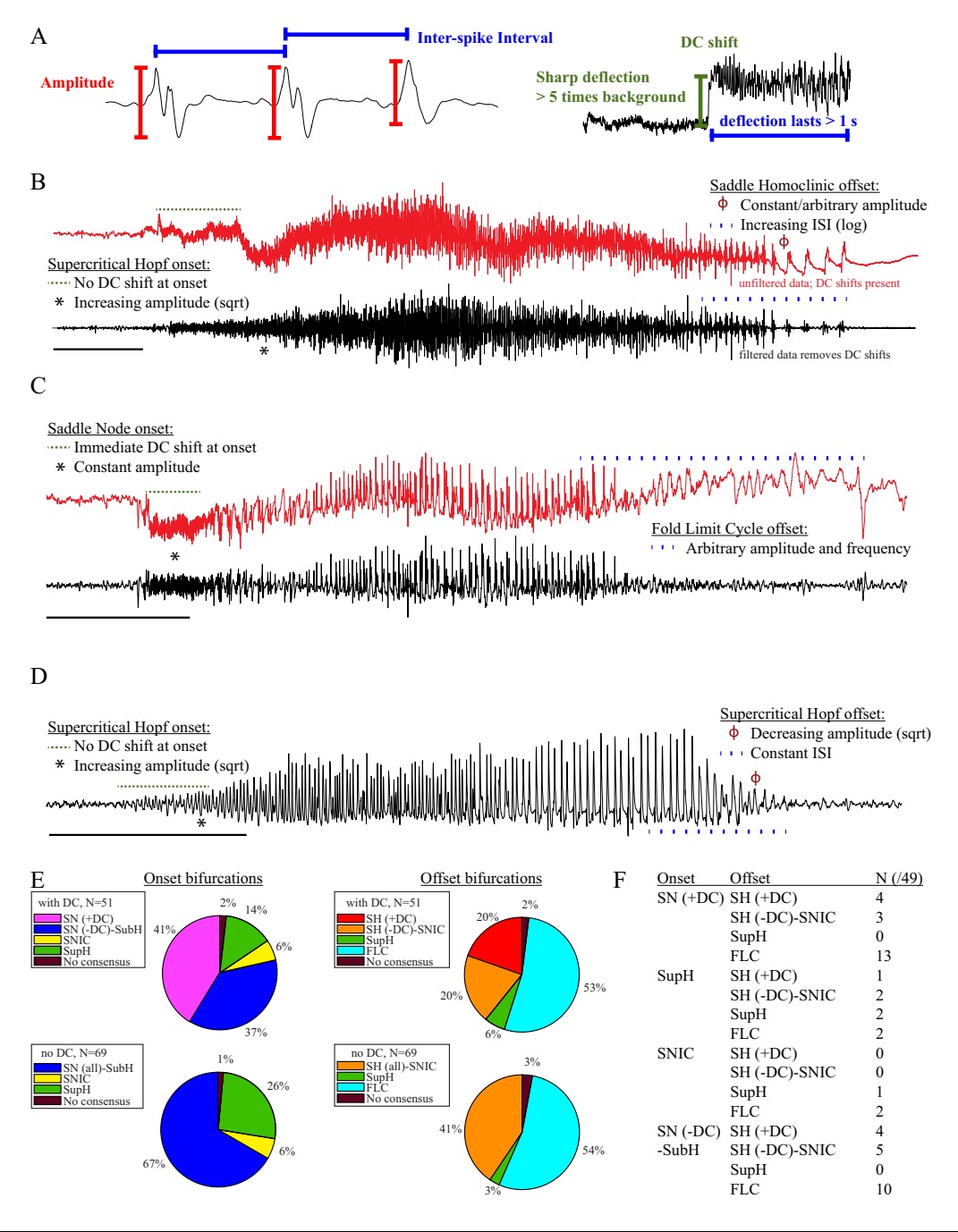

**Figure 2.** Seizure dynamics taxonomy. (A) Interspike intervals (ISI) and peak-to-peak amplitude were measured for every spike during seizures. DC shift was defined as sharp (<0.5 s) deflections that rise >5 times the background variance and persist for at least 1 s. (B) DC-coupled (red) and high-pass filtered (black) data of a seizure shows SupH onset and SH offset. (C) SN onset characterized by DC shift at onset. Ambiguous seizure offsets were included in the analysis as FLC. (D) SupH onset and offset. Although this patient did not have DC-coupled recordings, the amplitude scaling is clearly most consistent with SupH. Scale bars: 10 s. (E) Final results for all onset and offset bifurcations tested. All four bifurcation types were present. (F) Final taxonomy of the 39 patients with onset+offset classifications. In patients with DC recordings, SH offset was sometimes distinguishable from SNIC.

no method to distinguish them. We then compared these same methods on 120 human seizures. We found that concordance was also reliable in human data (Appendix I.5). These results show that the chosen features are capable of distinguishing the different bifurcations reliably for both human visualization and algorithms, that human seizure dynamics are consistent with the modeled

bifurcations, and that human reviewers can use the methods described in the next paragraphs to classify onset and offset bifurcations reliably. In addition, we found that human reviewers were more reliable than the automated algorithm in noisy clinical data. We then used the human markings on the clinical data for the taxonomy in *Figure 2*. Note, however, that the identification of bifurcations from empirical data is notoriously difficult and generally cannot unambiguously prove that a given bifurcation is present, although it allows assessment of self-consistency. Further investigation may use additional tools, such as perturbations of the system, to corroborate these results further.

*Onset dynamics* - To analyze onset dynamics, we first investigated all 51 patients that had been recorded using equipment capable of visualizing DC shifts. Of note, DC shift alone has previously been shown to be highly correlated to the seizure onset zone (*Ikeda et al., 1996*; *Ikeda et al., 1999*; *Kanazawa et al., 2015*). Many seizures (41%) started with constant amplitude spikes and a DC shift, signifying SN bifurcation (*Figure 2E*). The second most frequent was similar amplitude/frequency dynamics without a DC shift (either SubH or SN, 37%), followed by SupH (14%) and SNIC (6%). Ambiguities in the classification were treated systematically as detailed in the Appendix (I.6), and only 1/51 seizures could not be agreed upon by the reviewers. Despite the inability to distinguish SN (-DC) and SubH, we demonstrate that at least three of the predicted bifurcation types are present at onset in human seizures. Importantly, some seizures display complex dynamics because they go through more than one bifurcation as the seizure begins. For example, nearly half of the seizures labeled as SN onset progressed into square-root amplitude scaling after 2–5 s, consistent with a switch to SupH dynamics; we only labeled the initial bifurcation herein. For this reason, we have slightly modified the definition of onset bifurcation as compared to the literature. Here the onset bifurcation is the one causing a departure from the resting state, even if it is not directly causing the onset of oscillations as there can be intermediate states (Appendix II.1).

We next looked at the onset bifurcations in the other 69 seizures that were recorded with non-DC coupled hardware. Of the four onset bifurcations, SN and SubH become indistinguishable without a DC shift available. Combining these two into a single group, we found that 67% were SN-SubH, 26% were SupH, 6% were SNIC, and only one seizure did not have reviewer consensus.

*Offset dynamics* – We first examined offsets in those patients with DC recordings for the most robust classification. The analysis of the interspike intervals (ISI) and spike amplitudes revealed a logarithmic/square-root slowing-down with constant amplitude in 20/51 patients, and of those 10 had DC shifts at offset. Thus, 10/51 (20%) were SH (+DC), while the remaining 10 (20%) were potentially SH(-DC) or SNIC (*Figure 2F*). The remaining 31 patients did not have slowing at the end of their seizure (53% FLC, 6% SupH, 2% no reviewer consensus).

For non-DC coupled data, we grouped SH and SNIC in the remaining 69 patients. The majority of seizures had arbitrary ISI and/or amplitude (54% FLC), while 41% had slowing down characteristic of SH(-DC)-SNIC. The remaining seizures either had constant ISI with amplitude that decreased as a square root (SupH, 3%) or had no consensus among reviewers (3%). To supplement the examples in *Figure 2*, as well as to clarify how to approach several challenging scenarios, there is a primer with examples of all the different bifurcations in Appendix I.7.

## Analyzing arbitrary dynamics

One significant challenge in the above analyses was the presence of noise. Within the theoretical model, 'arbitrary' dynamics refer to abrupt changes at seizure onset/offset without clear scaling laws to or from zero. We would still expect a smooth behavior for amplitude and frequency close to (not at) the onset/offset point, and specific trends (increasing, constant, decreasing) are possible even though not prescribed (Appendix V). However, in human data analysis 'arbitrary' includes a wide range of other behaviors, especially noise. It is important to note that the taxonomy above includes the first seizure that could be analyzed from every patient—we did not restrict the analysis to 'clean' seizures. Some of the seizures were noisy, either from technical concerns or physiological effects of the seizure. We chose this method in order to provide a robust, real-world demonstration of this analysis. Because noise can be classified as 'arbitrary,' this analysis may overestimate the numbers of SubH-SN (-DC) onsets and FLC offsets. However, it was clear that this limitation was not merely technical: several patients had complex physiological dynamics. Four of the FLC seizures were highly unusual from a clinical perspective: one had increasing amplitude for the last 10 s, one consisted of low-voltage fast activity that ended abruptly without any other change, one ended with irregular

spike waves, and one had accelerating frequency at the end (Appendix I.8 and V). These examples highlight the vast heterogeneity of seizures, as well as the need for a valid taxonomy that allows for scientific discussion of the critical dynamical properties.

## Results

### Taxonomy of Seizure Dynamics (TSD)

Sixteen seizure dynamotypes– The preceding data validate that at least three types of onset and offset are systematically present in human focal epilepsy. As detailed above, real clinical data are challenging: the lack of DC shift makes it difficult to distinguish some bifurcations, and the noisiness of EEG is hard to distinguish from arbitrary dynamics. Nevertheless, these results show robust evidence that human seizures conform to both the onset and offset bifurcations predicted by our framework. These combinations lead to a taxonomy containing 16 dynamotypes of electrographic seizures (*Jirsa et al., 2014*). We identified the dynamotypes in patients with DC recordings. Two patients did not achieve reviewer consensus (one onset, one offset), leaving 49 patients (*Figure 2F*). We identified 12 different dynamotypes, with the limitation that several of the dynamotypes cannot be fully distinguished in the absence of a DC shift (e.g. SH(-DC) – SNIC offsets). The taxonomy was dominated by seizures with either SN or SubH onsets and slowing (SH-SNIC) or arbitrary dynamics (FLC) at the end. In this cohort of focal onset seizures, the SupH and SNIC onsets were less common, accounting for all four dynamotypes that were absent.

### Correlation between clinical data and seizure dynamotype

We compared all available clinical metadata from patients with their dynamotype and found no correlation between seizure type and patient gender, pathology, or localization. There was a correlation with age, as older patients tended to have more SupH onsets (Appendix I.9). We also compared these results with a prior visual classification that identifies seven basic seizure onset patterns (*Perucca et al., 2014*), and found 6/7 patterns without any apparent relationship to clinical data or pathology (Appendix I.10). There were no significant similarities between the dynamotype and the visual classification.

### Seizure dynamics vary in human epilepsy

While analyzing this dataset, we noted that one patient had two consecutive seizures belonging to different types: one supH/supH and one supH/SH, raising the possibility that an individual may express different types of seizures. This finding was surprising, as many clinicians assume that a person's seizure should be 'stereotyped', that is consistent over time. In fact, multiple medical devices have been designed under the presumption that a patient's seizures would have similar appearance over time (*RNS System in Epilepsy Study Group and Morrell, 2011*; *Cook et al., 2013*). To test whether individuals display different types of seizures over time, we used a unique dataset from Melbourne in which patients had intracranial EEG recorded continuously for many months (*Cook et al., 2013*). We analyzed over 2000 seizures from 13 patients. Given the size of the sample, we limited the analysis to the most straightforward metric: the ISI at seizure offset to determine whether there was slowing at seizure termination. This allowed us to differentiate the SH/SNIC from the supH/FLC bifurcations (i.e. slowing-down or constant ISI). There were 658 seizures of sufficient length (>25 s) to measure the offset ISI. To be conservative, we only classified seizures as slowing-down or constant if such a determination was unequivocal and labeled the rest as 'not assessed,' meaning that the determination was not readily evident on brief visual inspection. As seen in *Figure 3*, all 13 patients expressed at least two offset patterns. Note that this is likely an underestimation of the heterogeneity in seizure types: these recordings did not contain DC coupling so onset bifurcations were not assessed, and we did not distinguish the offsets into all four types. Nevertheless, we can unambiguously conclude that individuals have seizures from different dynamotypes over time.

### Organizational principles of seizure dynamics based on bifurcation theory

The clinical data above show that seizures can be classified based on their onset/offset bifurcations and that a patient's seizures may display multiple dynamotypes. To gain a deeper understanding of

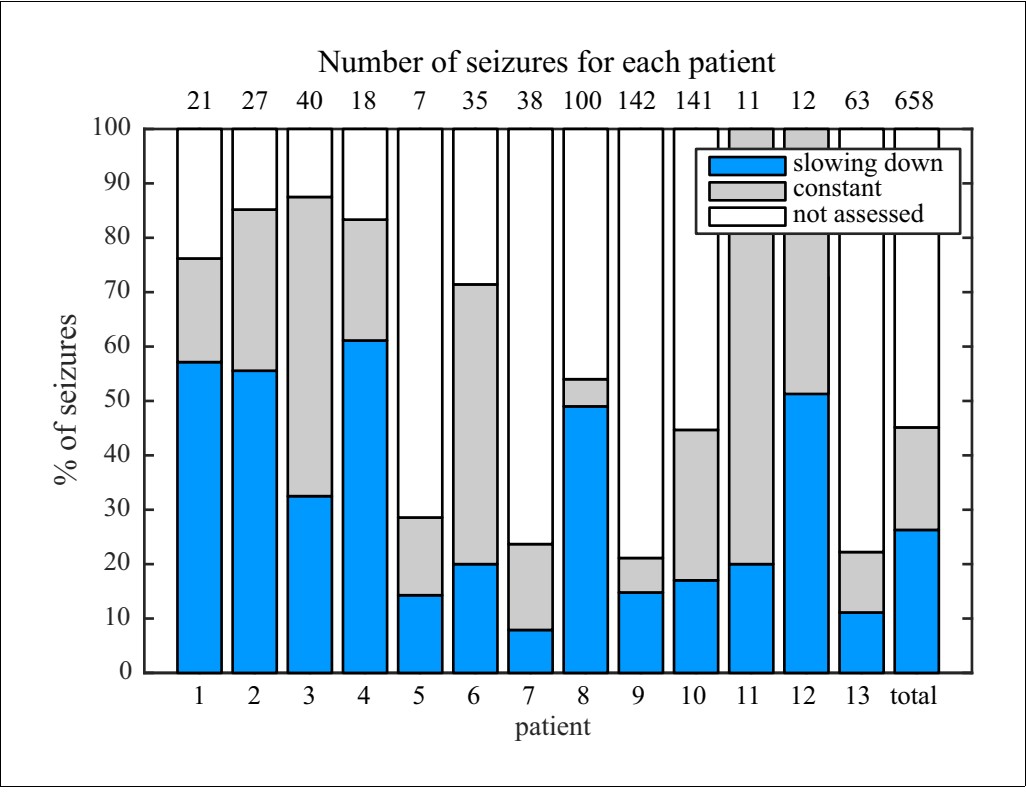

**Figure 3.** Manual classification of seizures from long-term intracranial recordings. As a conservative analysis, colored bars show seizures with unequivocal slowing-down (blue) or constant (grey) scaling, demonstrating that all 13 patients had both types of seizure offset. Additional seizures that had more difficult classification (white) were not needed for this analysis, as each patient had already demonstrated both offset types.

the relations between dynamotypes, we formalize these findings within a single unifying mathematical framework, which can account for all these behaviors. During seizures, the firing activity of neurons becomes organized, enabling the emergence of oscillatory activity that can be observed in electrographic recordings. This greatly reduces the degrees of freedom necessary to describe the observed activity, that is a small number of differential equations are sufficient to describe the collective behavior (*Figure 4A*). We here consider a system with the minimum number of variables necessary to produce oscillatory activity, two. Based on the parameter values, two states can be distinguished: resting (fixed point) or oscillatory (limit cycle). When these two states coexist for the same range of parameter values (bistability), transitions between them can be promoted by noise if the system is sufficiently close to a bifurcation (*Lopes da Silva et al., 2003a*; *Kalitzin et al., 2010*). However, the statistics of ictal durations (*Suffczynski et al., 2006*) points to the existence of a deterministic process governing this transition for seizure offset and possibly for the onset. This can be achieved with the addition of a third variable acting on the timescale of ictal duration. We previously validated this approach with the 'Epileptor,' a set of five differential equations able to account for the most dominant dynamotype (*Jirsa et al., 2014*) (SN/SH, also known as 'square-wave' bursting *Rinzel, 1987*). In the Epileptor, the transition from 'normal' to seizure state and back again, as well as the seizure dynamics, are controlled by a collective permittivity variable that evolves on a slow time scale. However, the Epileptor accounts in principle for only a single dynamotype and is not sufficient to explain the data presented above (note that systematic parameter variations also show a range of other bifurcations *El Houssaini et al., 2020*, although these variations are model-specific and not canonical). The fact that individuals can express different types of seizures over time leads to two predictions that must be included within a model of human seizure dynamics: different dynamotypes must coexist in the same model, and there must be an endogenous mechanism by which the brain can transition slowly between dynamotypes.

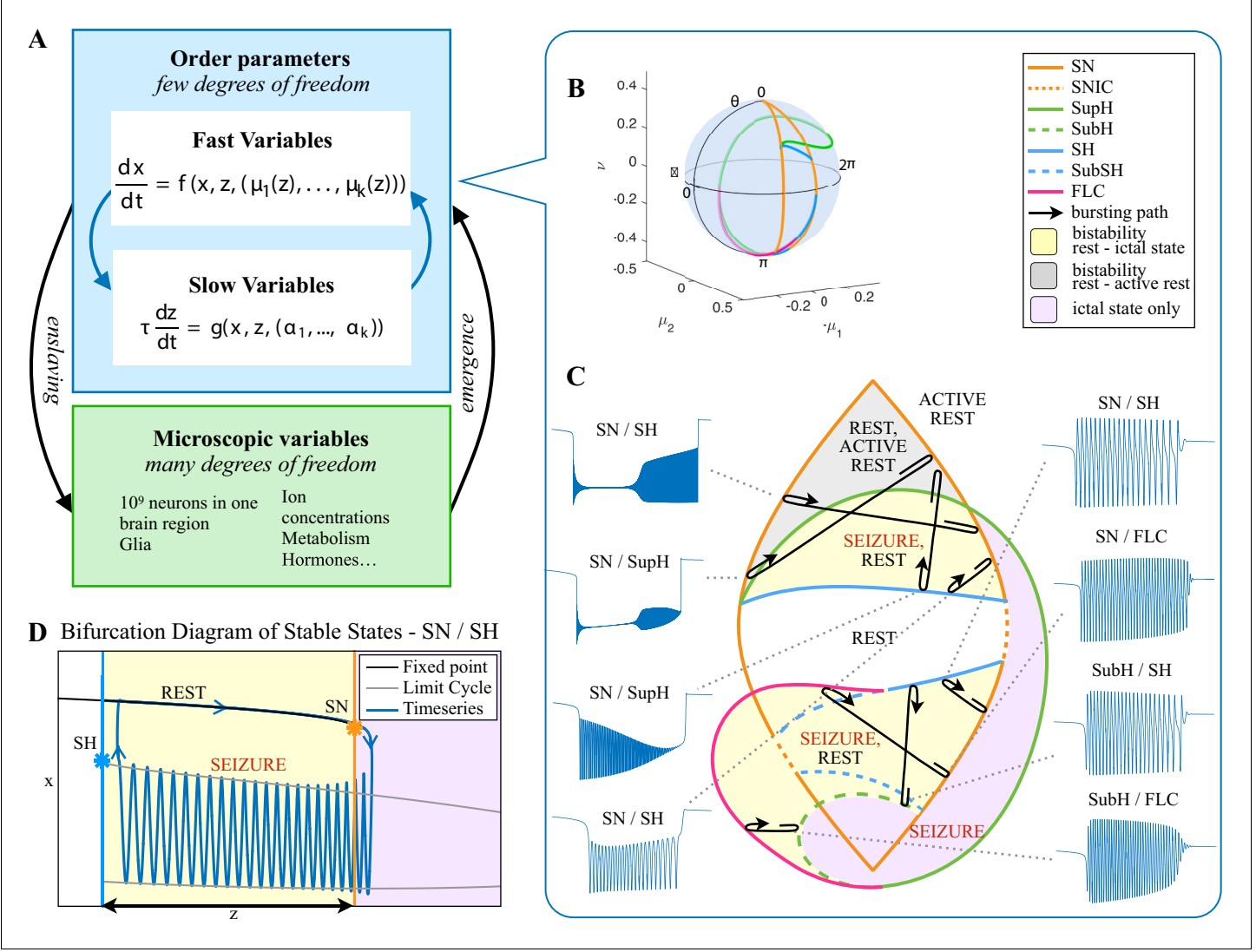

**Figure 4.** Modeling seizures. (**A**) During a seizure the microscopic variables, which compose a brain region, organize so that the emergent global activity can be described by a few collective variables. These collective variables, on the other hand, act on the microscopic variables, 'enslaving' them. (**B**) The fast variables can be in different states depending on the values of the parameters. In our model, we have three parameters $(\mu_2, -\mu_1, \nu)$; however, the relevant dynamics occurs on a sphere with fixed radius. We thus consider the two-parameter spherical surface $(\theta, \phi)$ that can be sketched with a flat map as shown in C. (**C**) Bifurcation curves divide the map in regions where different states are possible: healthy state (white), ictal state (violet), coexistence between healthy and ictal states (yellow), coexistence between healthy and 'active' rest (a non-oscillatory state with a different baseline than the healthy state (gray)). When a seizure starts because the system crosses an onset bifurcation, the slow variable enables movement along the black arrow (a path in the map) to bring the system back to rest. Note that in this model, the system alternates between the resting and seizing states within the bistability regions. The shape of the arrow is meant to better show the trajectory followed by the system, however movement in the model occurs back and forth along the same curve. Insets show example of timeseries for different paths. SubSH: Subcritical Saddle Homoclinic, an unstable limit cycle that occupies a small portion of the map but is incapable of starting/stopping seizures, and thus is not included in *Figure 1* nor the rest of the analysis. (**D**) An expanded view of one trajectory followed in C. The path (double arrow) represents movement from resting to seizure state and back by crossing the bifurcations, which in this case are SN and SH. This activity forms a seizure in the time series. The resting state is represented by a black line (fixed point), the minimum and maximum of the amplitude of the seizure (limit cycle) by gray lines.

Addressing these predictions within the framework of bifurcation analysis provides the entry point to propose a general taxonomy of seizure dynamics and postulate the existence of an ultraslow modulation. Previous mathematical work demonstrated that the procedure to build a minimal model for the SN/SH type (*Golubitsky et al., 2001*) provides a two-dimensional map for the parameter space of the fast variables on which all the six bifurcations can be placed (*Dumortier et al., 1991*). In effect, the map is a representation of the range of states in which a brain region can exist,

oscillatory (ictal) and non-oscillatory (interictal), and the transitions between them. The oscillatory state produces spiking activity that is described by the fast variables (millisecond scale activity). However, on a slower time scale of the order of seizure length, the brain can move toward a transition to an interictal state, as described by a slow variable. Migrations to different locations on the map can occur on a usually even slower timescale (10's-1000's of seconds), which we call here 'ultraslow'. Saggio et al. showed that the use of an ultraslow variable allows full exploration of the map (*Saggio et al., 2017*). Applying these general mathematical principles to epilepsy implies that any brain region able to generate SN/SH seizures can potentially generate other types by navigating to different dynamical regimes (i.e. changing the parameters of differential equations). That work also showed that a large number of physiological neuron and neural population models can be mapped upon a canonical dynamic model under certain mild conditions (existence of a Bogdanov-Takens point). All physiological parameters are then absorbed in only three generic parameters, which span a three-dimensional parameter space, in which all bifurcations are represented. Detailed bifurcation analysis reveals that all neighborhood relations between bifurcations can be displayed (without loss of generality) as projections onto the spherical surface within the parameter space, yielding a canonical 2D map (*Saggio et al., 2017*), shown in *Figure 4B–C*. This map displays the basic topology of all possible relations between bifurcation lines, including identity of the bifurcation and the organization of its nearest neighborhood including proximity and intersection of bifurcations. As one (or multiple) of the parameters is continuously varied, trajectories are traced out in this map, eventually connecting two bifurcation lines and thus establishing a seizure's dynamotype. The navigation of the canonical 2D map (that is a flat projection of a spherical surface) through continuous parameter changes selectively generates dynamotypes and justifies that we call it a *seizure map*. The state of a brain region at any moment can be represented as a location on the map, which defines its dynamical properties. Regions in the map that correspond to different regimes, including quiescent and ictal states, are separated by bifurcation curves. Seizures are represented as black arrows, each arrow corresponding to one dynamotype. To produce a seizure, the system, which is initially in the quiescent state within the bistability region, heads toward the onset bifurcation curve. When this curve is reached, the quiescent state disappears and the system is forced to go into the oscillatory seizure regime within the bistability region. This transition in state causes an inversion in the trajectory of brain state, with the system now heading toward the offset bifurcation curve. When the offset is reached, the system goes back to rest and inverts direction again. Movement along the black arrow is produced by slow (of the order of the ictal length) mechanisms leading to seizure offset. Note that the movement towards the onset and offset bifurcations at this timescale occurs in both cases from within the bistability region. Ultraslow movements on the seizure map are responsible for changing the location of the brain state while at rest (as may happen during the night and day cycle) and enable the expression of different types of seizures as observed clinically. This framework thus provides a potential explanation for the clinical observation of multiple types of seizures in a single patient, and the seizure map provides a hypothesis to describe how a patient's current state (i.e. location on the map) can affect seizure dynamics (whether a seizure is likely to occur, and what type is most likely). *Figure 4C* depicts paths (black arrows) for seven of the 16 dynamotypes placed on a two-parameter map. Adding one additional parameter allows this map to be extended and create seven other types in three dimensions, while the final two types require even higher dimensions to create (*Saggio et al., 2017*). These higher dimensional types require very fine parameter tuning, and thus are less likely to occur (*Golubitsky et al., 2001*; *Saggio et al., 2017*). TSD does not predict the likelihood of dynamotypes, but in conjunction with the seizure map and choice of slow dynamics, a hierarchy of seizures can be established, which is supported by our clinical data (Appendix II. 5), for instance the dynamotypes that occurred the most (e.g. SN/FLC and SN/SH) were predicted to be among the most likely to occur.

It is important to distinguish the two types of fluctuations within the brain map. The slow permittivity variable affects the general brain state, or position on the map, on the scale of minutes to hours. It represents underlying, and sometimes varying, conditions of the system than determine the position in the map, which has broad physiological implications. There are also fast fluctuations on the scale of ms to s, better described as perturbations of the brain state from its current location on the map. These perturbations are modeled as 'noise' within the model, but in reality they also include many physiological phenomena such as afferent signals and neural potentials—in effect anything that perturbs the system (*Jirsa et al., 2014*). Thus, although the model refers to the addition

of 'noise' to the system, these effects can readily be attributed to physiological neural activity. Both types of fluctuations can lead to seizures by pushing the system across the bifurcation.

## Ultraslow fluctuations to navigate the seizure map

It is important to note that the topology of the map in *Figure 4* was initially proven mathematically to be generic and rigorously valid for bursting (*Dumortier et al., 1991*; *Baer et al., 2006*). This invariance establishes the ground truth to define the relationships between the different bifurcations in the proximity of the SN/SH type, which leads to a key prediction: transitions between certain types may be more common due to their proximity on the map. For example, considering the bistability region in the upper part of the map, we note that the offset curves of SH and SupH approach until they meet. When the curves are very close, even small fluctuations in the parameters can cause a transition between types. If fluctuating internal conditions allow individuals to move around these regions of convergence, patients may have seizures belonging to different types over time, as observed in our longitudinal analysis. The model predicts that transitions between specific types are more likely to occur if these types are close in the map, in the sense just shown of bifurcation curves approaching each other within the same bistability region. On the contrary, transitions between types belonging to distant bistability regions require stronger changes in the ultraslow permittivity variable(s) and are thus less likely to occur.

Two dynamotypes are paired when they share the same seizure onset or the same offset, and a continuous change of parameters in the seizure map can lead from one type to the other, sometimes even during a seizure. The seizure map predicts possible pairings as motions toward different bifurcations while a seizure is ongoing. As proof of concept, we found several examples of such fluctuations in our cohort. In *Figure 5A*, one patient's seizure had constant ISI and square-root amplitude scaling for approximately 70 s, properties exhibited when approaching the SupH bifurcation. The seizure appeared to be terminating, but then it abruptly restarted to terminate with slowing-down ISI and constant amplitude (SH/SNIC bifurcation). We found five examples of this behavior in our data (out of >2000 seizures) and reproduced it with our model. By definition, the dynamotype includes only the onset and final offset bifurcation, but the behavior during this seizure is intriguing and can be explained by the model. We considered a path for the SN/SupH type with an ultraslow drift of the offset point that changed the path to SN/SH and added noise to all variables to simulate fluctuations. With these settings, we ran 100 simulations, which generated several different dynamotypes, predominantly SN/SH, SupH/SH, and SN/SupH (Appendix III). Several had transitions during the seizure from one bifurcation to another, 10 of which in the same manner as the data in *Figure 5A*, switching from SupH to SH offset. Thus, the clinical example is one of the most favorable combinations within the model, as the SupH and SH bifurcations are so close that small fluctuations can cause the switch.

Status epilepticus: Explorations of the seizure map also demonstrated another effect sometimes seen clinically: status epilepticus. Simulations with the previous settings in some cases produced continuous seizures that did not resolve by the end of the simulation, equivalent to status epilepticus (*El Houssaini et al., 2015*). We analyzed the corresponding trajectories on the map to determine how this had occurred. Prolonged seizures occurred when the brain state crossed the SN onset curve but was unable to return to rest through the offset bifurcation and remained mainly in the violet 'seizure only' region in *Figure 5G*. The slow variable naturally drives the state toward offset, but in these cases was continually overridden by noise, causing the state to 'escape' from the bistability region. We then analyzed this effect by simulating various levels of noise and showed that there was a clear correlation between the noise variance and the likelihood of entering status epilepticus (Appendix IV). We compared these results with our clinical data, which had two examples of non-convulsive status epilepticus. In both cases, the seizures began in typical fashion, but instead of terminating began having long periods of constant ISI with varying periods of amplitude fluctuations. There were many abrupt transition periods during which the ISI and amplitudes became arbitrary. After these brief periods of disorganization, the dynamics returned to constant ISI. We compared the dynamics of our model results with these human seizures and found that the transition between different dynamics is quite similar. In *Figure 5D*, we show a portion of two human seizures and one example of the simulation (*Figure 5E–F*) and movement on the map (*Figure 5H–I*). Further demonstration of the patients' status epilepticus is provided in Appendix IV. The patterns of organized- alternating with disorganized- firing, often known clinically as 'waxing and waning seizures,' are entirely consistent

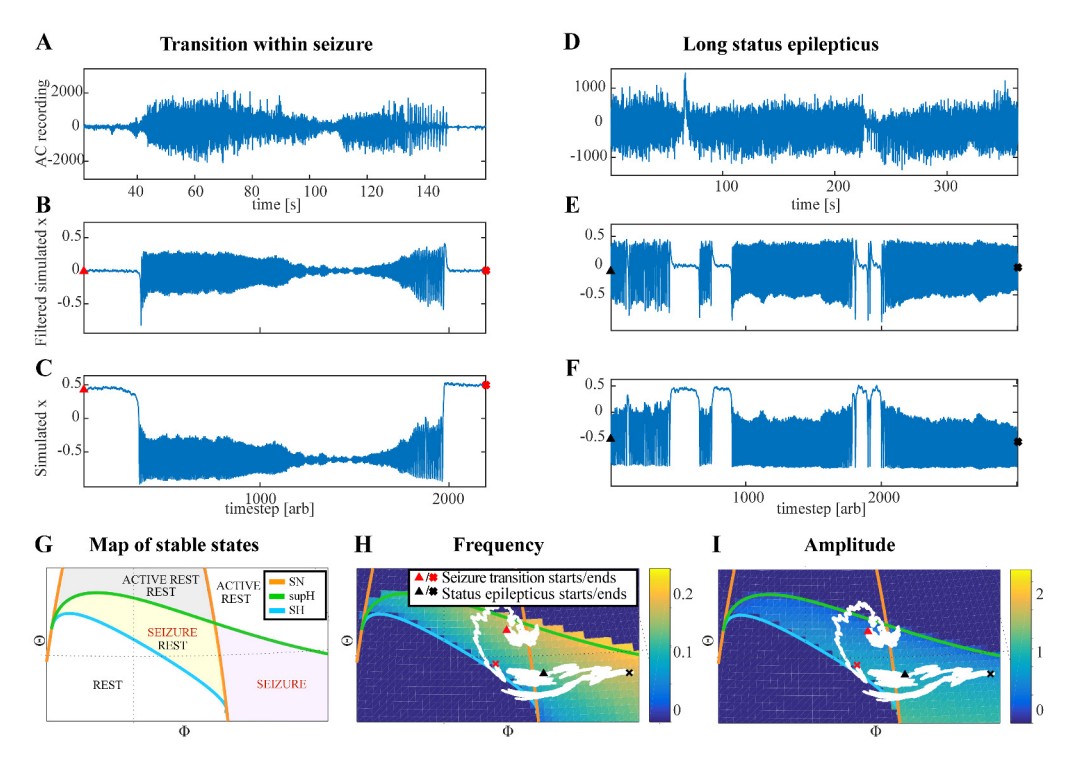

**Figure 5.** Fluctuations in the ultra-slow modulation of the path causes changes in the dynamics of the seizure. (**A**) A recording in which the seizure begins to go toward a SupH offset (square-root decreasing amplitude), but the amplitude increases again and the final offset is SH. (**B**) A portion of the simulation of the model in *Saggio et al., 2017* done to reproduce the dynamics observed in the recording. This timeseries is high pass filtered to simulate the effect of AC recordings. The non-filtered simulation is shown in (**C**). (**D**) A portion of a long status epilepticus recorded in one patient. The status epilepticus was characterized by transitions in the dynamics (such as at 70 s and 230 s). (**E**) High-pass filtered, and (**F**) unfiltered portion of model data simulating the behavior in D. (**G**) A zoomed, flattened projection of the map in *Figure 4B* (see Appendix II, IV), for reference in H-I. (**H,I**) Amplitude and frequency maps showing trajectories (white) of the modelled seizures in C,F. The seizure from C begins (red triangle) and naturally moves upwards toward the supH bifurcation, but a change in the ultraslow drift prior to termination pushes the system downward, changing the offset from SupH to SH. Conversely, the seizure from F begins (black triangle) and repeatedly moves towards the SH termination due to the ultraslow drift, but is pushed back into the seizure regime repeatedly by high levels of noise, which overrides the role of the slow variable in terminating the seizure.

with the model: the seizure undergoes periods in which it progresses toward termination, then because of noise it reverts to a point farther away from the offset bifurcation, as described previously (*Kramer et al., 2012*).

Accelerating seizure: We then analyzed the seizure offset that increased in frequency described in Appendix V. In this case, we explored conditions on the map that could produce 'speeding up' at the end of the seizure. We identified multiple trajectories in the map of brain states capable of producing these unusual seizure dynamics. As in the case of status epilepticus, these unusual patterns are dependent upon the relative position within the brain map, in this case occurring when brain states along a trajectory are affected by multiple bifurcations that are in close proximity (*Appendix 1—figures 17–19*). These results demonstrate the explanatory value of the seizure map and show how it provides a rational explanation for a wide range of physiological dynamics.

## Discussion

Seizures have been recognized clinically for millennia, but after nearly a century of electrographic recordings we still do not have a translatable method of characterizing their dynamics. We here address this issue and provide the first principled approach toward the organization of seizures in a Taxonomy of Seizure Dynamics (TSD). TSD establishes 16 dynamotypes of seizures, which could be extended to more exotic dynamotypes when considering non-planar bifurcations (see Appendix II.6). As TSD provides the classification of seizures, it remains completely unbiased to each seizure

type. This invariance is broken by the seizure map, which establishes relations between dynamotypes and introduces a bias in the taxonomy, laying the grounds for a hierarchy of dynamotypes. The hierarchy is based on the mathematical consequences of how bifurcations are related to each other (*Saggio et al., 2017*). The relations can be considered as structural in the sense that they rely on the static properties of location, shape, branching and topology of bifurcation curves in the seizure map. The implications are functional in the sense that they determine the non-static properties of a seizure's discharge patterns including frequency, acceleration/deceleration, amplitude and amplitude growth. As such TSD and the seizure map provide another example of the ubiquitous link of structure and function in biology.

Our classification aims at precisely identifying the seizure type in terms of dynamics, without any dependence upon specific symptoms, pathology, or localization. Thus, it is highly complementary to the classical operational classifications used by clinicians to diagnose and treat patients, which are based upon those factors without addressing dynamics (*Fisher et al., 2017*). The dynamotype describes the behavior of the seizure itself, while the clinical classification describes the patient's symptoms: together, both classifications are synergistic and can be used to improve patient stratification, providing more insight into diagnosis and treatment. TSD is based on simple, invariant, objective metrics that have compelling scientific rationale. With DC-coupled recordings, it is possible to distinguish the types with high fidelity, even with visual inspection. This method is thus readily available to clinicians, as many standard EEG acquisition devices now have excellent resolution near DC (<0.1 Hz).

Our interpretation of the results relies on some key assumptions. First, we are assuming that the onset and offset of seizures are brought about by bifurcations. Another common mechanism in the literature is noise-induced transitions (*Lopes da Silva et al., 2003a*), which our model can reproduce (see Appendix II.3 for a discussion of how this would affect our classification). Second, we rely on the assumption of timescale separation between the dynamics of the spikes within seizures and the slower dynamics controlling seizure threshold and termination (both in theory and in the simulated sample of data). If this assumption does not hold, different phenomena could occur and the scaling laws could be impossible to identify. Third, we only considered planar bifurcations for simplification (Appendix II.6). These assumptions were the axioms for developing the theory and data analysis. Future work will address the validity and consequences of these simplifying assumptions.

Similar to sleep, seizures are universal from insects to humans, leading to the proposal that seizures are an inherent property of a brain, that is they are endogenous to the brain, perhaps as an emergent property of complex neuronal networks (*Jirsa et al., 2014*). This may explain why, despite the vast range of genetic, structural, chemical, and developmental conditions that cause epilepsy, seizures have a remarkably limited set of dynamical behaviors. It is therefore not surprising that elementary mathematical laws can describe their electrophysiological signature. However, clinical interpretation of seizure dynamics has been almost universally based on simple observation, in which clinicians report the frequency and morphology of spikes. This method is helpful to identify primary generalized epilepsies, but within focal seizures has limited clinical use. TSD by itself does not have any bias either regarding the dynamotype, but when linked to the seizure map shows certain dynamotypes to be more prevalent than others. Empirically, low voltage fast activity (*Wetjen et al., 2009*) and focal DC shifts *Ikeda et al., 1999* have been found to be highly predictive of the true seizure focus, which are both patterns corresponding to common dynamotypes, suggesting clinical utility.

One limitation of previous clinical descriptions of seizures is that it has been unclear which dynamical features are relevant. There is high variability in the frequency and morphology of spikes due to individual fluctuations and noise, as seen in spontaneous seizures recorded in humans and experimental models (*Jirsa et al., 2014*). Our analysis identifies the invariant dynamics that impose important constraints on the system. A crucial aspect of our approach is that it allows us to disentangle characteristics that are necessary to describe the dynamics from other seizure related phenomena that are not fundamental to our simplified model (e.g. spike and wave complexes, preictal spikes, sentinel spikes (see Appendix I.2)). This is not to suggest that such biomarkers are not relevant to epilepsy: they have well-known correlation with the epileptogenic zone (*Conrad et al., 2020*) and can predict the occurrence of the first spontaneous seizure during experimental epileptogenesis (*Chauvière et al., 2012*). The Epileptor model, which comprises the SN-SH dynamotype, generates both spikes and seizures (*Jirsa et al., 2014*). The spikes are important indicators of the organization

of the networks, but not part of the generic features of onset and offset dynamics and thus not appear in the generic model (*Saggio et al., 2017*). If additional mechanisms are introduced based on other forms of reasoning, as in *Jirsa et al., 2014* via a second population, spikes can be included in the dynamics. It is important to note that we only analyzed the particular case of drug-resistant epilepsies investigated with invasive intracranial recordings; however, it is unlikely that this theory is specific to such epilepsies as seizures forced in non-epileptic networks follow the same universal rules (*Jirsa et al., 2014*). We note that prior work focusing on individual bifurcations are all also entirely consistent with TSD, which encompasses all these onset and offset possibilities and further shows how they interact (Appendix VII).

This work does not include data from generalized onset epilepsies, as these are not typically recorded with intracranial EEG. Our taxonomy is, however, fully consistent with past work on generalized epilepsy dynamics (*Wendling et al., 2016*). Absence seizures, for instance, begin with sudden onset and offset of ~3 Hz large amplitude oscillations without a DC shift (*Slaght et al., 2004*) and terminate abruptly without slowing down to zero, which would point to a SubH/FLC type (which is the most likely dynamotype (*Golubitsky et al., 2001*), see Appendix II.5). In this work with focal epilepsies, the most common dynamotypes were the SN (+DC)/FLC and SN/SH (which are also likely types).

When comparing our results with a past visual classification system of spike frequency (*Perucca et al., 2014*), we found no correlation with pathology in our cohort of 120 patients. However, our cohort did not have any patients with tuberous sclerosis, which was the only pathology associated with burst suppression in that prior work. Combining the data from both studies, the different patterns appear to be either evenly distributed or too rare to find robust correlations with pathology. Similarly, dynamotypes are not strongly correlated with pathology. In terms of the seizure map, we hypothesize that what determines the seizure dynamics is not the pathology per se, but the location of the brain on the map. Specific pathologies may predispose to certain regions, but there are many complex dynamics affecting brain state and many conditions that can produce similar dynamics. This coincides with the idea of the seizure map showing the full range of potential seizure onset and offset activity.

There is great clinical and research potential in characterizing a seizure's dynamotype, as it provides a unique perspective on brain networks. The current standards of epilepsy care focus on phenotype, genotype, and the time/location of seizure onset. While those methods have obvious utility, they do not address the underlying dynamics and thus have left several questions unanswered for decades. How do seizures start, stop and spread? How do we tell the difference between inter- or pre-ictal spiking and seizure initiation? Is it possible to measure the distance to seizure threshold, that is determine seizure risk at a given moment? How do we compare two different seizures? Is it possible to measure if a treatment is working by moving the brain 'farther away' from seizure onset, rather than waiting to see if seizures recur? These questions all require an understanding of the dynamics—an understanding that is not addressed by the current clinical tools. This is where the utility of the dynamotype is manifest. At its most basic, the dynamotype is a quick description of the key dynamics of a particular seizure, a clinical language that focuses on the aspects that are most important. This would supplement current visual descriptions, which typically are limited to amplitude and frequency. But there are many deeper applications of this tool as well. We have previously demonstrated that very different biophysical mechanisms can produce the same dynamotype (*Jirsa et al., 2014*). Here, we show that seizures from 120 patients contain almost the entire taxonomy of dynamotypes, and that a wide array of focal pathologies can be grouped into similar dynamotypes. Our interpretation is that this is because the seizures depend heavily on 'local' dynamics, that is the current brain state (location on the map) and acute perturbations (noise in the system), more than that a single pathology would necessarily predispose to a specific location on the seizure map.

There are many other potential applications for TSD in basic research as well. For instance, our group recently published an analysis quantifying how epileptogenesis progresses in the tetanus toxin model in rats (*Crisp et al., 2020*). That work showed that the dynamotype evolved over time, starting with SN and moving into SNIC (*Appendix 1—figure 4*) and sometimes SupH onsets over the course of weeks. A clinical trial is currently underway in France using the SN-SH dynamotype to model seizure foci and spread (*HBP, 2018*). Future versions of such tools could utilize the whole taxonomy to be much more comprehensive, tailoring models to the key underlying dynamics of specific

patients. These models would greatly enhance modern network analytic tools (*Stacey et al., 2020*), which would be greatly enhanced with a rational model to describe the underlying dynamics.

One novel aspect of dynamotype is that understanding the underlying dynamics can help in the design of strategies to control seizures, such as with electrical stimulation (*Kalitzin et al., 2010*). Studies on neuronal bursters (*Izhikevich, 2000*), which are organized in similar dynamic types, demonstrate that types have different sensitivity to stimulation. For example, SubH onset acts as resonators, which require a resonant frequency in the stimulus to trigger oscillations, while SN onset behaves as an integrator in which the nature of the stimulus (excitatory or inhibitory) rather than the frequency plays a key role (*Izhikevich, 2000*). There is a long history of using perturbation to probe the proximity of a nearby bifurcation in disciplines such as electrical power (*Chow et al., 1990*) and reservoirs (*Heppell et al., 2000*). Past work on stimulation to assess epileptogenicity (*Alarcón, 2005*; *Kalitzin et al., 2005*; *David et al., 2010*) is similar to such work and would be greatly enhanced with the insight gained from this model to understand the nearby bifurcations. There is a long history of bifurcation research in other fields that may also be applicable to seizures, such as using perturbations to assess proximity to a SubH or SupH bifurcation (*Bryant and Wiesenfeld, 1986*; *Vohra et al., 1994*; *Yaghoobi et al., 2001*). Further theoretical and clinical work is necessary to assess whether knowledge of the dynamotype could also improve the ability to abort seizures with tailored stimulation. The second important prediction is that the synchronization properties of coupled bursters are bifurcation-dependent (*Wang et al., 2011*; *Reimbayev and Belykh, 2014*; *Belykh et al., 2015*). This is a key issue for seizure propagation, as it predicts that the ability of a seizure to spread is dependent upon its type. Since the spatiotemporal organization of the seizure is part of the data features used to personalize brain network models (Virtual Epileptic Patient *Jirsa et al., 2017*; *Proix et al., 2017*) and functional connectivity based approaches (*Hutchings et al., 2015*; *Sinha et al., 2017*; *Taylor et al., 2017*), the choice of the right dynamotype is critical for successful patient modeling and clinical translation.

Another significant contribution of this work concerns the dynamics of the slow permittivity variable to explain how slow changes in the behavior/state of a brain region can bring it closer or farther away from different bifurcations, that is seizure threshold. The fact that all 13 patients had seizures belonging to at least two types implies that the permittivity variable moves on a dynamical map in which different types of bifurcations can occur. Each parameter of the map should be considered as a representation of a manifold of physiological variables that cooperate to produce a particular change in the system. Given the slow timescale at which these changes occur, neurochemical substances (e.g. hormones, neuromodulators etc.) are the best candidates. Within the permittivity variable we can here distinguish two timescales: a slow timescale of the order of the ictal length, and an ultra-slow timescale of the order of the interictal length (hours, days, months, years). Typical examples include the circadian regulation of seizures (*Karoly et al., 2017*) and catamenial epilepsy. Interestingly, both males and females display ultraslow (weeks) modulation of seizure probability both in rats (*Baud et al., 2019*) and humans (*Karoly et al., 2017*; *Baud et al., 2018*), further suggesting that these results are species- and sex-independent. Those results and ours strongly support the proposal that patients move closer and farther away from seizure threshold (i.e. 'travel the map') during their lifetime. This interpretation may also be helpful in assessing a brain's current proximity to seizure bifurcations, that is predict the risk of seizures occurring. Several features are altered when nearing the onset bifurcations, such as preictal spikes (*Jirsa et al., 2014*), variance of the signal (*Meisel et al., 2015*), and reaction to electrical probing of cortical excitability (*Freestone et al., 2011*). Recent work has shown that interictal discharges act like system perturbations that behave like the slow approach to bifurcations, just as predicted by our model (*Chang et al., 2018*). Seizure forecasting, based on electrographic recordings, is already enhanced when circadian rhythms are used to inform the model (*Karoly et al., 2017*). If the ultraslow physiological correlates of the map's parameter could be identified, measured, and manipulated, this would open new possibilities to assess when the patient is moving toward unsafe regions of the map and to alter their trajectory, that is control seizures before they occur.

Our proposed framework provides an organizational principle of seizure dynamics, which, when linked to canonical dynamic systems, identifies a generic seizure map charting out characteristics of dynamotypes including prevalence of a dynamotype and possible pairings of dynamotypes that can occur during a seizure, as well as others that are prohibited. TSD does not describe all possible seizure features, but relies on seizure onset and offset classification, which is why we defend a

complementary approach, combining it with traditional operational classifications. However, it provides a unique avenue to classify seizures based upon their key dynamical features, while providing insight into how seizures become more or less likely to occur at a given time. A corollary is that, although TSD has been developed in the context of seizure dynamics, it likely extends to physiological function of the healthy brain (e.g. alterations between REM and slow wave sleep, and the appearance of gamma frequencies or ripples during slow wave sleep) and stipulates the existence of at least two time scales in any theory of the brain. Slow time scales are present in many theories of brain function, but typically have been limited to the domain of learning and adaptation, thus functionally separated from fast processes. Here, the functional integration and co-evolution of the fast neuroelectric and slow permittive time scales suggests emergent and inherent properties of brain processes.

## Acknowledgements

We thank Masao Matsuhashi for efforts in data acquisition.

## Additional information

### Competing interests

Tomohiko Murai, Akio Ikeda: Department of Epilepsy, Movement Disorders and Physiology is the Industry-Academia Collaboration Courses, supported by a grant from Eisai Corporation, Nihon Kohden Corporation, Otsuka Pharmaceutical Co., and UCB Japan Co. The other authors declare that no competing interests exist.

### Funding

| Funder | Grant reference number | Author |
|---|---|---|
| European Union Seventh Framework Programme | 211713 | Matthias Dümpelmann Andreas Schulze-Bonhage |
| DFG/Cluster of Excellence BrainLinks-BrainTools | 1086 | Matthias Dümpelmann Andreas Schulze-Bonhage |
| Ministry of Education, Culture, Sports, Science and Technology | 15H05871 | Tomohiko Murai Akio Ikeda |
| Ministry of Education, Culture, Sports, Science and Technology | 19H03574 | Tomohiko Murai Akio Ikeda |
| National Health and Medical Research Council | GNT1183119 | Mark Cook Levin Kuhlmann |
| Ligue Francaise contre l'Epilepsie | | Maria Luisa Saggio |
| Horizon 2020 Framework Programme | 785907 | Viktor Jirsa |
| Horizon 2020 Framework Programme | 945539 | Viktor Jirsa |
| Horizon 2020 | m-Gate project Marie Skłodowska-Curie grant agreement no. 765549 | Mitsuyoshi Nakatani |
| Fondation pour la Recherche Médicale | DIC 20161236442 | Christophe Bernard Viktor Jirsa |
| SATT Sud Est | The Virtual Brain | Christophe Bernard Viktor Jirsa |
| Agence Nationale de la Recherche | EPINOV | Viktor Jirsa |
| National Institutes of Health | K01-ES026839 | Stephen V Gliske |

| National Institutes of Health | K08-NS069783 | Stephen V Gliske<br>William C Stacey |
|---|---|---|
| National Institutes of Health | R01-NS094399 | Dakota Crisp<br>Jared M Scott<br>Stephen V Gliske<br>Jack Lin<br>William C Stacey |
| Michigan Medicine | Robbins Family Research Fund | Dakota Crisp<br>William C Stacey |
| Michigan Medicine | Lucas Family Research Fund | Dakota Crisp<br>William C Stacey |

The funders had no role in study design, data collection and interpretation, or the decision to submit the work for publication.

## Author contributions

Maria Luisa Saggio, Conceptualization, Resources, Data curation, Software, Formal analysis, Validation, Investigation, Visualization, Methodology, Writing - original draft, Writing - review and editing; Dakota Crisp, Conceptualization, Resources, Data curation, Software, Validation, Investigation, Visualization, Methodology, Writing - original draft, Writing - review and editing; Jared M Scott, Formal analysis, Visualization, Methodology; Philippa Karoly, Levin Kuhlmann, Tomohiko Murai, Matthias Dümpelmann, Mark Cook, Data curation; Mitsuyoshi Nakatani, Formal analysis, Validation, Investigation, Visualization, Methodology; Andreas Schulze-Bonhage, Data curation, Funding acquisition, Writing- review and editing; Akio Ikeda, Data curation, Funding acquisition; Stephen V Gliske, Formal analysis; Jack Lin, Formal analysis, Validation; Christophe Bernard, Conceptualization, Formal analysis, Supervision, Funding acquisition, Investigation, Methodology, Project administration, Writing - review and editing; Viktor Jirsa, Conceptualization, Data curation, Software, Formal analysis, Supervision, Funding acquisition, Validation, Investigation, Methodology, Project administration, Writing - review and editing; William C Stacey, Conceptualization, Resources, Data curation, Software, Formal analysis, Supervision, Funding acquisition, Validation, Visualization, Methodology, Writing - original draft, Project administration, Writing - review and editing

## Author ORCIDs

Levin Kuhlmann  https://orcid.org/0000-0002-5108-6348
Andreas Schulze-Bonhage  https://orcid.org/0000-0003-2382-0506
Mark Cook  https://orcid.org/0000-0002-8875-4135
Stephen V Gliske  https://orcid.org/0000-0002-2259-2612
Christophe Bernard  https://orcid.org/0000-0003-3014-1966
Viktor Jirsa  https://orcid.org/0000-0002-8251-8860
William C Stacey  https://orcid.org/0000-0002-8359-8057

## Ethics

Human subjects: Data from the seven epilepsy centers were all collected with each institution's clinical EEG equipment and deidentified locally prior to data sharing. All data were acquired during the course of standard clinical care as part of the evaluation for potential epilepsy surgery, and no additional physical risks were present. All patients consented to have their data saved, deidentified, and shared, as approved by each local institution's review board policy.

## Decision letter and Author response

Decision letter https://doi.org/10.7554/eLife.55632.sa1
Author response https://doi.org/10.7554/eLife.55632.sa2

## Additional files

### Supplementary files

• Transparent reporting form

### Data availability

Code and data for validation analysis in "Accurate Identification of bifurcations" are freely available for download (https://doi.org/10.7302/ejhy-5h41).

The following dataset was generated:

| Author(s) | Year | Dataset title | Dataset URL | Database and Identifier |
|---|---|---|---|---|
| Crisp D, Saggio M, Scott JM, Stacey WC, Nakatani M, Gliske SV, Lin J | 2019 | A taxonomy of seizure dynamotypes - Code & Data | https://doi.org/10.7302/ejhy-5h41 | University of Michigan - Deep Blue, 10.7302/ejhy-5h41 |

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

## Appendix 1

# I. Methods

## I.1 Data collection

### Human EEG data

Data were collected from seven international epilepsy centers: University of Michigan, Mayo Clinic, Hospital of the University of Pennsylvania, Children's Hospital of Philadelphia, University of Melbourne, University of Freiburg, and Kyoto University Hospital (*Ihle et al., 2012*; *Cook et al., 2013*; *Kanazawa et al., 2015*; *Wagenaar et al., 2015*). All EEG data used in this work were from either grid or depth intracranial electrodes. All sampling rates were >200 Hz and antialiasing filters > 100 Hz, but there was variability between centers (XLTek, Nicolet, Natus, Nihon Kohden, NeuroVista). The Melbourne patients had ambulatory devices that recorded data for several months [3], while all the others were acquired during acute inpatient recording sessions. As the analysis was limited to amplitude and interspike intervals that should not be affected by the different techniques used, we did not stratify by center, but we did verify that the results did not depend upon center for offset dynamics. For onset dynamics, there were only two centers (Kyoto, Michigan) that had amplifiers that recorded low enough frequency content (high pass filter 0.016 Hz) (*Kanazawa et al., 2015*) to allow analysis of direct current (DC, i.e. very low frequency) shifts, and so only those centers were included in the analyses that involved DC coupling.

In every patient, EEG from a single electrode was used for analysis. The electrode was chosen in each case by reading the official clinical report to identify the seizure onset zone and choosing the electrode with the highest amplitude within the seizure.

### Simulated seizures

Validation of the classification scheme required a gold standard dataset, which we generated by simulating 60 random iterations of each type of onset and offset bifurcation. The simulation used the model and methods to begin/end at specific bifurcations described in *Saggio et al., 2017* and a function to generate a trajectory across each bifurcation. The trajectory was an arc of great circle as described in section II for the model, but the crossing point on the bifurcation curve and the inclination of the path with regards to the bifurcation curve changed randomly at each iteration to produce adequate variability within each bifurcation type. These conditions were implemented to create a robust data set that would mimic biologic variability. We simulated an equal number of seizures per bifurcation. The final result was 240 seizure onset and 240 seizure offsets, each with random characteristics and spanning a wide range of physiologically-relevant parameters. The 480 seizures were used to assess the accuracy of both the human and algorithm classification schemes.

For each bifurcation, we created an algorithm able to choose a random point on the bifurcation curve (see section V) and a random point on a curve we designed to run parallel to the bifurcation curve. A trajectory linking the two points would cross the bifurcation curve with different inclinations at each iteration. The combination of a random point on the bifurcation curve and different inclination gives different amplitude/frequency behaviors, as shown in *Appendix 1—figures 18* and *19*. Given that variability in the non-prescribed trends is due to the presence of other bifurcations nearby, we ignored portions of the bifurcation curves far from others, such as the portion of SupH behind the sphere as compared to the bistability regions. Two of the onsets (SubH and SupH) required a small amount of noise to be added to the simulations in order for the system to be able to leave the unstable equilibrium. The velocity of the slow variable, *c*, was choosen for each type to be small enough to ensure time scale separation so that the prescribed scaling law would be visible.

As these simulations had arbitrary timescale and different levels of background noise, we then post-processed the data to blind the reviewers. We first added normal random noise (MATLAB's 'normrnd', mu = 0, sigma = 0.01) to all samples of every signal based upon the

existing noise. We then rescaled all seizures so that the final interspike intervals would be ~0.2 s, then downsampled ('decimate' function in Matlab) so the sampling rate would be ≤ 200 Hz. All codes are available for download at https://doi.org/10.7302/ejhy-5h41.

## I.2 Spike analysis

As stated in the main body, we define a 'spike' in dynamical terms: any prominent sharp transient associated with the dynamical process, which for these human EEGs means any fast transient <250 ms long and amplitude discernable from the background. Inter-spike interval (ISI) is defined as the time between sequential spikes. Amplitude for a given spike is defined as the absolute maximum peak-to-peak difference in a window of time ranging from the halfway point of the first spike and the halfway point of the latter spike (see *Figure 2A* in main body). Beginning/ending spikes utilize only the halfway point of the nearest spike. All analyzed data were decimated to ~200 Hz for efficient analysis. In order to remove slow transients and identify the local spike amplitudes, raw EEG data were first high-pass filtered (MATLAB's 'highpass', 1 Hz). Peaks were found in Matlab using the 'findpeaks' function, after manually optimizing amplitude, time, and prominence for each patient. This process required iterating the features until visually confirming that spikes were correctly detected, using plots in Matlab with the spike detections superimposed upon the EEG signal (see *Appendix 1—figures 1– 9*). Due to the wide variability between patients, we were unable to develop a reliable automated method to determine the spike locations in every patient.

Sentinel spikes. Some seizures began with a single high amplitude 'sentinel spike,' which typically was present in many electrodes, often far more than were involved in the subsequent seizure. We used this spike to indicate the initial seizure onset time based upon the decision of our clinical experts. However, these spikes were not included in the analysis of dynamotype because 1) the sentinel spikes often occur on channels that are not involved in the later seizure, which calls into question their suitability as a canonical feature of the dynamotype; 2) their presence has no clear relationship with the subsequent dynamotype; 3) according to the basic dynamics principles, they are not a canonical feature of any of the onset bifurcations; and 4) in typical human recordings it is not possible to discern when these are true spike-wave discharges versus filtered transients of a DC shift (e.g. the first spikes seen in *Figure 2B and C*, *Appendix 1—figures 1* and *3*). Thus, in this work, we note their presence to determine seizure onset time, but at present do not include them within a framework of a canonical model of seizure dynamics.

Spike-wave complexes. Some seizures had spike-wave complexes in which the aftergoing slow wave was very prominent, similar to the waveforms seen in absence epilepsy. In these cases, the highest amplitude of the whole complex was used, even if it was the slow wave, and only a single event was counted from each complex (see *Appendix 1—Figure 3*).

Clonic spiking. As described previously (*Jirsa et al., 2014*), when there was clonic spiking at the end seizures, the interclonic interval was used to evaluate the presence of slowing down, and the ISI between successive fast spikes within each clonic burst were ignored (see *Figure 2B* and *Appendix 1—figures 4, 7*).

It is important to note that this analysis does not address every feature of the seizures. There are many phenomena, such as the three listed above, the shape and frequency of the spikes, and other complex patterns, that are not invariant features of the bifurcations. While these are important to the clinical description and can be relevant to understand the underlying dynamics, in the framework here proposed they do not contribute to defining the dynamics of seizure onset and/or offset, and are thus uninvolved in dynamotypes.

## I.3 Visual classification of seizure dynamics

The key to differentiating the bifurcation type is to identify the invariant dynamical features, which can be summarized as the presence of a DC shift and the behavior of the ISI and amplitude (see *Figure 1*). These features are typically quite easy to distinguish. The only prominent ambiguity is that it is not feasible under clinical conditions to distinguish between the logarithmic and square root functions at offset, as previously described (*Jirsa et al., 2014*).

Thus, our first test was to determine if human reviewers can classify the different bifurcations visually using simple rules.

The presence of a DC shift and the general trends of ISI and amplitude can readily be determined upon visual analysis. The basis for this analysis is to determine whether the amplitude and ISI scale to zero. For ISI, this appears as a decreasing frequency as T approaches 0, that is slowing down at seizure offset, or speeding up after seizure onset. For amplitude, it appears as a gradual change in the spike amplitudes, with the spike at T = 0 being very small compared to the baseline, then increasing further away from T = 0. This description is qualitative but is readily applied to typical EEG data. Since the onset/offset dynamics are typically defined by only 5–10 spikes and there is considerable noise and variability in real EEG signals, rigorous curve fitting is rarely possible (though we included it in the examples in *Appendix 1—figures 1–9*). Just as with clinical EEG reading, we found that a much simpler and more reliable classification system was to visualize the ISI and amplitude plots of the first and last 10 spikes and determine whether the trends were scaling to zero, constant, or arbitrary, and if there was a DC shift. For amplitude, we defined 'scaling to zero' as steadily diminishing to less than three times the background level near T = 0. For ISI, we define it as steadily larger ISI near T = 0, with the last two ISI >50% larger than the mean ISI 10 s prior. All analysis for onset and offset concentrated on the first/last 5 s of data, but occasionally used up to 15 s to observe the full patterns. Using these definitions, we developed a visual classification system (*Appendix 1—table 1*).

**Appendix 1—table 1.** Visual classification system. Classification relies upon visualization of the given features. In case of multiple features present, the bifurcation listed on top takes priority. DC shift: a sharp deflection >5 times the background that occurs in <0.5 s, then lasts >1 s. Constant: the value is consistent and does not trend upward or downward for 10 spikes. Arbitrary: no consistent unimodal trend over 10 spikes.

| **Onset** | |
| --- | --- |
| DC Shift | SN(+DC) |
| Amplitude increasing | SupH |
| Interspike interval (ISI) decreasing (i.e. frequency increasing) | SNIC |
| ISI and Amplitude constant/arbitrary (no DC shift) | SN(-DC) or SubH |
| **Offset** | |
| DC Shift | SH(+DC) |
| Amplitude decreasing | SupH |
| ISI increasing (frequency decreasing) (no DC shift) | SH(-DC) or SNIC |
| ISI and amplitude constant/arbitrary (no DC shift) | FLC |

We also developed an analytical tool to use quantitative features and machine learning to identify the dynamotype. The goal of this analysis was to determine if the features used in the qualitative study were robust. We designed features based upon the visual classification system in *Appendix 1—table 1*, focused on quantifying DC shift, amplitude trends, and interspike interval trends. Definitions, feature computation, and feature descriptions are as follows:

Baseline Definition: Several features require definition of the baseline, that is seizure activity vs. non-seizure activity. All analyzed waveforms include a period of baseline, followed by seizure activity, followed by more baseline. Computationally, the term 'baseline' below is defined here as the segment of the waveform that started before/after a seizure. For an onset baseline, this segment was taken as the start of the waveform up until seizure onset. For an offset baseline, this segment was taken as the point after seizure offset to the end of the waveform.

DC vs. Non-DC: Data acquired at Kyoto (Nihon Kohden EEG 1100 amplifier) or Michigan (Natus Quantum amplifier), both of which record down to 0.016 Hz, were included in the DC cohort, and all other data was considered 'non-DC'. Non-DC data were first filtered with 1 Hz highpass filter (Matlab 'highpass'), then all features extracted. On the DC data, features were computed on the raw, unprocessed EEG.

Preprocessing steps: 1) Onset/offset times, as well as the relative bifurcation window lengths, were determined by a trained epilepsy specialist. 2) Spikes were identified using findpeaks.m (Matlab) to locate upper (maxima) and lower (minima) spikes. 3) Seizure polarity was determined by determining whether the median of the upper or lower spikes had a greater absolute difference from the baseline median. The value with larger difference was chosen as the true 'spikes' and were used for all further analysis. All amplitudes were taken as absolute values. This step accounts for the fact that spikes can be either positive or negative in intracranial electrodes.

Feature 1 – ISI trend: The ISI of all spikes were computed and plotted consecutively. A simple line was fit to the data with a least square algorithm and the slope of the line fit was extracted as the overall ISI trend. For onset (offset), the first (last) 5–15 spikes were used, using as many as possible until a clear inflection point in the line. The order was reversed for offset.

Feature 2 – Amplitude trend: This feature was computed exactly as in ISI trend, except the peak-to-peak spike amplitudes were used in place of the ISI.

Features 3 and 4 – normalized upper and lower peak median: The signed distance between the median of the upper peaks and the baseline median was computed (upm). That same was done for the lower peaks (lpm). If $|upm| > |lpm|$, then normalized values were $nupm = 1$, $nlpm = upm/lpm$. If $|lpm| > |upm|$, then $nlpm = -1$, and $nupm = -lpm/upm$ (note that lpm is negative in all cases). These features identified DC shifts.

## I.5 Validation of classification methods

In our prior work with the Epileptor model (the SN/SH type), we proved the goodness-of-fit to the logarithmic equation for terminal ISI (*Jirsa et al., 2014*). However, in the current work we found that GoF methods were not robust when discerning between multiple seizure types in noisy data. This is because the number of samples (spikes) is small, while the combined uncertainty in the noise as well as the classification of the different types can be significant. The result was that the variability due to noise was often more prominent than the differences between bifurcation types. This difficulty is not surprising, as seizures are notoriously difficult for automated algorithms to identify under clinical (noisy) conditions, and classification will be even more difficult. More importantly, even under ideal conditions any such analysis would be limited due to the lack of a gold standard. We therefore decided on an alternate and more rigorous approach, appropriate for the data quality of clinical seizures.

In order to validate our classification system, we used a multi-step approach. The key to this validation was the generation of a gold-standard with simulated data, in which we know which bifurcation is present. Using the gold standard, we first used machine learning tools to assess how accurately our chosen features are capable of identifying the differences between the bifurcations. We then tested how accurately human reviewers could identify each bifurcation using visual review. After proving that the human reviewers were accurate and reliable in the gold standard, we tested their reliability in the true clinical dataset. These steps are detailed below.

## Step 1: Feature validation and bootstrapping:

OBJECTIVE: Determine whether the chosen features capable of discerning between the four bifurcations in the gold standard.

METHODS: Using the simulated seizures (240 onsets, 240 offsets), supervised learning was chosen to quantify how well the simple features in I.4 captured the differences between the different dynamotypes. This was performed by fitting labeled features to a statistical model to generate a goodness-of-fit measurement. Specifically, we used a multinomial logistic regression model ('mnrfit', Matlab), which outputs a model parameter 'deviance' that estimates the Goodness-of-Fit. The GoF was compared with the gold standard in the simulation data. Note that this method is not used for classification nor to determine the 'best' potential fits. Rather, we used the GoF statistic to assess whether our chosen features were capable of discerning between the four bifurcations. While there certainly could be

better features, we chose these because they recapitulate what experts use for visual analysis, which is what this test is trying to validate.

To provide reference for the GoF computation, we performed a permutation test by randomly scrambling the bifurcation labels and re-fitted the data to the model. This was repeated 10,000 times to provide the distribution of random GoF, which can then be compared with the index case and provide a p-value, a process known as bootstrapping.

RESULTS: For both onset and offset bifurcations, the GoF with true labels was better than all 10,000 permutations (p<1e-4). This result indicates that the four features are very effective in capturing the differences between the four onset and offset bifurcations and extremely unlikely to be due to chance.

CONCLUSION: The chosen features are capable of discerning between the bifurcations.

## Step 2: Human visual analysis of simulated data:

OBJECTIVE: Validate whether visual analysis by human reviewers accurately identifies the different bifurcations in the gold standard.

METHODS: Using the simulated seizures (240 onsets, 240 offsets), three independent reviewers labeled all bifurcations using an in-house Matlab viewing program that kept labels blinded while also randomizing the order in which the seizures were viewed. DC-coupled data were shown with two waveforms: raw data and 1 Hz highpass filtered data ('highpass', Matlab), in order to highlight DC components while also allowing viewing of spike dynamics. Reviewers had to choose exactly one of the four potential bifurcations for each example.

RESULTS: For onsets, all three reviewers unanimously agreed in 80%, 2/3 agreed in 18.3%, and no agreement was found in 4/240 (1.7%). Compared to ground truth, reviewer 1, 2, and 3 had 100%, 97.92%, and 80.42% accuracy, respectively. For offsets, there was majority agreement in all 240 seizures, 78.75% of which were unanimous. The Fleiss Kappa score for these results has a p-value that is lower than the smallest number possible to express in Matlab (p<4.94e-324). Compared to ground truth, reviewer 1 and 2 both had accuracies of 99.85% while reviewer 3 had 78.75%.

CONCLUSION: These results clearly show that the visual classification system can distinguish the four bifurcation types, and that human reviewers are accurate and consistent.

## Step 3: Validation on clinical data

OBJECTIVE: The strong results of Steps 1 and 2 validate the use of these methods in clinical data. The goal of the present step is to determine reliability of the three reviewers when labeling the human seizures, and to use these labels for the taxonomy in *Figure 2*.

METHODS: We evaluated the 51 DC-coupled seizures and 69 non-DC-coupled seizures separately (see main paper). We used similar methods as in Step two to classify the human seizures. One important difference in this analysis is that SubH onsets were grouped with SN (-DC), and SNIC offsets with SH (-DC), as described. We calculated the Fleiss Kappa statistic between the three reviewers to assess inter-rater agreement.

RESULTS: As seen in *Appendix 1—table 3*, the three reviewers had high agreement, especially in the DC-coupled data. We also used the majority labels with the bootstrapped automated feature analysis as in step 1, to determine how well the given features captured the specific chosen bifurcation. These results showed that the DC onset and offset both had excellent GoF with the chosen features. Non-DC offset was also excellent, but the non-DC onset had less consistent results (p=0.0969). This demonstrates the usefulness of DC-coupled recordings to identify the onset bifurcations.

**Appendix 1—table 2.** Reviewer agreement on simulated data. Reviewers were highly consistent with each other in the simulated data. The interrater reliability is extremely high, as the p-value (Fleiss Kappa, three reviewers) is negligible.

|  | Unanimous | 2/3 agree | No consensus | P value |
|---|---|---|---|---|
| Onset (N = 240) | 192 | 44 | 4 | p<4.9 e-324 |
| Offset (N = 240) | 189 | 51 | 0 | p<4.9 e-324 |

**Appendix 1—table 3.** Accuracy identifying bifurcations in clinical data. A. Reviewers were highly consistent with each other scoring the bifurcations in the clinical data. (Fleiss Kappa, three reviewers). Each patient was stratified based upon whether their EEG had DC-coupled recordings, as the non-DC group could not disambiguate SN or SH with DC shifts. B. The model fit of the automated features to each chosen bifurcation was performed versus human majority vote, then bootstrapped as in Step 1. The chosen model parameters were clearly highly descriptive of chosen bifurcations in DC onset and offset and non-DC offset. As expected, the lack of DC coupling makes it difficult to disambiguate the four onset bifurcations.

**A. Reviewer agreement on human data**

|  | Dc (N = 51) | Non-DC (N = 69) |
| --- | --- | --- |
| Onset | p=1.78e-15 | p=4.51e-12 |
| Offset | p=4.51e-11 | p=9.72e-4 |

**B. Automated features permutation test**

|  | DC | Non-DC |
| --- | --- | --- |
| Onset | p<1e-4 | p=0.0969 |
| Offset | p=7e-4 | p<1e-4 |

CONCLUSION: These results show that the labels are highly consistent between different reviewers. Given the results from Steps 1 and 2, we conclude that is it highly likely that the bifurcations are present in the data and that these methods are correctly identifying them. Given these positive results, we used the majority-vote classification for the taxonomy seen in *Figure 2*.

## I.6 Challenges with classification

There were several conditions in the data that required rules for disambiguation between the different bifurcations:

1. Amplitude scaling. Amplitude was often arbitrary in our clinical data, showing considerable variability from spike to spike. This variability sometimes made it difficult to determine if there was scaling to zero (square-root). In those cases, we observed the spikes closest to T = 0: if the first 5–10 spikes appeared to arise from the baseline noise and consistently grow in later spikes, it was classified as square-root scaling (SupH). If the first 5–10 spikes were clearly >3 times higher than baseline noise, the seizure was classified as constant (i.e. arbitrary) amplitude even if later spikes increased in amplitude.
2. Late amplitude increase. In many seizures, after ~10 s it was common for the spiking activity to increase in amplitude as the seizure spread to neighboring electrodes. This late increase was not part of the onset bifurcation, and only the initial trend during the first 10 s was used for classification, which was typically not SupH.
3. 'Arbitrary' scaling usually constant. Although the 'non-zero' scaling listed in *Figure 1* theoretically includes many arbitrary patterns, we found that the vast majority of examples manifested as a constant value, similar to the analysis in *Jirsa et al., 2014*. For instance, nearly all SN onsets had constant amplitude and ISI, and SH offsets had constant amplitude.
4. Mixed dynamics. In cases where features from more than one bifurcation were present in the data (e.g. DC shift and amplitude scaling), we used the following prioritization order: DC shift → amplitude scaling → frequency scaling. The highest priority was used as the final bifurcation, for example if DC shift and amplitude scaling were both present, it was classified as SN (onset) or SH (offset). *Appendix 1—table 1* presents the bifurcations in this order.
5. Unusual dynamics. There were rare examples in which the dynamical behavior was highly unusual, all of which we present in this Appendix (*Appendix 1—figures 6*, *10*, *17–19*). Each of these patterns was classified as 'arbitrary'. See section I.8 for further details.

Despite these rules, there were examples in which the human reviewers were either incorrect in the simulated data or did not agree with each other in the clinical data. While

these errors were uncommon, we evaluated each case to determine the conditions responsible. The following are a list of the conditions that can make classification difficult. 'a' and 'b' were specific to the simulated data, as it is necessary to know the ground truth to identify these problems.

1. Misclassifying arbitrary bifurcations (in simulated data). This occurred when an arbitrary bifurcation had parameters that produced trends very similar to scaling laws (either amplitude or ISI). This is expected, as the full range of 'arbitrary' can encompass some of the other parameters.
2. Distinguishing SN from SupH onsets (in simulated data). This was due to a technical aspect of the simulation, which is a manifestation of the proximity of the SN and SupH bifurcations as described in the first paragraph of 'Ultraslow fluctuations to navigate the seizure map' in the main body. In order to produce the full range of SupH parameters, some of the simulations passed through the SN bifurcation first, producing a DC shift, immediately followed by amplitude scaling. By our definition, this bifurcation is defined as a SN; however, some of these examples were produced by parameters designed to generate SupH bifurcations. This is an issue of labeling of the simulation parameters but is not a failing of model or TSD; as seen in *Figure 5* and explained in the text, TSD predicts this very effect, and we saw several examples of it in our clinical data (SN and SupH coexisting in the same human epileptic network).

The following phenomena are not described by the canonical features of the bifurcations and were difficult for the reviewers to classify. The majority of disagreements between reviewers contained at least one of the following:

1. Polyspikes or spike waves at seizure onset, especially prior to low voltage fast activity
2. Preictal spikes
3. Noisy data, especially 'extraneous' spikes that disrupt scaling laws and 'Unusual dynamics' (#5 in preceding section)
4. Ambiguous seizure onset/offset times
5. Low voltage fast activity that speeds up at seizure onset

These patterns (a-g) are inherent challenges of seizure data, and such conditions have the potential for ambiguous classification. However, we do not consider this to be a failure of our method—rather, our framework provides a method of identifying and quantifying these conditions. While some of these phenomena are common, they are not invariant features that define the bifurcation. In effect, the fact that they deviate from the basics of TSD not only allows us to identify them as atypical, but also to quantify *how and why* they are atypical.

## I.7 Examples of classifying clinical bifurcations

Due to the large degree of heterogeneity and noise in clinical recordings, it is common to have patterns that do not fit perfectly with the theoretical dynamotype. The following figures demonstrate how the method described above can be used to classify real-world scenarios reliably. Each of the examples below had a consensus among the three reviewers except as noted.

The key to this approach is to follow the prioritization described in *Appendix 1—table 1*, I.3 and I.6. If there is a DC shift, the dynamotype is SN onset (or SH offset), regardless of other patterns. If not, amplitude scaling determines if it is SupH, followed by frequency scaling for SNIC, and finally other patterns are SubH/SN onset (FLC offset). Following this progression as indicated, we validated this with three independent reviewers. Note that *Figure 2* has several clear examples of the classification, only one of which is repeated in this section. The consensus values in *Figure 2* and *Appendix 1—tables 2* and *3* are validation that this method is reliable. The data are presented in the same priority as *Appendix 1— table 1*. Each figure first shows the raw data, often with a filtered version directly below, as well as the spike detections that are used to determine ISI and amplitude. Below the raw data are the ISI and amplitude plots, which are used to assess for trends. However, these quantitative tools and equation fits are not necessary for the classification scheme, which we

have proven to be accurate with human review rather than choosing the 'best fit' from an algorithm (Appendix I.5). We include these graphical fits and RMSE for constant and square root/logarithmic equations to demonstrate how this tool may be used to determine parameters for these equations, which can be used for quantitative analysis, comparison, and robust discussion of seizure dynamics in future work.

Each example shown has some heterogeneity—perfect examples are quite rare. Instead, these examples are meant to show how to classify under realistic clinical conditions. It is critical to note that, following the steps in *Appendix 1—table 1*, each of these was reliably classified independently. We have included some of the 'challenges' listed in I.6 to show how they can be approached. In addition, where possible we also describe how the other half of the seizure would have been classified (e.g. how to classify the offsets in *Figure 1A,B*, which were chosen to show onsets). These are included for transparency, as it is important to be able to deal with the heterogeneity seen in clinical conditions. We do not claim this (or any other) classification scheme can describe all seizures; however, our classification provides the language and framework to identify which specific features of a seizure are dynamical anomalies. Thus, even when strict classification of dynamotype is ambiguous, our method provides the means to describe *how it is ambiguous*. However, it is important to note that, as a whole, the reviewers were able to reach a high level of agreement (see *Figure 2*, *Appendix 1—tables 2* and *3*)—most seizures can be classified reliably by following these rules.

 1: SN (+DC)
 2: SupH onset after uncertain start (better SupH onset seen in *Figure 2* and *Appendix 1—figure 8*)
 3: SNIC onset after uncertain start
 4: SNIC onsets from rat data
 5: SubH or SN (-DC)
 6: SubH or SN (-DC)
 7: SNIC or SH (-DC)
 8: SupH offset (also SupH onset)
 9: FLC offset (additional examples in *Appendix 1—figures 3*, *6*, *10*, *17*)

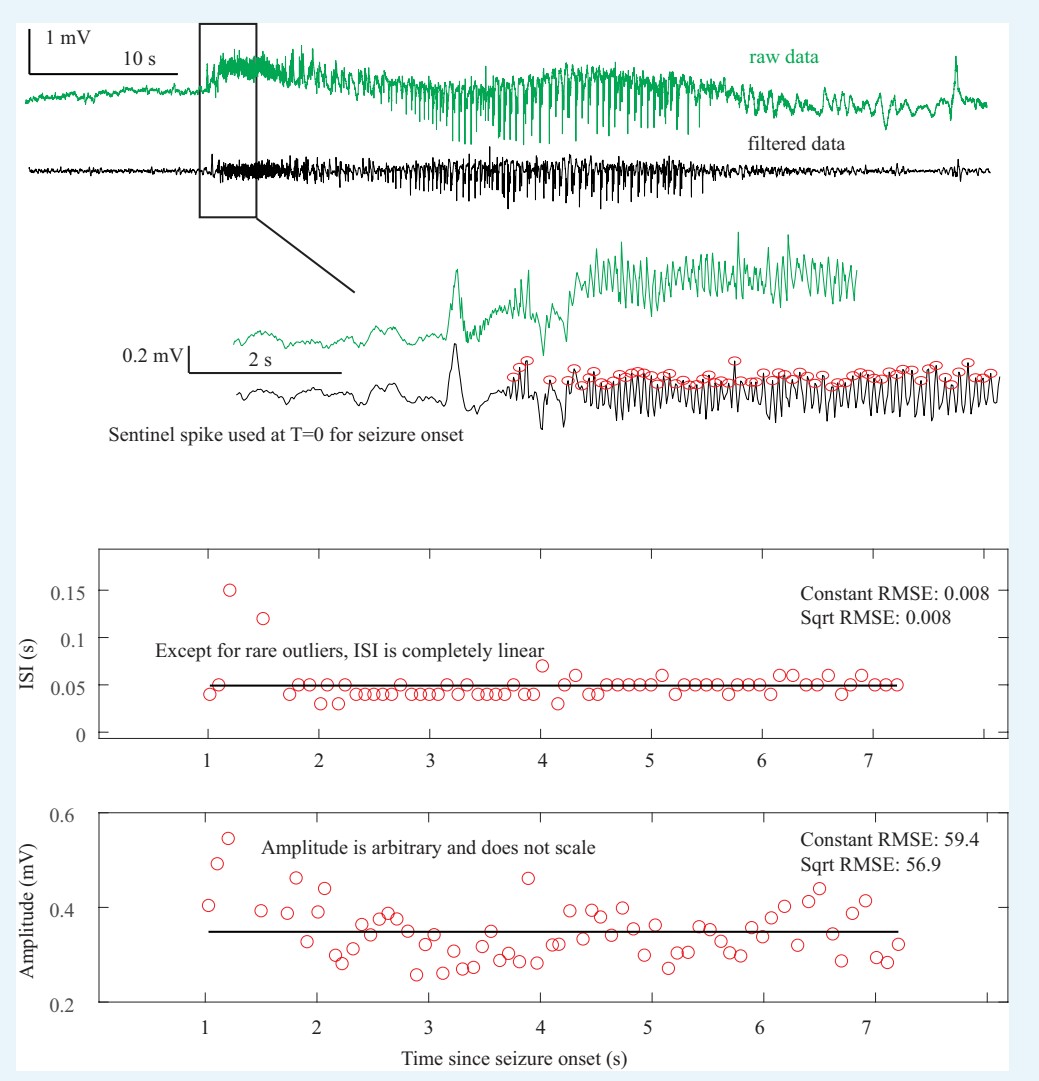

**Appendix 1—figure 1.** Saddle Node Onset DC-coupled recordings (green, top) show DC shift that occurs immediately upon seizure onset. To determine spike amplitudes and ISI, data must be high-pass filtered (black). Those filtered data are then used to identify the spikes (red circles) for analysis. The interspike intervals (ISI, middle) and amplitude (bottom) of each spike are then plotted versus time since seizure onset for visual analysis and curve fitting. The seizure begins with fast 20 Hz firing at arbitrary amplitude for over 5 s. This is followed by changing amplitude, irregular firing, and then clonic bursting, but those are clearly after the initial onset. The combination of arbitrary amplitude, ISI, and DC shift is consistent with Saddle Node. *Onset Classification*: DC shift: yes, therefore SN onset. All three reviewers agreed. *Offset Classification:57640* This is an example of a challenging offset. Analysis is uncertain because seizure offset time is unclear (is it when the fast spikes stop, when the slower oscillations stop, when the low voltage fast activity stops, or when the large final spike occurs?). Thus, the method is uncertain. DC shift: no. Amplitude decreasing: uncertain. ISI increasing: uncertain. This offset is ambiguous and reviewers were unable to reach consensus. One reviewer felt the high amplitude spiking fit with increasing ISI (SH or SNIC). Another felt it should be FLC because of the arbitrary patterns. Another was uncertain how to score given the ambiguity of seizure offset. It was listed as 'no consensus'.

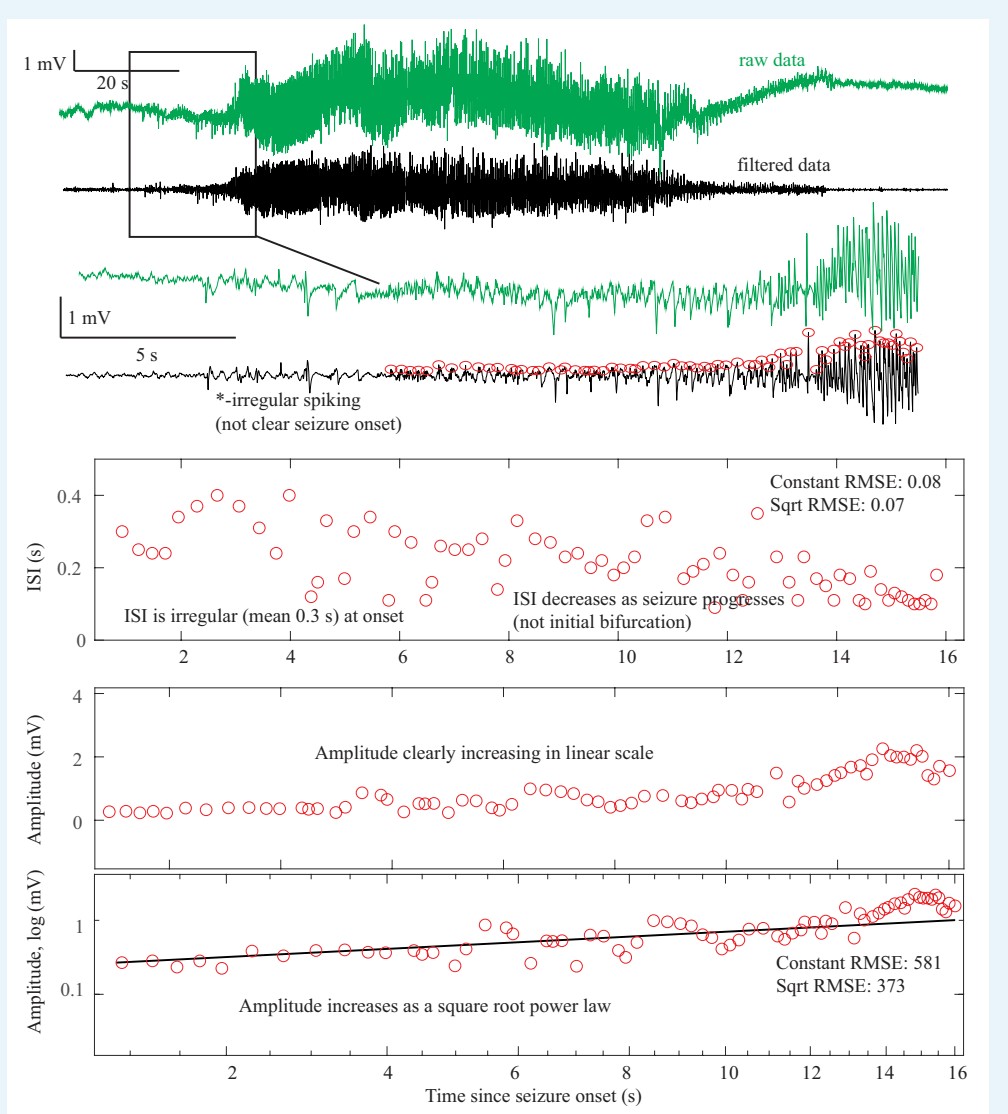

**Appendix 1—figure 2.** Supercritical Hopf Onset DC-coupled recordings (green, top) do not have a DC shift at onset. The ISI is arbitrary at onset, then after 10 s starts to decrease (middle). This irregular spiking prior to the seizure (*) is not part of a clear seizure onset, does not conform to any initial bifurcation, and thus was not chosen as the unequivocal seizure onset. The sustained spiking that begins at 8 s, however, follows the SupH dynamics very well. Amplitude shows steady increase in linear scale, which appears as a straight light in loglog plots, consistent with a square root power law (loglog plot). This seizure is most consistent with SupH due to the amplitude scaling. *Onset Classification*: This onset is somewhat challenging because there was some disagreement about when the seizure started. A more straightforward SupH onset is seen in *Figure 2* and *Appendix 1—figure 8*. In this case, two reviewers felt the irregular spiking was not a sustained start of the seizure and potentially was just preictal. One reviewer thought it could be the start and the seizure might be an arbitrary pattern (SubH or SN (-DC)). After discussion, reviewers agreed that the primary sustained pattern was increasing amplitude. Final analysis—DC shift: no. Amplitude increasing: yes, therefore SupH onset, but with note made of irregular spiking at seizure onset of uncertain significance. *Offset Classification*: This offset is difficult because there is decreasing amplitude like the onset, but then a persistent low voltage fast activity. DC shift: no. Amplitude decreasing: There was disagreement about whether the decrease comprised the final dynamotype of the seizure terminus, or if the low voltage fast activity that persisted at the end was a separate pattern. ISI increasing: no. Thus there was disagreement about

whether this should be a SupH or FLC: there is a decreasing amplitude similar to the SupH onset pattern, followed by 15 s of constant ISI low voltage spiking.

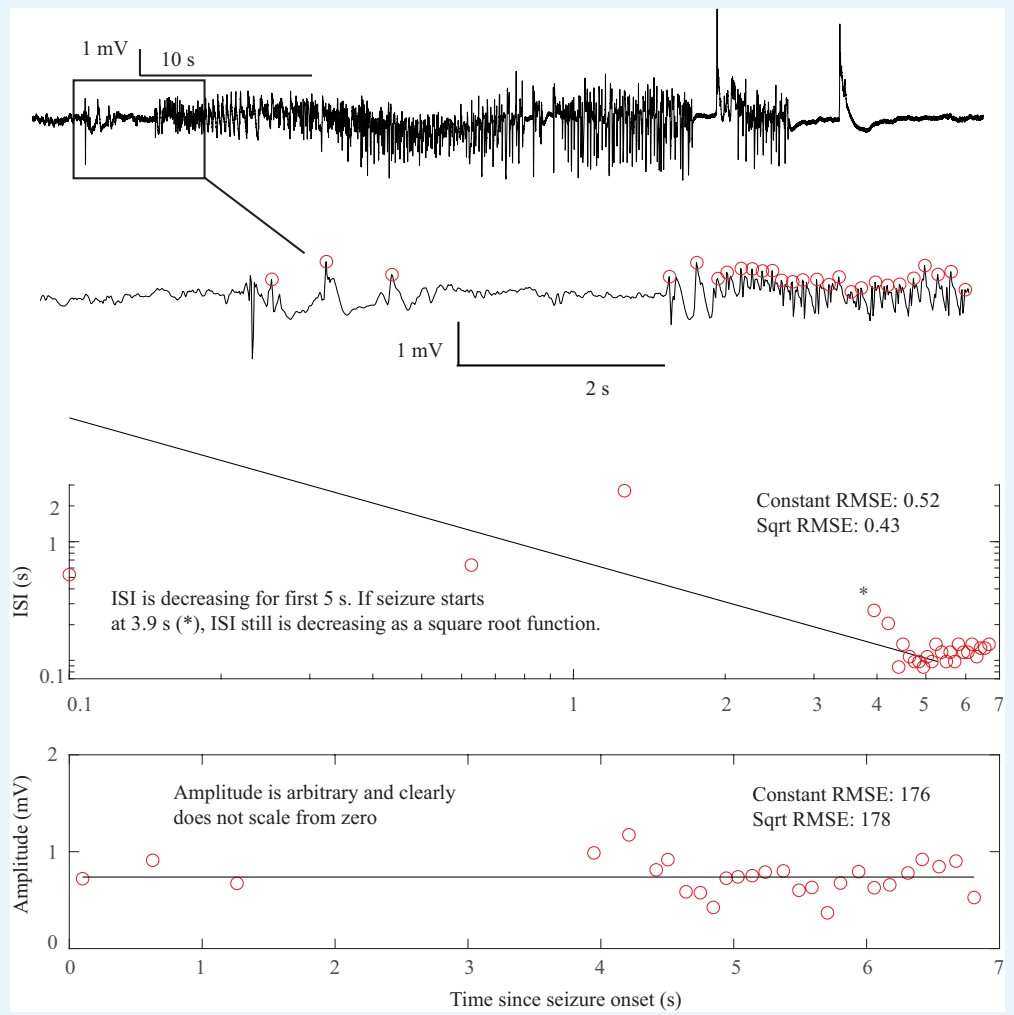

**Appendix 1—figure 3.** SNIC Onset. The seizure begins at some point after a sentinel spike. The true starting time of this seizure is debatable, as there are three spike-waves with large ISI and amplitude but they do not persist. At 3.9 s after the sentinel spike (*) there is a fourth spike wave that leads into the unequivocal seizure onset, with accelerating frequency until 5 s. Whether one chooses T = 0 or T = 3.9 as the starting time can change the determination. At 3.9 s, the pattern starts with high amplitude spike waves that accelerate in frequency, characteristic of the SNIC. At 0 s, there was uncertainty whether the initial spike waves should be treated like a SNIC with a pause, or an arbitrary pattern. Spike wave discharges were present in both patients that we classified as SNIC, but are not a requirement. In this case, visual inspection was more reliable than fitting equations to make the determination because seizure onset was irregular. *Onset Classification*: DC shift: no. Amplitude scaling: no. ISI decreasing: yes, therefore a SNIC. However, there was concern that by starting at T = 0 this could be arbitrary rather than decreasing. After discussion it was felt this was most consistent with SNIC, as it appears to have accelerating ISI with a pause, rather than a truly arbitrary pattern. However, only 2/3 reviewers agreed. *Offset Classification*: DC shift: no. Amplitude or ISI scaling: no. This was classified as FLC. It has irregular gaps during the seizure that are not well described by the basic dynamotypes.

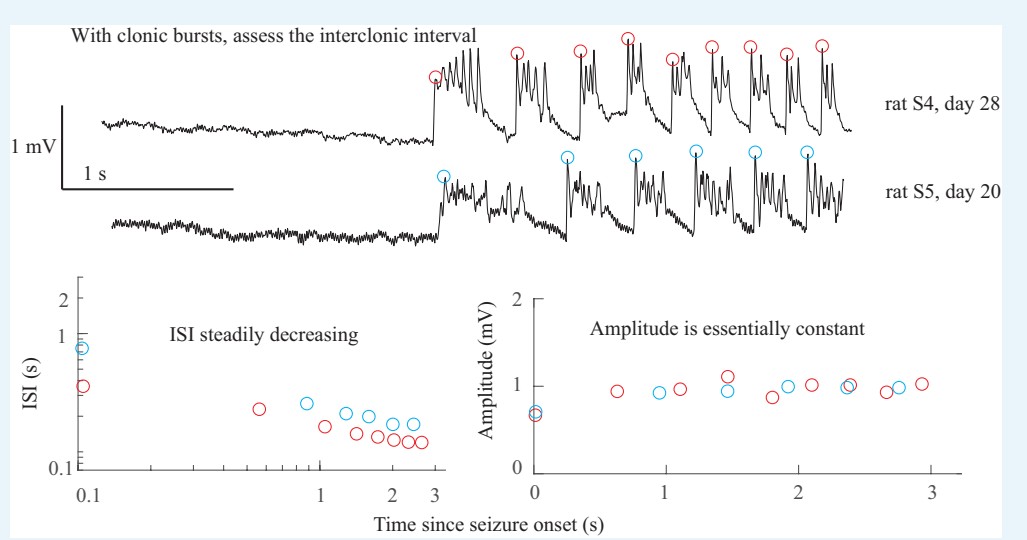

**Appendix 1—figure 4.** SNIC Onset in rats. There were no unequivocal examples of SNIC onset in our human data. However, our group recently published data from the tetanus toxin model in rats that has excellent SNIC onsets (*Crisp et al., 2020*). Shown are data from two rats, which were shown in *Figure 5* of that paper (see that work for details on experimental procedures). Both animals have clonic bursts, and in the presence of such discharges the dynamotype is characterized by the inter-clonic interval, rather than the fast runs of spikes within the burst (*Jirsa et al., 2014*; *Bauer et al., 2017*). In both cases the seizure begins with increasing ISI and essentially constant amplitude. *Onset Classification*: DC shift: no. Amplitude increasing: no. ISI decreasing: yes, therefore SNIC onset. Note, there is sometimes a continuum between some of the bifurcations; here, the clonic bursts have some characteristics of a DC shift, but since they all return to baseline before the next burst this is not considered a sustained DC shift. Additional details are discussed in *Crisp et al., 2020*. *Offset Classification*: Seizure offset is not seen in this view.

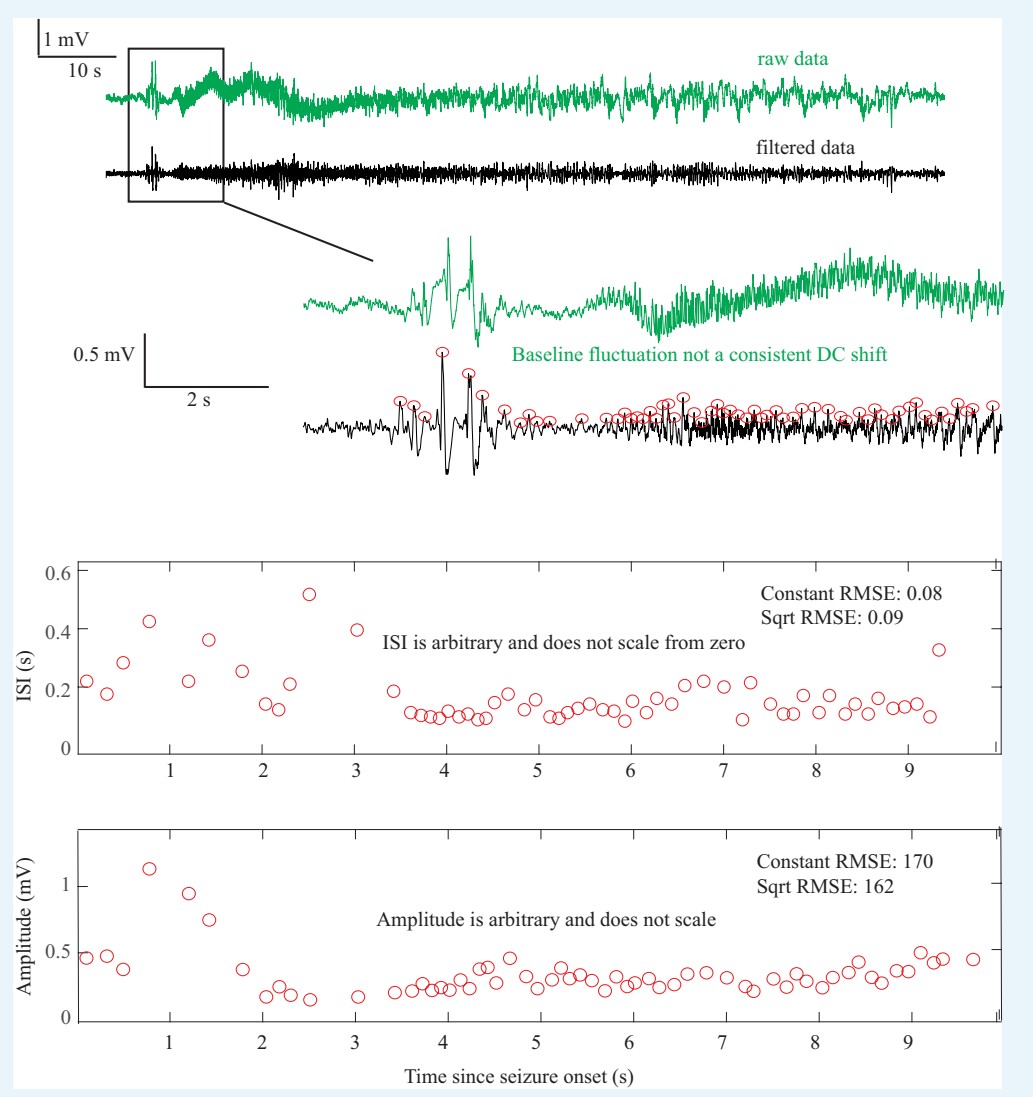

**Appendix 1—figure 5.** Saddle Node or Subcritical Hopf Onset DC-coupled recordings (green, top) show some baseline fluctuations that are not consistent and too slow to be considered a DC shift. This recording was capable of showing DC, but this patient's data did not have a clear DC onset. The ISI and amplitudes are both arbitrary and do not scale from zero. This seizure is consistent with either SN or SubH. *Onset Classification*: Seizure onset time is somewhat ambiguous, but in this case does not matter for the classification because there is no consistent change regardless of starting point. DC shift: no. Amplitude increasing: no (not consistent). ISI decreasing: no (not consistent). Therefore, this is an arbitrary pattern and is a SN (-DC) or SubH. All reviewers agreed. *Offset Classification*: Seizure offset is not seen in this view.

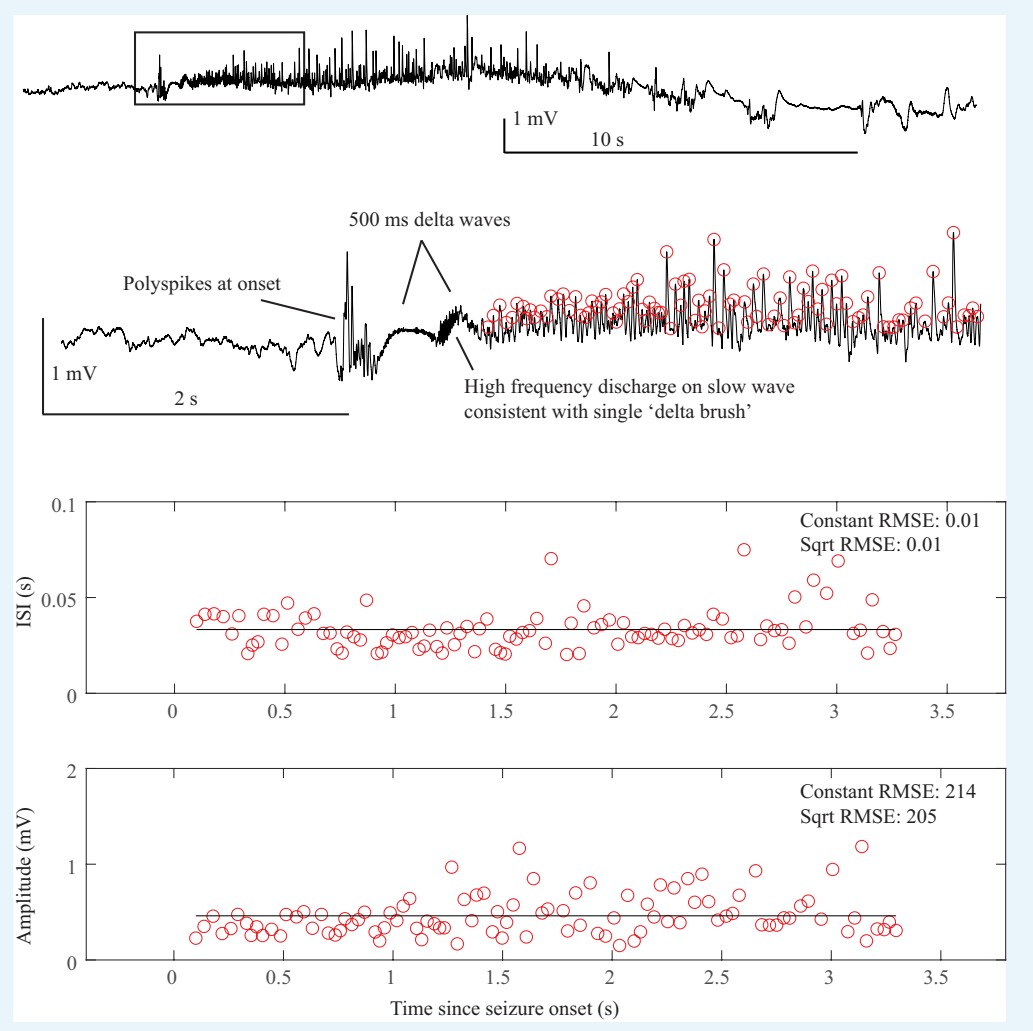

**Appendix 1—figure 6.** SN or SubH onset after delta brush. After a sentinel spike, there is a burst of fast spikes, followed by low-voltage fast spikes that lead into an unusual slow wave similar to the 'delta brush' pattern. The onset pattern is unusual: there are fast polyspikes with arbitrary amplitude after the sentinel spike, then the delta brush, and it is unclear whether there was a DC shift or a slow wave. These early patterns do not fit with any bifurcation. This seizure was labeled as SN or SubH due to the arbitrary amplitude and constant ISI, but is an outlier due to unusual pattern. The delta brush pattern was consistent with Perucca type (vii), see *Appendix 1—table 5*. *Onset Classification*: Onset has several unusual patterns in a row. DC shift: no (there appears to be two slow waves at onset, which both begin to recover rather than maintaining a DC shift). Amplitude increasing: no (spikes are quite variable). ISI decreasing: no. Thus, this is an arbitrary pattern and fits with SN (-DC) or SubH. All reviewers agreed. *Offset classification*: Offset time is somewhat ambiguous, but likely occurs 10 s before the end of the tracing, and is followed by two large polyspike discharges. DC shift: no. Amplitude decreasing: no. ISI increasing: no. This is an arbitrary pattern, so is likely FLC offset. However, this seizure termination is unusual because the offset time is uncertain.

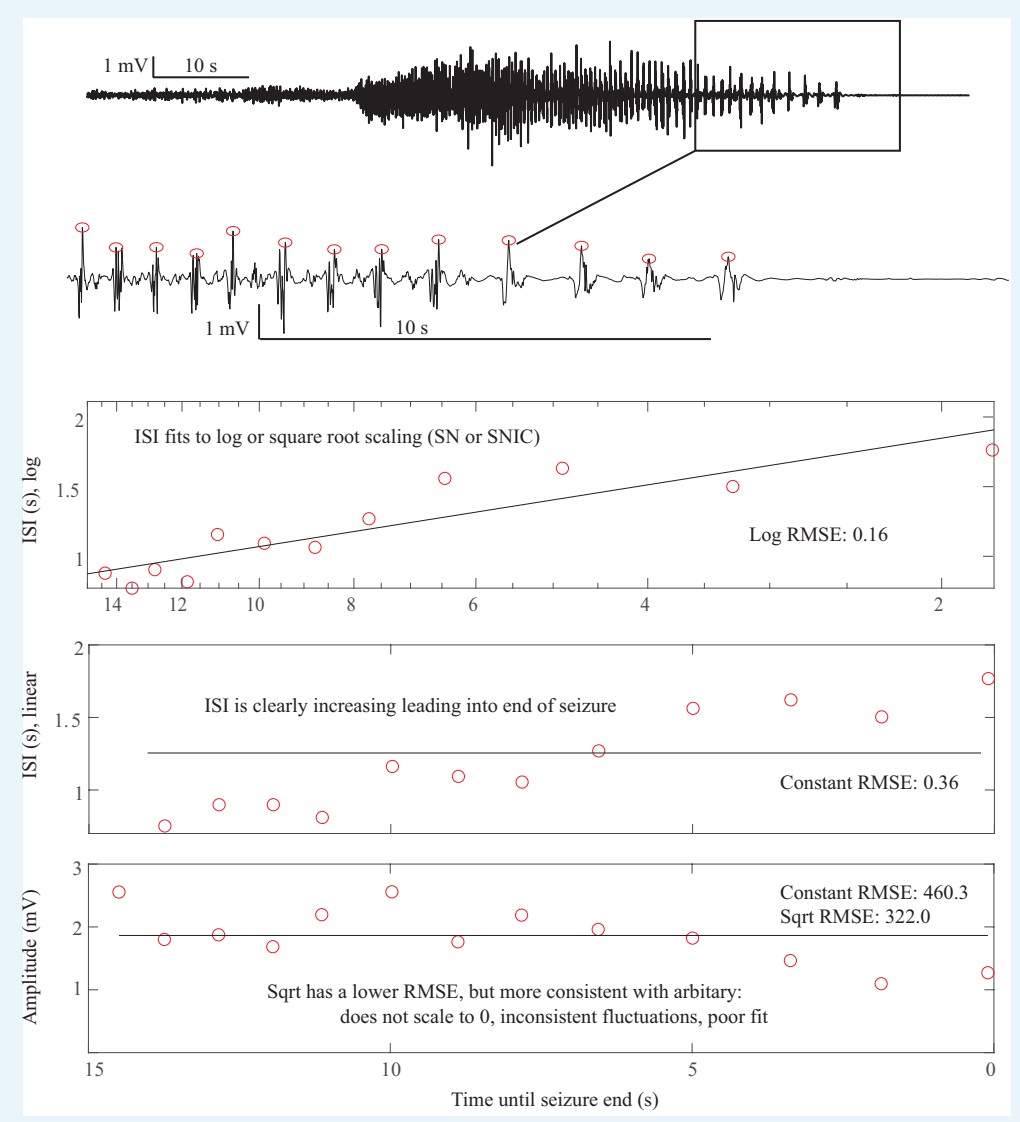

**Appendix 1—figure 7.** Saddle Homoclinic or SNIC offset. In this case, the terminal ISI clearly increases as the end approaches and the amplitude of spikes does not scale all the way to zero (top). For visualization, we reverse the spike plots to coincide with the direction of the seizure, but to determine offset t='time until end of seizure', that is it is counting down. Hence the '0' is at the right of the plots. ISI shows log scaling in 'semilogx' plots, which continues out to 25 s (not shown). The amplitude is somewhat variable but does not diminish to 0. This is consistent with a SH offset. This could also be a SNIC, as a square root function (loglog plot) is very similar at this scale for the ISI (not shown). *Offset Classification*: DC shift: no. Amplitude decreasing: no. ISI increasing: yes. This is a SH (-DC) or SNIC. This is a very common seizure termination dynamotype. *Onset Classification*: Difficult to see at this magnification: seizure started with low voltage fast activity near the end of the 10 s scale bar. DC shift: no. Amplitude increasing: no (seizure started about 10 s before the visible increase). ISI decreasing: no. This is a SN (-DC) or SubH.

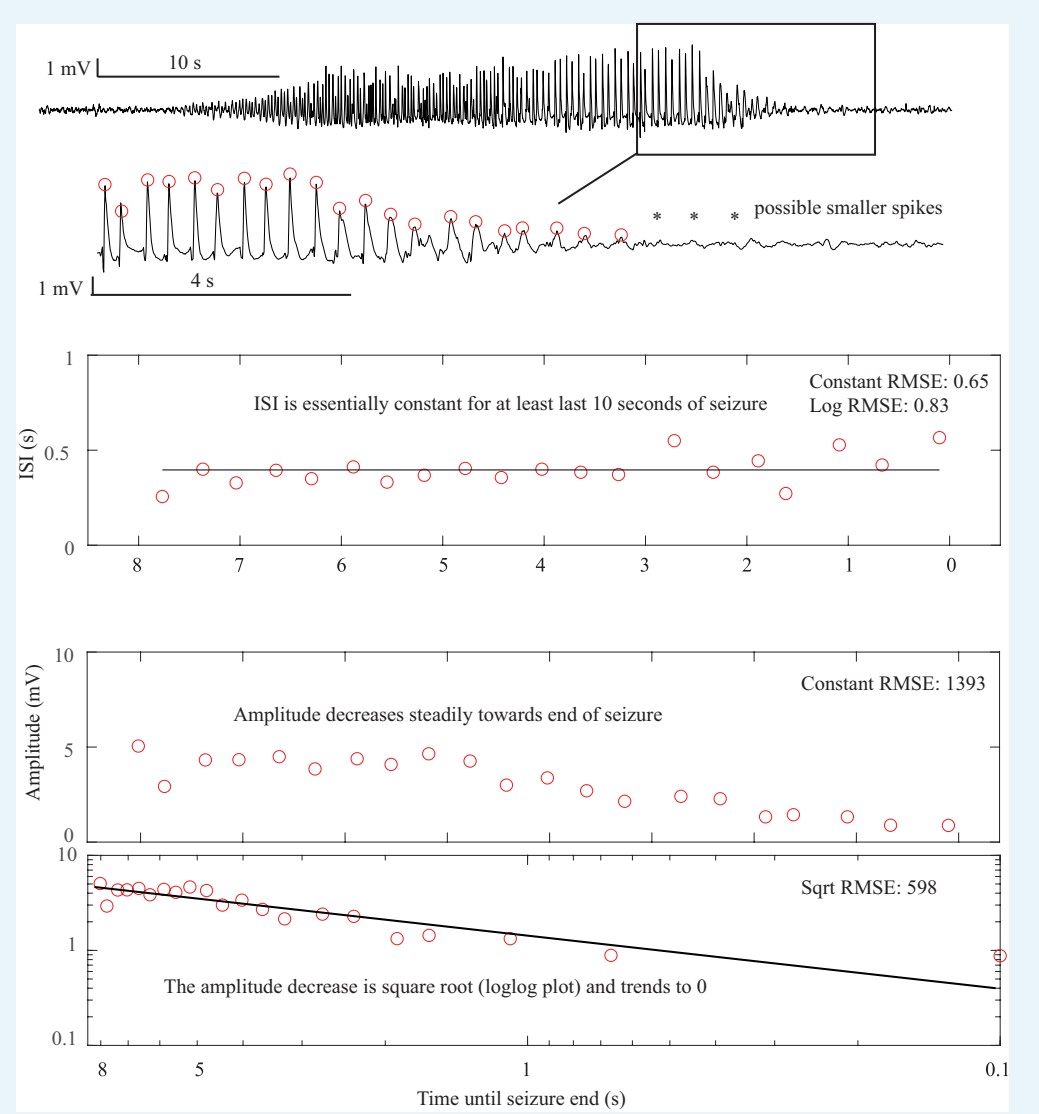

**Appendix 1—figure 8.** Supercritical Hopf onset and offset. This is the seizure from *Figure 2D*. Here, the terminal ISI is nearly constant and does not have any 'slowing down.' The amplitude, on the other hand, clearly diminishes and trends to zero as a square root power law by the end of the seizure. There are even some potential smaller spikes seen after the seizure (*), as if the spikes have vanished into the background noise. In this case, the seizure stops because the amplitude, rather than the frequency, has gone to zero. The constant ISI and diminishing amplitude are indicative of the SupH offset. The same pattern occurs at onset. *Offset Classification*: DC shift: no. Amplitude decreasing: yes. This is a SupH offset. The ISI is also constant, which is consistent but not necessary for the classification due to prioritization. *Onset Classification*: DC shift: no. Amplitude increasing: yes. This is a SupH onset.

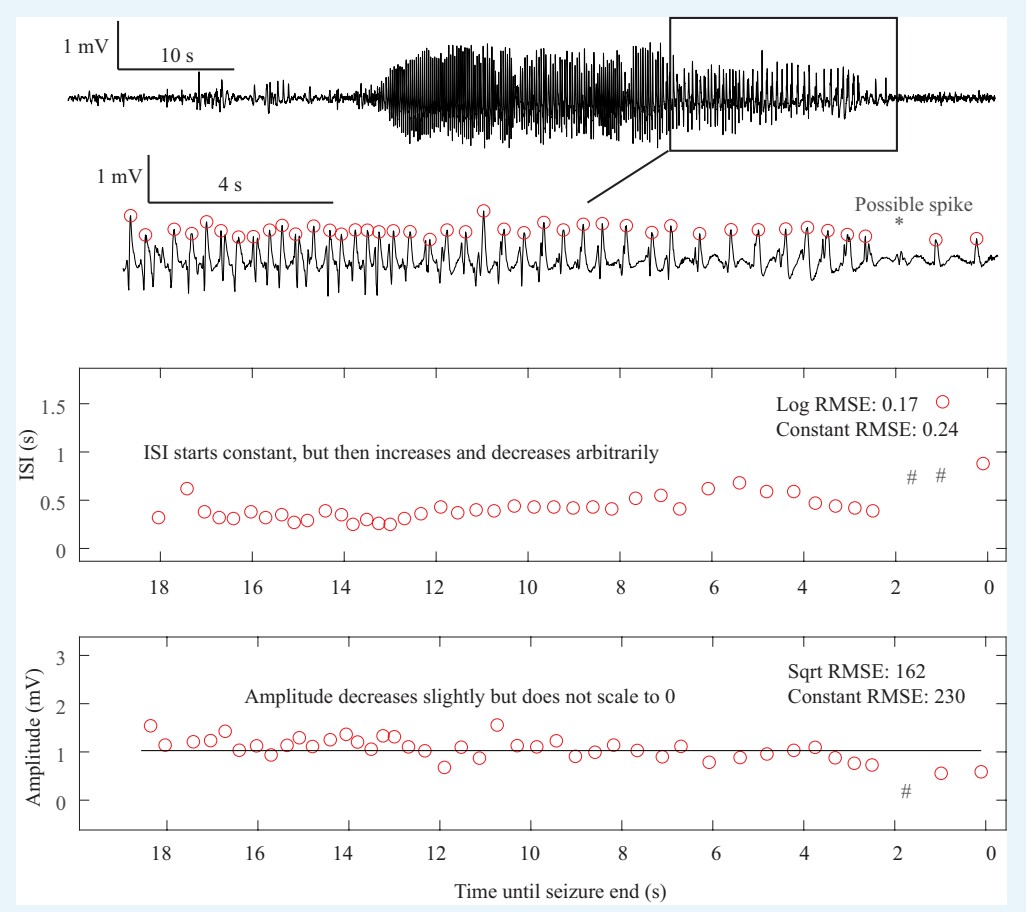

**Appendix 1—figure 9.** Fold Limit Cycle offset. The primary feature of the FLC is the lack of scaling laws. In this case, the ISI is constant until 8 s prior to the termination, then begins to increase until 6.5 s, then decreases until 2 s, then increases again. Even if including the one low-amplitude 'failed' spike at the *, the ISI is still arbitrary and does not follow log or square root scaling laws (#- location of ISI if the extra spike is included). The end of the seizure is abrupt. This arbitrary pattern is consistent with FLC offset. Other examples of FLC-terminal seizures are shown in *Appendix 1—figures 3*, *6*, *10*, *17*. *Offset Classification*: DC shift: no. Amplitude decreasing: no. ISI increasing: no (not consistent). Thus, this is a FLC offset. *Onset Classification*: No consensus, as it had ambiguous onset time. If it started early it could be a SN(-DC) or SubH, or if it started later it could be a SupH.

## I.8 Unusual seizure dynamics

There were rare seizures that contained some unusual dynamical phenomena in the amplitude and ISI. We labeled these as arbitrary, which led to classification as SN or SubH at onset or FLC at offset. However, it is important to reiterate that the vast majority of 'non-zero scaling' dynamics were constant values, rather than arbitrary patterns that appear clinically unusual. We have identified several examples of these unexpected patterns that we wish to highlight (one onset, four offsets). For onset, there was a single example in which a delta brush pattern arises at seizure onset after several polyspikes, shown in *Appendix 1— figure 5*. The pattern lasted <1 s then progressed to arbitrary spiking. While this pattern is 'arbitrary,' it is very distinct from the other patterns seen with SN-SubH onset. For offset, there were four seizures encompassing a wide range of unusual dynamics. *Appendix 1— figure 10* shows three that were classified as FLC. Each of these tracings is highly irregular for a classical seizure offset, and some might question whether the electrode is truly in the seizure focus or if the seizure was correctly labeled; however, in each case a clinical epileptologist verified that this was indeed the focus and there was clear clinical correlation

with these events, ending at seizure offset. These dynamics were not felt to be due to spatial undersampling, as the seizure onset in each case was clearly on the given electrode, and all neighboring electrodes had similar offset dynamics. The fourth offset example had an increasing frequency, and is analyzed in section V (*Appendix 1—figures 17–19*).

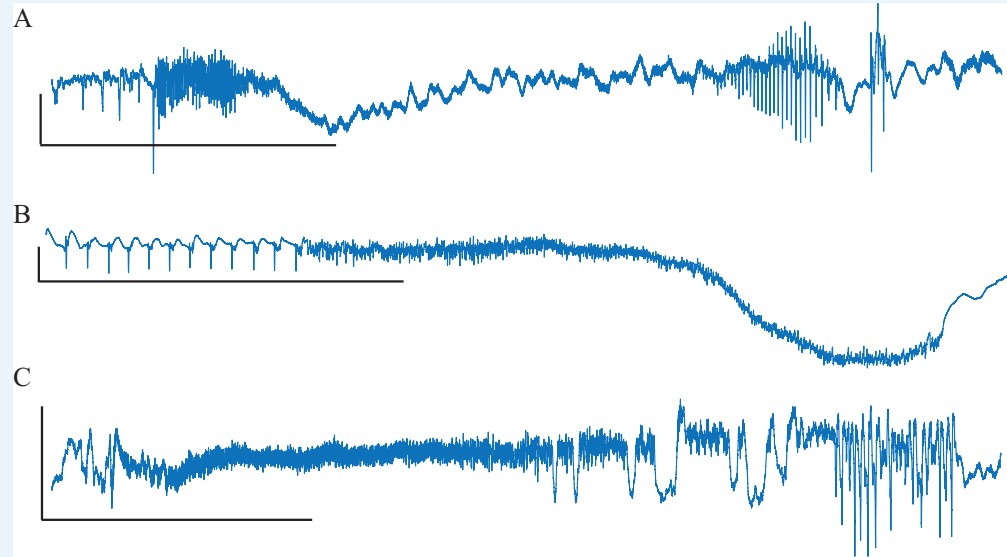

**Appendix 1—figure 10.** Unusual Fold Limit Cycle seizures. Each example is a single electrode from a separate patient, showing the voltage trace at the seizure focus. (**A**) Spike amplitude increases at end of seizure. (**B**) Abrupt start/stop of low voltage fast activity, without any change in frequency. (Note: the slow downward deflection was too slow to be labeled a DC shift). (**C**) Seizure with low voltage fast that progresses into arbitrary spike waves. Scale bars: 1 mV by 10 s.

## I.9 Patient metadata

Full clinical metadata were available for 88 of the patients (*Appendix 1—table 4*). We stratified all patients by their onset and offset bifurcations and evaluated the prevalence of gender, age, pathology, and seizure localization in each case. Due to the vast heterogeneity of electrode locations, all electrodes were stratified into the four main lobes (frontal, occipital, parietal, temporal). Note that not all patients had onset bifurcations available, and some of the rows do not reconcile because some patients had portions of the metadata missing, most commonly the pathology. For simplification and to include all data, all SN and SH were treated as if there were no DC coupling, and so were grouped with the SubH and SNIC bifurcations as shown. There was a correlation between patient age and onset bifurcation, with a higher proportion of SupH in ages 41–60 ($p=0.0019$ after Bonferroni correction). There was no significant correlation between any of the bifurcations and pathology, or location (chi-square test). Note that some bifurcations were rare, so even with 88 patients they did not have sufficient numbers to make strong conclusions.

**Appendix 1—table 4.** Comparison of bifurcations with metadata. All available patient metadata from all centers are included. CD: cortical dysplasia (including two heterotopias). MTS: mesial temporal sclerosis. T: tumor. O: other. F: frontal. Oc: Occipital. P: parietal. Te: temporal.

| Onset | Sex | | Age | | | Pathology | | | | Location | | | |
|---|---|---|---|---|---|---|---|---|---|---|---|---|---|
| | M | F | 1–20 | 21–40 | 41–60 | CD | MTS | T | O | Te | F | Oc | P |
| SN-SubH | 30 | 23 | 13 | 29 | 10 | 20 | 17 | 1 | 1 | 26 | 12 | 5 | 8 |
| SNIC | 3 | 3 | 4 | 2 | 0 | 2 | 2 | 1 | 1 | 2 | 0 | 0 | 0 |
| SupH | 10 | 4 | 2 | 3 | 9 | 2 | 5 | 2 | 0 | 9 | 4 | 2 | 2 |
| Offset | | | | | | | | | | | | | |
| SH-SNIC | 17 | 13 | 5 | 16 | 8 | 10 | 10 | 1 | 1 | 17 | 9 | 2 | 3 |
| SupH | 1 | 0 | 0 | 0 | 1 | 0 | 1 | 0 | 0 | 1 | 1 | 0 | 0 |
| FLC | 23 | 19 | 12 | 19 | 9 | 14 | 12 | 3 | 1 | 20 | 8 | 5 | 8 |

We also compared our results with a previous classification method (**Appendix 1—table 5**). The classifications from **Perucca et al., 2014** are as listed: (i) low-voltage fast, (ii) low-frequency high-amplitude periodic spikes, (iii) sharp activity, (iv) spike-and-wave activity, (v) burst of high-amplitude polyspikes, (vi) burst suppression, and (vii) delta brush. Additionally, we created the category (U) to indicate seizure onsets that were not explained by the Perucca classification. In our analysis, we did not find any seizure onsets that corresponded with (vi). There was no significant correlation between the onset types and any of the patient metadata (chi-square). In addition, we compared these classifications with the onset and offset bifurcations. A chi-squared test was performed on all combinations of Jirsa onset (i.e. based upon bifurcations in **Jirsa et al., 2014**), Jirsa offset, and Perucca onset. There was no significance between any of the pairings. Note that two patients had type (vii) (delta brush, see **Appendix 1—figure 5**), but did not have the expected anti-NMDA receptor encephalitis that was originally described with this pattern (**Schmitt et al., 2012**).

**Appendix 1—table 5.** Comparison of visual with dynamic classifications. Top: Metadata stratified by Perucca classification. There is no clear correlation between Perucca onset type and any of the clinical metadata. Bottom: Comparison of Perucca visual onset classification with our bifurcation analysis. There was no significant correlation between the two classifications. Note that numbers sometimes do not reconcile between different sections and between **Appendix 1—table 4** and 5due to lack of metadata in some patients. U: unclassifiable.

| Onset | Sex | | Age | | | Pathology | | | | Location | | | |
|---|---|---|---|---|---|---|---|---|---|---|---|---|---|
| | M | F | 1–10 | 21–40 | 41–60 | CD | MTS | T | O | Te | F | Oc | P |
| i. | 15 | 15 | 7 | 10 | 12 | 9 | 10 | 1 | 1 | 18 | 5 | 1 | 5 |
| ii. | 9 | 8 | 5 | 8 | 4 | 6 | 6 | 2 | 0 | 7 | 5 | 4 | 3 |
| iii. | 13 | 10 | 6 | 14 | 3 | 7 | 8 | 0 | 1 | 13 | 7 | 0 | 4 |
| iv. | 3 | 2 | 2 | 3 | 0 | 2 | 2 | 0 | 0 | 3 | 1 | 0 | 1 |
| v. | 2 | 2 | 2 | 2 | 0 | 1 | 0 | 0 | 1 | 0 | 1 | 2 | 1 |
| vii. | 1 | 1 | 0 | 2 | 0 | 0 | 0 | 0 | 0 | 0 | 1 | 0 | 0 |
| U | 5 | 2 | 1 | 5 | 1 | 1 | 2 | 2 | 1 | 6 | 4 | 0 | 0 |
| | | | Jirsa onset | | | Jirsa offset | | | | | | | |
| | | | SN/SubH | SNIC | SupH | Sh/SNIC | SupH | FLC | | | | | |

*Appendix 1—table 5 continued on next page*

*Appendix 1—table 5 continued*

| Onset | Sex | | Age | | | Pathology | | | | Location | | | |
|---|---|---|---|---|---|---|---|---|---|---|---|---|---|
| | M | F | 1–10 | 21–40 | 41–60 | CD | MTS | T | O | Te | F | Oc | P |
| Perucca Onset | | i. | | 21 | 2 | 8 | 12 | 0 | 17 | | | | |
| | | ii. | | 14 | 2 | 6 | 10 | 0 | 13 | | | | |
| | | iii. | | 22 | 1 | 4 | 12 | 1 | 14 | | | | |
| | | iv. | | 4 | 1 | 0 | 3 | 0 | 3 | | | | |
| | | v. | | 3 | 0 | 2 | 1 | 1 | 3 | | | | |
| | | vii. | | 2 | 0 | 0 | 0 | 0 | 2 | | | | |
| | | U | | 1 | 0 | 2 | 2 | 0 | 0 | | | | |
| Jirsa Offset | SH/SNIC | | | 30 | 3 | 14 | | | | | | | |
| | SupH | | | 1 | 1 | 3 | | | | | | | |
| | FLC | | | 53 | 3 | 7 | | | | | | | |

## II. Construction of the dynamotype model and the dynamic map

In this work, we used the model in **Saggio et al., 2017** and we refer to that paper for details. It consists of three ordinary differential equations:

Equation 1:

$$\frac{dx}{dt} = -y$$

$$\frac{dy}{dz} = x^3 - \mu_2(z)x - \mu_1(z) - y\left(\nu(z) + x + x^2\right)$$

$$\frac{dz}{dt} = -c\left(\sqrt{(x - x_S(z))^2 + y^2} - d^*\right)$$

The first two of them are based on the normal form of the unfolding of the degenerate Takens-Bogdanov singularity (**Dumortier et al., 1991**). They depend upon three parameters $(\mu_2, \mu_1, \nu)$. We can consider a sphere, with radius R, centered at the origin in the three-dimensional parameter space. On the spherical surface there are curves of bifurcations that divide the surface into regions with different sets of attractors. It can be shown that, up to a certain value of the radius, the bifurcation diagrams on the spherical surface do not change (**Dumortier et al., 1991**). For this reason, the number of parameters relevant to describe the bifurcations in the system can be reduced, from the three Cartesian parameters $(\mu_2, \mu_1, \nu)$ to the two spherical parameters $(\theta, \varphi)$ keeping the radius fixed. In the present work we use a radius R=0.4. This reduction allows for easier analysis and visualization of the parameter space. A flat sketch of the spherical surface and its bifurcations can be found in the map in **Figure 4C**. A flat projection of the real map, as shown in **Figure 5**, is obtained with Lambert equal area azimuthal projection. Details on how to reconstruct these maps can be found in **Saggio et al., 2017**. Movement on the sphere is promoted by making the three Cartesian parameters (or the two spherical ones) depending on a third variable, z, acting on a slower timescale described by the parameter $0 < c \ll 1$. Movement is implemented so that, when the fast subsystem $(x, y)$ is in the resting state $(x_S, 0)$, it moves towards the onset bifurcation, and when the distance from the resting state is bigger than $d^*$ it moves toward the offset bifurcation. When $d^* > 0$ this gives periodic bursting.

The shape of the path along which the fast subsystem moves is taken to be the arc of the great circle linking the offset point $A$ to the onset point $B$ on the sphere. It is described by the following parameterization:

Equation 2:

$$\mu = \begin{pmatrix} \mu_2 \\ -\mu_1 \\ \nu \end{pmatrix}$$

$$\mu(z) = R(\boldsymbol{e} \cos z + \boldsymbol{f} \sin z)$$

where $\boldsymbol{e} = \boldsymbol{A}/R$ and $\boldsymbol{f} = ((\boldsymbol{A} \times \boldsymbol{B}) \times \boldsymbol{A})/\|(\boldsymbol{A} \times \boldsymbol{B}) \times \boldsymbol{A}\|$. An example of path for each type is sketched in *Figure 4C* with a black arrow.

In this model, the dynamics of $z$ exploit feedback from the fast subsystem. This is possible because the fast subsystem exhibits bistability and hysteresis along the bursting path ('hysteresis-loop bursting' in Izhikevich nomenclature; *Izhikevich, 2000*). This allows having bursting with one slow variable only. Other possibilities exist, in which the slow subsystem oscillates independently from the fast one ('slow-wave bursting' in Izhikevich nomenclature; *Izhikevich, 2000*). The hysteresis-loop mechanism is well suited to model the mechanisms leading to seizure termination, which are thought to be triggered by the presence of the seizure itself, as done in *Jirsa et al., 2014*. For this reason, in this work, we focus on this type of mechanism for bursting. However, a slow-wave mechanism may be possible and we refer to *Saggio et al., 2017* for details on how to obtain slow-wave bursters for all the types of the taxonomy using the same map.

## II.1 Definition of onset

While previous works on the taxonomy define the onset bifurcation as the bifurcation that starts the oscillatory activity (*Izhikevich, 2000*; *Jirsa et al., 2014*; *Saggio et al., 2017*), we here define the onset bifurcation as the first one that destabilizes the resting state (or healthy condition). This affects the types in which transition to the 'active rest' (onset bifurcation with the present definition) precedes the transition to the limit cycle (onset bifurcation with the definition in the literature). This does not change the mechanism or dynamics of seizure initiation, merely the nomenclature. We chose this method because under clinical conditions the 'seizure initiation' is defined as the first departure from resting state. We note that this definition, which we used for practical reasons, highlights that the mechanisms triggered by the seizure, such as the inversion in the direction of the slow variable, may be triggered before oscillations are evident. Examples of this effect are shown in the next paragraph, in which we use a different label for two of the types in *Saggio et al., 2017* in which SupH (which started the oscillations) was preceded by SN (which destabilized the resting state): SupH/SH becomes a different realization of SN/SH, and SupH/SupH becomes a different realization of SN/SupH.

## II.2 Examples of dynamotypes

In *Appendix 1—figure 11*, we show bifurcation diagrams for the types in the bistability region in the upper portion of the map from *Figure 4C*. Note that, for all types except SN/SH, additional bifurcations (aside from the first and last used in the dynamotype name) may be necessary for the type to exist (to allow the presence of the hysteresis-loop). In particular, all the types in this portion of the map show a baseline jump, due to the crossing of the SN bifurcation. By our definition, these would be classified as SN onsets. However, in two of the examples shown it is not until the trajectory then crosses the SupH bifurcation that the oscillation begins. These are the conditions which our previous work had classified as SupH onsets, and which we now classify as SN. We based this decision on the fact that our human data showed this precise combination several times, but clinicians always identified the SN as the start of the seizure. The ability to identify a seizure trajectory prior to crossing the oscillatory bifurcation has intriguing implications for future work.

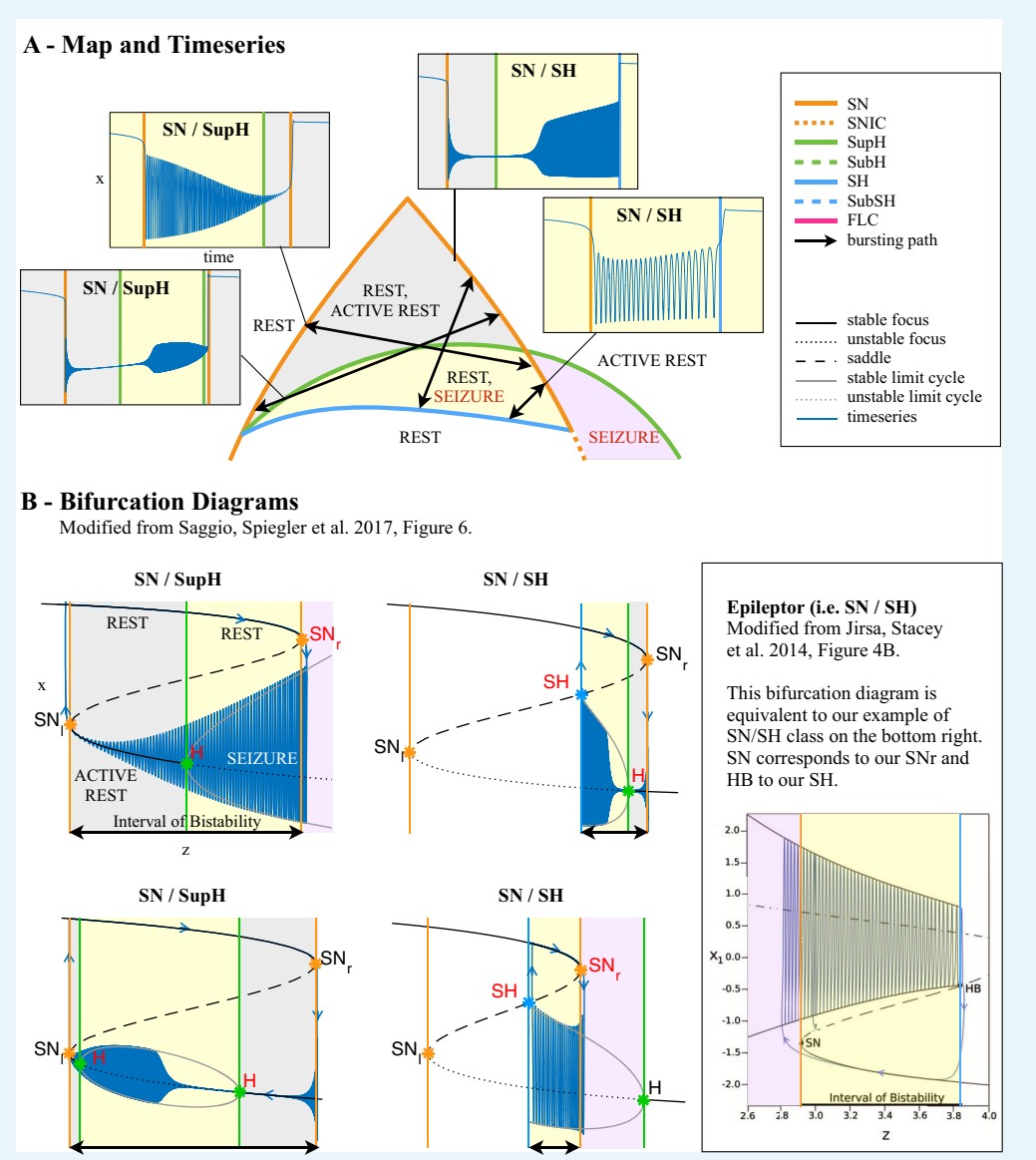

**Appendix 1—figure 11.** Dynamotypes with baseline shift. (**A**) Zoom of the bistability region in the upper part of *Figure 4C*. In this region, the dynamotypes are combinations of SN, SupH, and SH. All begin with a SN bifurcation, which causes a baseline shift. The background of the timeseries is shaded with the same color of the portion the map traversed. Vertical colored lines mark the value of z at which bifurcations on the map are crossed. Two of the trajectories (top right, bottom left) do not have sustained oscillations until after crossing the SupH bifurcation a short time after the SN onset. (**B**) Bifurcation diagrams for the same types. Onset and offset bifurcations for the oscillatory phase are marked with red letters. $SN_r$ refers to the SN curve on the right of the map, $SN_l$ to the SN at the left. Inset: comparison with the 'Epileptor' type from *Jirsa et al., 2014*.

In *Appendix 1—figure 12*, we show bifurcation diagrams for the types in the bistability region in the lower portion of the map. Here types occur without baseline shifts, even when starting with a SN bifurcation. This makes it impossible to discern SN from SubH onsets based on *Figure 1*.

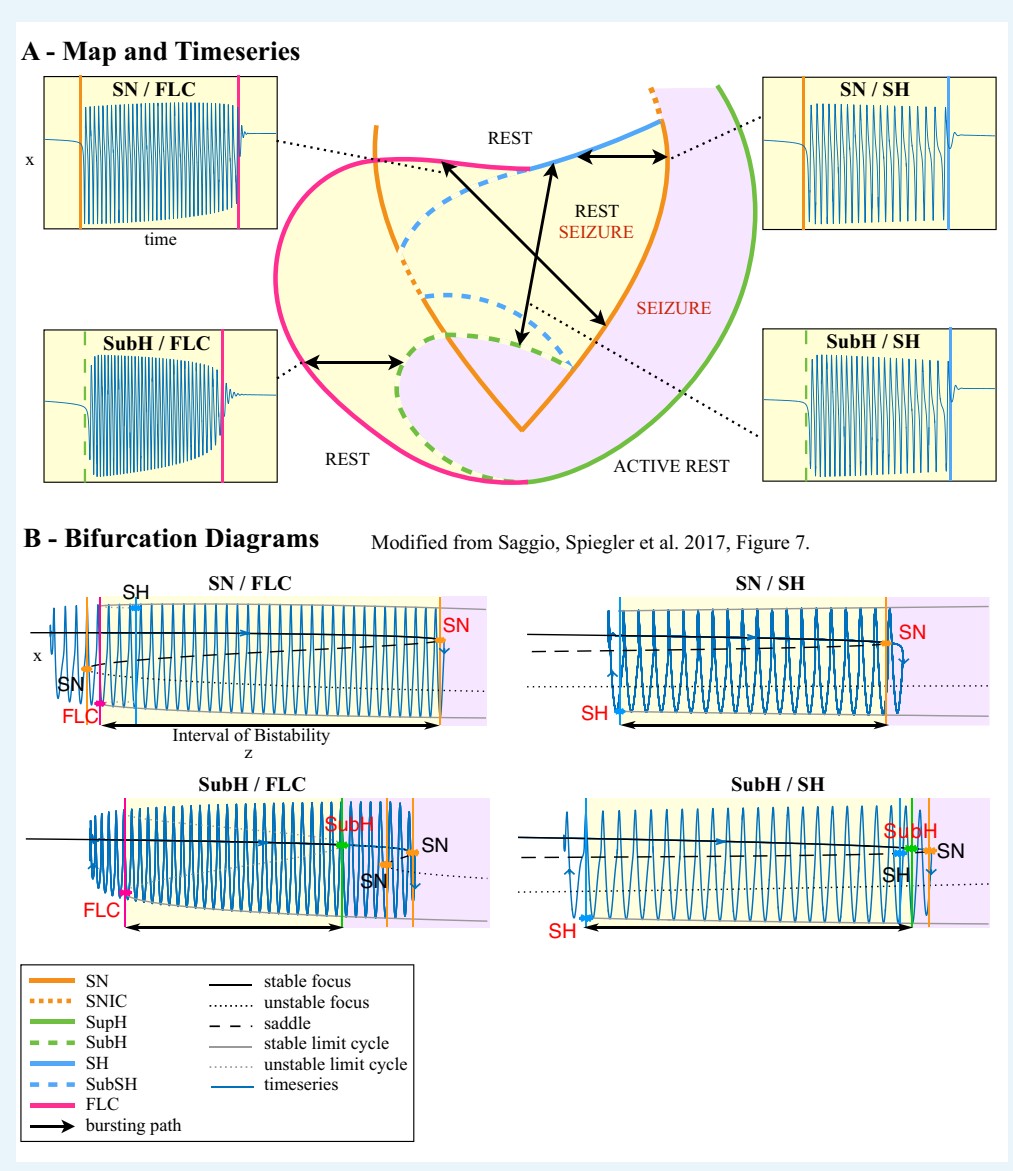

**Appendix 1—figure 12.** Dynamotypes without baseline shift. (**A**) Zoom of the bistability region in the lower part of *Figure 4C*. For each type possible in this region we show an example of the timeseries. The background of the timeseries is shaded with the same color of the portion the map traversed. Vertical colored lines mark the value of z at which bifurcations on the map are crossed. (**B**) Bifurcation diagrams for the same types. Onset and offset bifurcations are marked with red letters.

## II.3 Ranking the complexity of each dynamotypes

We can use mathematical arguments to rank the types according to how complex they are, with more complex types being more difficult to encounter in models, and potentially also in nature. The first criterion to define the complexity of a type is based on the minimum number of parameters (called 'codimension') necessary to describe a map in which a path for this type can be placed (see *Golubitsky et al., 2001*, in which this criterion is introduced, for a more mathematical definition). We base this analysis on the hysteresis loop types proposed in *Saggio et al., 2017*, and which are illustrated in *Figure 4* and *Appendix 1—figures 11, 12*. The basis of the dynamic trajectory (or 'bursting path') of this seizure model is that, once the seizure begins, there is a feedback mechanism in which the slow permittivity variable

tends to terminate the seizure. Using this seizure mechanism, the least complex type is SubH/FLC, which requires a codimension of two. Following that are SN/SH, SN/SupH, SN/FLC, SupH/SH, SupH/SupH and SubH/SH. These types require a codimension three, which is the basis for our current model (*Saggio et al., 2017*). In *Figure 4C* of the main paper we show a representation of this map that is two-dimensional for more readability, but the map is in fact lying on a spherical surface in a three-parameter space (*Figure 4B*). The other types of the taxonomy require higher codimensions (*Saggio et al., 2017*). We can further rank types with the same codimension based on how many bifurcation curves their paths must cross on the map (*Saggio et al., 2017*). For example, type SN/SH (two curves to cross) is less complex than the other codimension three types (three or more curves to cross).

In *Appendix 1—figure 13* we show the types' ranking superimposed with results from our classification of human seizures (adapted from *Figure 2F*). In the clinical data (DC recordings only), the most common types are SN/FLC, SN/SH and possibly SubH/FLC and SubH/SH, which have low complexity (complexity 2 and 3). However, it is worth recalling that types with FLC offset may be overestimated due to noise. Thus, the clinical data are consistent with the predicted complexity. Interestingly, though not assessed in this work, absence seizures may be best described with the most common SubH/FLC, which can be found in the lower bistability region of the map (see Discussion). Note that the number of seizures in each type is based upon our method of identifying the first bifurcation, while the complexity is based upon the initiation of the limit cycle (*Saggio et al., 2017*). Reconciling these methods would likely result in some of the SN onset seizures being reclassified as SupH. Additionally, it should be noted that this complexity refers only to seizures that are produced by the hysteresis mechanism. Other plausible mechanisms are independent changes in the permittivity variable and fluctuations due to noise, both of which can produce seizures with different codimensions/complexity than those shown here.

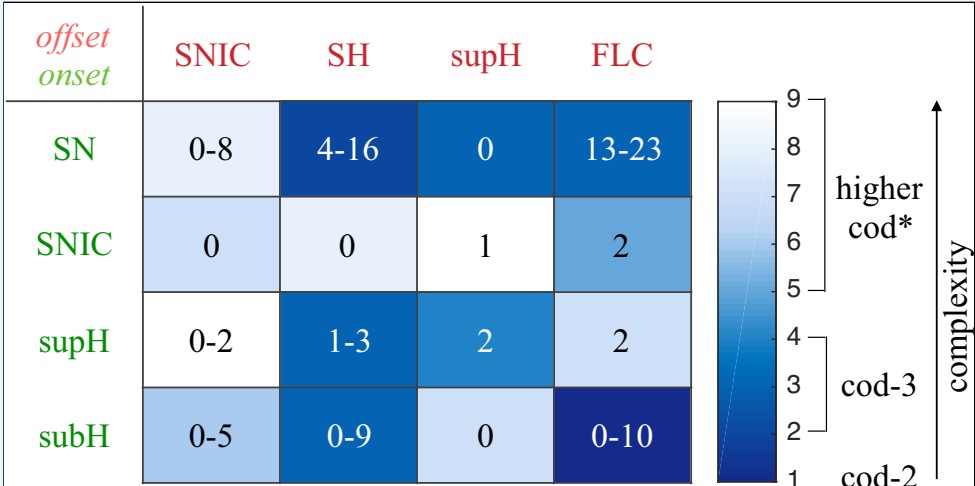

**Appendix 1—figure 13.** Complexity of types and data classification results. The complexity of the 16 dynamotypes is shown, with darker colors being the least complex. Superimposed on that are the number of seizures from *Figure 2E–F* in each type. Note that some of the seizures could not be distinguished between SH and SNIC offsets, or between SN (-DC) and SubH, so for these types there is a range. *-The codimension of some of these types is presumed, see *Saggio et al., 2017* for details.

## II.4 Modeling noise-driven or bifurcation-driven transitions

When $d^* > 0$ this model produces periodic bursting, while $d^* = 0$ gives no bursting at all since the whole system would be in a fixed point. In the presence of noise, we have periodic bursting if the variance of the noise is smaller than $d^* > 0$. However, by setting a $d^*$ value close

to the noise variance, and initial conditions for z within the bistability region, we have a different scenario. In this case a seizure can be initiated if noise is strong enough to bring the system beyond the separatrix (i.e. outside the domain of attraction of the fixed point and within the domain of attraction of the limit cycle). Noise-induced transitions like this one have been proposed as a mechanism to initiate a seizure (**Lopes da Silva et al., 2003b**). However, in our model, once the fast subsystem has left the fixed point of the fast variables, the whole system is no longer in a fixed point and the slow variable will activate to bring to seizure offset. Setting $d^*$ close to the noise variance thus allows a mixed scenario in which the onset is noise-driven (with the transition to seizure being more likely the closer the system is to the bifurcation) and offset is determined by a deterministic slow variable. Noise-driven transitions require the presence of bistability. The probability of having such a transition increases when approaching the bifurcation point, which gives origin to this bistability, from within the bistability region, since the separatrix gets closer to the resting state. In the presence of noise thus this mechanism can contribute to seizure onset. Noise-driven transitions lack any scaling law. However, the only two bifurcations here that allow for noise-driven transition in their proximity (because they feature bistability) are SN and SubH, which also lack any scaling law. This implies that, if noise-induced transitions rather than bifurcations are causing seizure onset in any seizure in our dataset, these seizures would be classified under the 'SN or SubH' label. The presence of such events in our dataset would not undermine our classification since we already know that 'SN or SubH' onset is overestimated. However, it is interesting that in these cases, even though the onset is not caused by a bifurcation, the label in our classification would describe the bifurcation that creates the bistability necessary for the noise-induced transition to occur.

## II.5 Biophysical meaning of the variables

As discussed in the main text, we use a bursting model as a phenomenological model for seizure generation and termination. This implies that the variables of the model do not have any explicit biophysical meaning. While we can make considerations, based on the timescales at which they operate, on which type of biophysical processes are best candidates to be represented within a given variable (e.g. neuroelectric processes for the fast variables, neurochemical substances for the slow ones), the biophysical meaning of the variables may depend on the specific type of epilepsy or even on the specific patient.

One strategy to identify the correlates of the model variables could be to rely on the comparison with more realistic biological models. Our phenomenological model is based on the unfolding of the degenerate Takens-Bogdanov singularity. This singularity appears to be quite common both in neural models (**Kirst et al., 2015**) and in neural population models, such as the Jansen-Rit or the Wendling-Chauvel models (**Touboul et al., 2011**). When this singularity exists, the bifurcation diagram in its surroundings is equivalent to the one shown in our model. This allows us to produce a mapping between the variables and parameters of the models in which this singularity can be found and those of the phenomenological model in **Saggio et al., 2017**. Applications of this model, however, do not necessarily depend on the understanding of the biophysical correlates of its variables. This is true for the investigation of the synchronization and propagation patterns linked to each dynamotype—the dynamic phenomena are invariants and do not depend upon the specific physiological parameters.

However, there is clearly great utility in determining how specific physiological parameters can be related to this model, specifically for the application of designing methods to stop or prevent seizures by acting upon the slow and ultra-slow variables. In particular, it will be important in future work to model the effect of stimulation on specific dynamotypes. In this case, an external current applied to the fast subsystem will modify the additive term $\mu_1$.

## II.6 Limitations of the model

Planar bifurcations. One limitation of the present study is that it considers only planar (i.e. two variables) bifurcations as possible onset or offset mechanisms. When considering higher dimensional systems, additional offset bifurcations exist, while the possible onsets are unchanged (*Dumortier et al., 1991*). However, the exact number of additional bifurcations is unknown (*Kuznetsov, 2004*).

Planar bifurcations are used to organize the bifurcation diagrams of many neural populations or field models for seizures, even high dimensional ones (*Breakspear et al., 2006*; *Marten et al., 2009*; *Touboul et al., 2011*; *Taylor et al., 2013*; *Meijer et al., 2015*). Phenomenological models in the literature are planar or have planar fast subsystems (*Kalitzin et al., 2010*; *Benjamin et al., 2012*; *Terry et al., 2012*; *Jirsa et al., 2014*; *Hutchings et al., 2015*; *Sinha et al., 2017*). The presence of additional variables, acting on a different timescale (still fast as compared to the slow permittivity variable z of the present paper), are used to create spike and wave discharges (*Marten et al., 2009*; *Wang et al., 2012*; *Jirsa et al., 2014*). In these cases, non-planar bifurcations may appear in the model, such as period doubling of limit cycles (*Marten et al., 2009*), but they are not used to start or stop a seizure, but rather to change the number of spikes riding on the wave. This does not affect the present work, which instead focuses solely on the onset/offset dynamics. We cannot exclude the possibility of non-planar bifurcations playing a role in seizure offset, but with current knowledge and available data it may not be possible to distinguish them from planar bifurcations. Using a planar system also excludes the possible role of low dimensional chaos in seizure generation (*Iasemidis et al., 1994*).

Bursting with higher dimensional fast subsystems. Our model is built upon the minimum number of dimensions necessary to produce the bursting behavior. Obviously, more complex models could also generate similar activity with different bifurcations (*Izhikevich, 2000*; *Kuznetsov, 2004*). Among those known, some are characterized by non-zero frequency at the bifurcation point: the subcritical flip of limit cycles (or period doubling), subcritical Neimark-Sacker, FLC on a homoclinic torus and Blue-sky bifurcations. Other have zero-frequency at the bifurcation point: Saddle Focus Homoclinic and Focus Focus Homoclinic. The latter requires at least a four-dimensional fast subsystem. In addition, planar bifurcations may display additional features when they occur in higher dimensions, such as baseline jumps in a FLC.

Other mechanisms of dynamic trajectories. As stated previously, our model assumes a hysteresis in the seizure trajectory, in which the slow variable acts to pull the system out of a seizure once it starts. There are other methods of starting seizures, such as independent changes of the slow permittivity variable causing a crossing of the bifurcation, which will be the subject of future work.

# III. Switching between dynamotypes due to fluctuations and ultra-slow modulations

In the model above, the path along which the fast subsystem slowly moves is a simple arc linking the offset and onset points. However, movements promoted by real changes of parameters can be more complex. As a proof of concept of the effects that this can have on the system, we considered modifications of the path due to (i) ultra-slow drifting of the offset and onset points $A$ and $B$ and (ii) fluctuations produced by noise.

The ultra-slow drift was obtained as in *Saggio et al., 2017*. The offset point moves along an arc of great circle linking the initial offset $A_1$ to the final offset $A_2$ with velocity $c_A \ll c$. This movement is promoted by an ultra-slow variable $u$. Analogously for the onset point $B$, from $B_1$ to $B_2$ with velocity $c_A \ll c$, movement is promoted by the ultra-slow variable $w$. The equations are:

$$\frac{du}{dt} = c_A$$

$$\frac{dw}{dt} = c_B$$

$$A = A(u) = R(g \cos u + h \sin u)$$

$B = B(w) = R(l \cos w + m \sin w)$ where $g = A_1/R$, $h = ((A_1 \times A_2) \times A_1)/\|(A_1 \times A_2) \times A_1\|$, $l = B_1/R$ and $m = ((B_1 \times B_2) \times B_1)/\|(B_1 \times B_2) \times B_1\|$.

Depending on the location on the map of the points $A_1$, $A_2$, $B_1$ and $B_2$, this type of drift can cause the system to produce seizures of different types in different moments, which is consistent with the results of our longitudinal analysis. The addition of noise fluctuations can potentially alter the trajectory of the seizure while the seizure is ongoing, which might explain some of the unusual seizures identified in the data, such as in *Figure 5*. For example, to obtain transitions between the SN / SupH type and a type with SH offset (*Figure 5A*), we simulated a seizure that began on a path for SN / SupH slowly drifting downwards toward a path for SN / SH. The two paths lie very close on the map. We then added normally distributed noise, which allowed the system to deflect off its normal trajectory. This noise then allowed to get close to different bifurcations within a single seizure. Transitions of the type observed in data (which is a subcase of type SN / SH) were observed in 10 of the 100 simulations we ran with the same settings. The other simulations produced mostly SN / SH seizures (43/100) and some SN / SupH (8/100), other forms of transitions within seizures and 12 examples of status epilepticus in which the system entered a long seizing state that did not resolve by the end of the simulation. When more than one seizure occurred in the same simulation, we classified only the first unless otherwise stated. Seizures were classified by determining the onset and offset bifurcations crossed by the path, rather than by analyzing the frequency/amplitude profile of the time series.

Of course, these rates of occurrence of types are not meant to reflect real rates in data, since parameters for the simulations were carefully chosen to obtain the specific transition within seizure observed in data. However, since simulation settings were the same for the 100 trials, these results show that fluctuations can easily cause different types to occur when these types are close on the map.

Integration settings for *Figure 5B-C*: Euler-Meruyama method; integration step 0.002 s; simulation length 10000 s; initial conditions were set to 0 for all variables; noise variance was 0.005 for fast variables, 0.0005 for the slow variable and 0.002 for ultra-slow variables. Model parameter settings: $c = 0.002$; $c_A = 0.00005$; $c_B = 0.000005$; $d^* = 0.3$; $R = 0.4$; $A_1 = (0.2731, -0.05494, 0.287)$; $B_1 = (0.3331, 0.074, 0.2087)$; $A_2 = (0.3524, 0.05646, 0.1806)$; $B_2 = (0.3496, 0.07955, 0.1774)$. The procedure to compute the amplitude and frequency of the limit cycle across the map, as shown in *Figure 5H-I*, is the same as in *Saggio et al., 2017*.

## IV. Analysis and simulation of status epilepticus

In 44 out of the 100 simulations we ran in the previous section, after generating one or more seizures the system escaped from the bistability region to be trapped in the 'seizure only region'. Activity in this region of the map is analogous to status epilepticus. The dynamic map allows us to analyze how this transition occurred. In 11 cases, this escape occurred when the imposed ultraslow drift caused the system to cross the SH curve at the lower border of the upper bistability region and then enter the 'seizure only' region through the SN bifurcation. In the remaining 33 cases, instead, the system escaped the bistability region directly through the SN curve, thus before the downward ultra-slow drift could play a role in the escape. Our simulations show an example of each scenario (*Appendix 1—figure 14*). To assess the role of noise in this phenomenon, we ran five simulations for each of several levels of noise variance. Decreasing the noise variance scaling factor $a$ from 0.0005 to 0.0002 diminished the number of times the system ended up in status epilepticus, from 3/5 to 1/5, then to 0/5 for $a = 0.0001$ and below. One of the status epilepticus obtained with a = 0.0005 is shown in *Figure 5*. These results indicate that high levels of noise variance can contribute to the system being trapped in the seizure only regime.

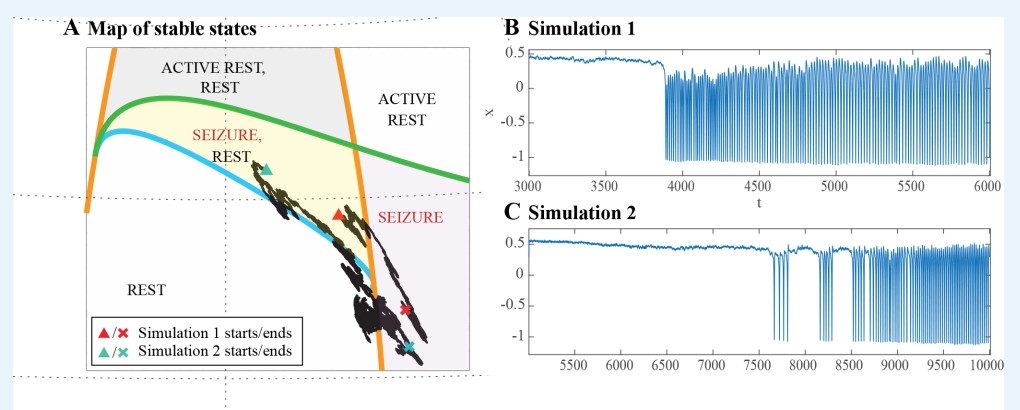

**Appendix 1—figure 14.** Escape from the bistability region into the seizure-only region. (**A**) In some of the simulations performed with an ultra-slow downward drift and noise, the system escaped from the bistability region (yellow) into the seizure-only region (lavender). The escape could occur directly through SN (orange curve), as in Simulation 1, or passing below SH (blue curve) and then through SN, as in Simulation 2. (**B-C**) Time series for Simulation 1 and 2. The clinical relevance of this is to illustrate how some episodes of status epilepticus might arise uniformly (B) while others have a stuttering onset (C).

Integration settings: Euler-Meruyama method; integration step 0.002 s; simulation length 10000 s; initial conditions were set to 0 for all variables; noise variance was $a*$ (10 for fast variables, 1 for the slow variable, 5 for ultra-slow variables). Tested values of $a$ were: 0.0005, 0.0002, 0.0001, 0.00005. Model parameter settings: $c = 0.0001$; $c_A = c_B = 0$; $d* = 0.3$; $R = 0.4$; $A_1 = (0.3483, 0.03698, 0.1931)$; $B_1 = (0.3331, 0.074, 0.2087)$; $A_2 = B_2 = (0.279, 0.2187, 0.1854)$.

These findings are very similar to examples from the clinical cohort. Two of the patients had unrecognized episodes of nonconvulsive status epilepticus lasting over 2 hr during sleep (see *Figure 5*). We analyzed these seizures in the same manner as the others. One began with a brief seizure very similar to the most common SN/SH type (though this was not DC-coupled, so the onset could potentially be SubH, and the offset SNIC). However, instead of terminating, it then transitioned through a SupH into irregular firing. The second example started with SupH and went immediately into irregular firing. After each seizure started, there were numerous transition periods in which the seizure changed dynamics. The first patient had long periods of arbitrary amplitude and ISI, which occasionally organized into a brief SN-SubH/SH pattern before reverting to the disorganized firing (*Appendix 1—figure 15*). In this patient, the seizure lasted over 24 hr, but the final offset could not be determined due to poor signal quality. In the second patient, there were prolonged periods of primarily constant ISI and amplitude, interspersed with brief periods of arbitrary firing, and the seizure finally ended with a SupH (*Appendix 1—figure 16*). There were also several unusual periods in which the ISI became uniform for several seconds. During these periods there was nearly perfect ~7 Hz spike wave discharges with constant amplitude, which were much more periodic than typical seizures. In both of these examples, the dynamical behavior is entirely consistent with the model: the seizure repeatedly approaches an offset bifurcation then returns to the seizure regime with disorganized (i.e. noisy) firing. This offset was SH in our simulations due to the chosen parameters, but can also be SupH under different conditions as it resides in the same bistability region, e.g. if the seizure started in the top of the region.

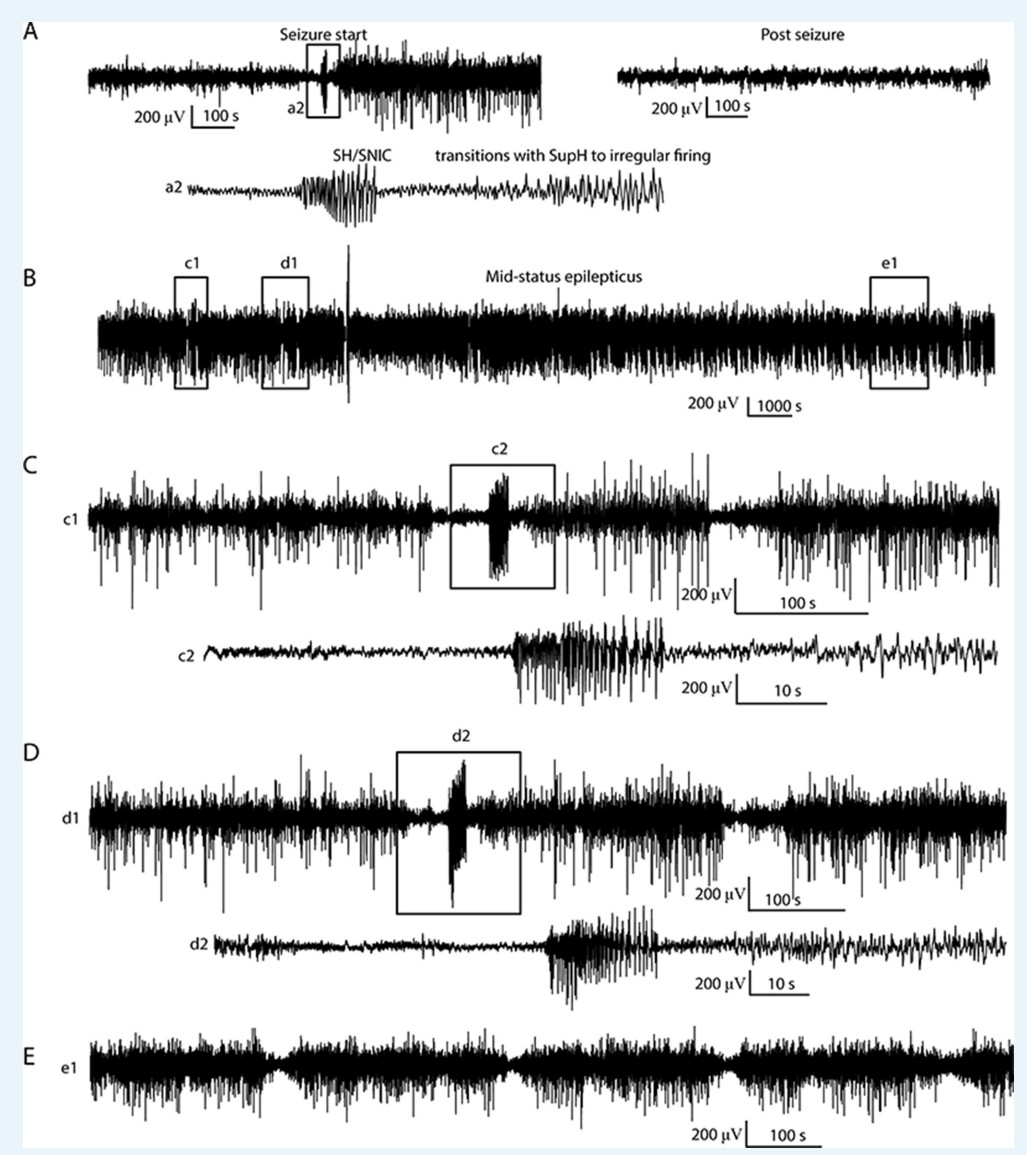

**Appendix 1—figure 15.** Status Epilepticus patient 1. (**A**) Initiation (left) of seizure that begins to terminate with a SH/SNIC bifurcation, then transitions into a SupH onset. The post-seizure baseline (right) returns to prior levels. The seizure lasted >24 hr, but due to signal dropout the exact time of termination was not recorded. Inset a2: expanded view of indicated portion of seizure start. (**B**) Subclinical status epilepticus was characterized by irregular firing interspersed with periods of organized, lower amplitude EEG. (**C, D**) Expanded view of (c1, d1) from B, with further expansion of (c2, d2). The irregular firing organizes several times into discrete seizures that nearly terminate with SH dynamics, but revert to the previous pattern. (**E**) Later in the seizure, there is waxing/waning of the firing pattern, similar to *Figure 5D–F*.

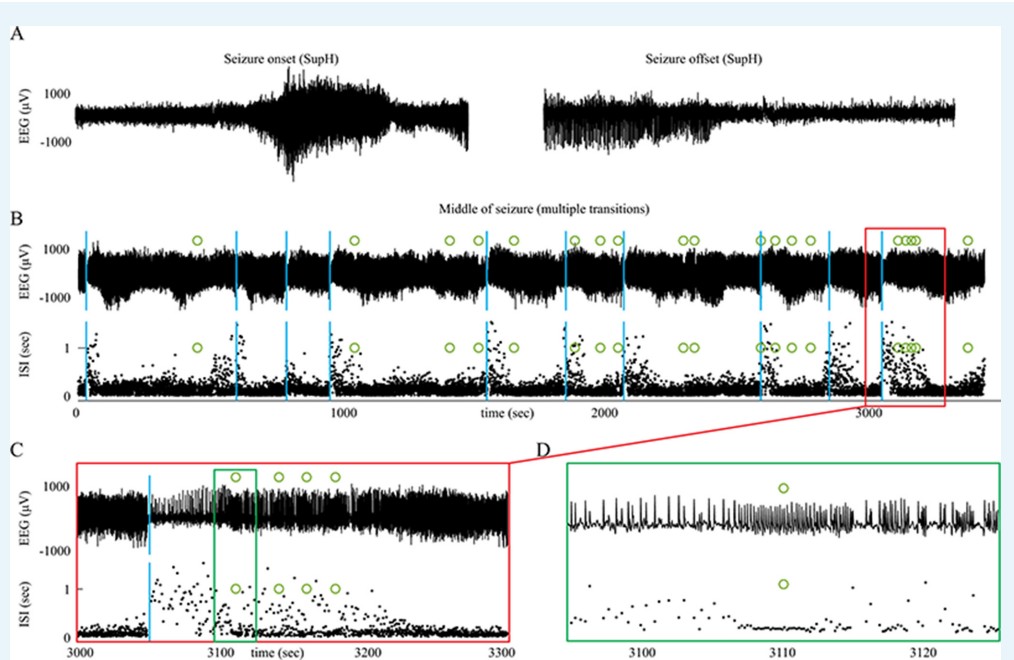

**Appendix 1—figure 16.** Status Epilepticus patient 2. (**A**) Onset and offset of a seizure lasting 5 hr were both SupH bifurcations. (**B**) During the seizure, there were many transitions in which the dynamics altered. Top: voltage trace. Bottom: interspike intervals (ISI) show multiple periods in which nearly-constant ISI is interrupted by disorganized firing. Blue lines indicate these transition periods. (**C**) Expanded view of red box in B. (**D**) Expanded view of green box in C. Green circles indicate runs of highly periodic, 7 Hz spike waves, which became more frequent later in seizure.

## V. Example of accelerating frequency at seizure offset

One of the seizures we classified as FLC in *Figure 2E* had increasing frequency towards the end, shown in *Appendix 1—figure 17*. The best fit to the ISI for this seizure was a reversed power law scaling. While acceleration at the end of a seizure is peculiar from the clinical point of view, it fits with both FLC and SupH bifurcations, which do not have specific scaling laws for the behavior of the frequency. As the amplitude of this seizure was arbitrary (rather than decreasing with square root), it is most consistent with the FLC. We analyze it here to show how different frequency behaviors can be obtained in the model.

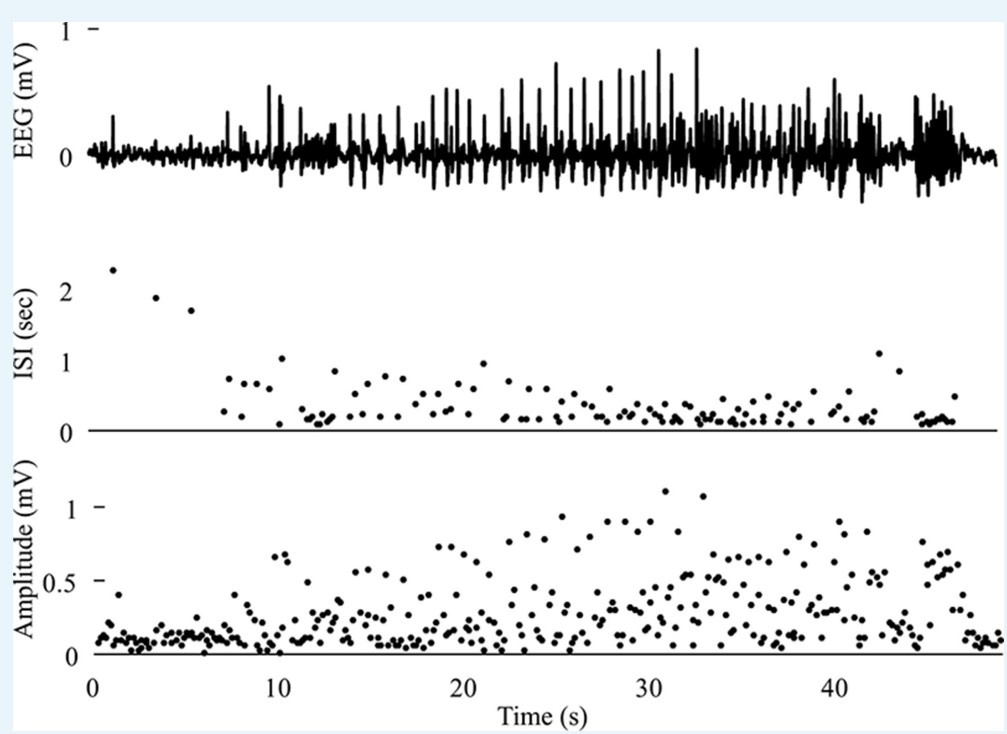

**Appendix 1—figure 17.** Seizure with acceleration of spike frequency at seizure terminus In this case, the ISI progressively decreases and follows a reverse power law. The amplitude is arbitrary. This seizure was labeled as a FLC bifurcation for taxonomical purposes as it did not fit with any other single bifurcation.

Seizures with FLC offset occur when the system is in the bistability region in the lower part of the map, which is shown in *Appendix 1—figure 18A*. This region has a wide range of frequency characteristics, depending on the specific path chosen. *Appendix 1—figure 18* shows the simulated results for three different paths that link a SN onset with FLC offset: it is possible to have seizures that have increasing, decreasing, or constant frequency. These differences occur because this region of the map has a great diversity of scaling laws in a small region, and the frequency has to change smoothly between them. In this particular case, even though FLC occurs with non-zero frequency, it is close to a SH curve which requires oscillations to slow down to zero. The closer the path chosen to the SH curve the slower the oscillations towards FLC offset.

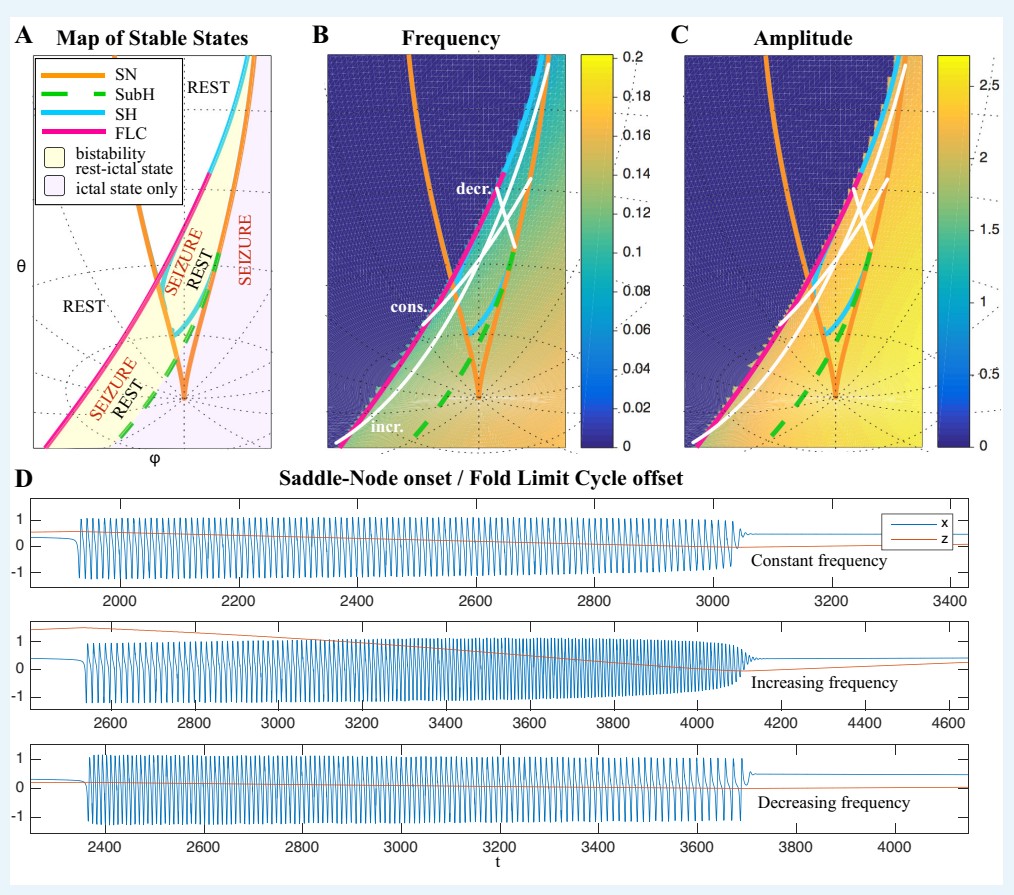

**Appendix 1—figure 18.** Varied frequency behavior of FLC offsets. Depending on how close the trajectory comes to the SH offset, an FLC offset can have a wide range of frequency behaviors. We show examples for three frequency trends for the FLC bifurcation. (**A**) Portion of the map in which the SubH/FLC type can be found. (**B-C**) Frequency and amplitude trajectories of three different seizures showing constant, increasing, and decreasing frequency. (**D**) Simulations for the paths shown in B-C. Note the last few spikes drop in amplitude precipitously, characteristic of the FLC releasing from the limit cycle (which has a finite amplitude when it disappears) and settling on the fixed point. This is different from the square-root decreasing amplitude of the SupH bifurcation, which is instead caused by a slow decrease in the amplitude of the limit cycle itself, in this latter case the limit cycle disappears when zero amplitude is reached.

Integration settings (same for the three simulations): Euler-Meruyama method; integration step 0.5 s; simulation length 15000 s; initial conditions were set to 0 for all variables; simulations without noise. Model parameter settings:

Constant frequency: $c = 0.001$; $d^* = 0.3$; $R = 0.4$; $A_1 = (0.1199, -0.0509, -0.3782)$; $B_1 = (0.2850, 0.0586, -0.2745)$.

Increasing frequency: $c = 0.002$; $d^* = 0.3$; $R = 0.4$; $A_1 = (-0.1031, -0.0996, -0.3734)$; $B_1 = (0.1436, 0.0331, -0.0622)$.

Decreasing frequency: $c = 0.003$; $d^* = 0.3$; $R = 0.4$; $A_1 = (0.2818, 0.0209, -0.2831)$; $B_1 = (0.2187, 0.0394, -0.3326)$.

Seizures with SupH offset occur when the system is in the bistability region in the upper part of the map, as shown in *Appendix 1—figure 19A*. Similar to the FLC case, the proximity of the SH curve, which has a frequency that goes to 0, affects the trajectories of SupH offsets (*Appendix 1—figure 19*).

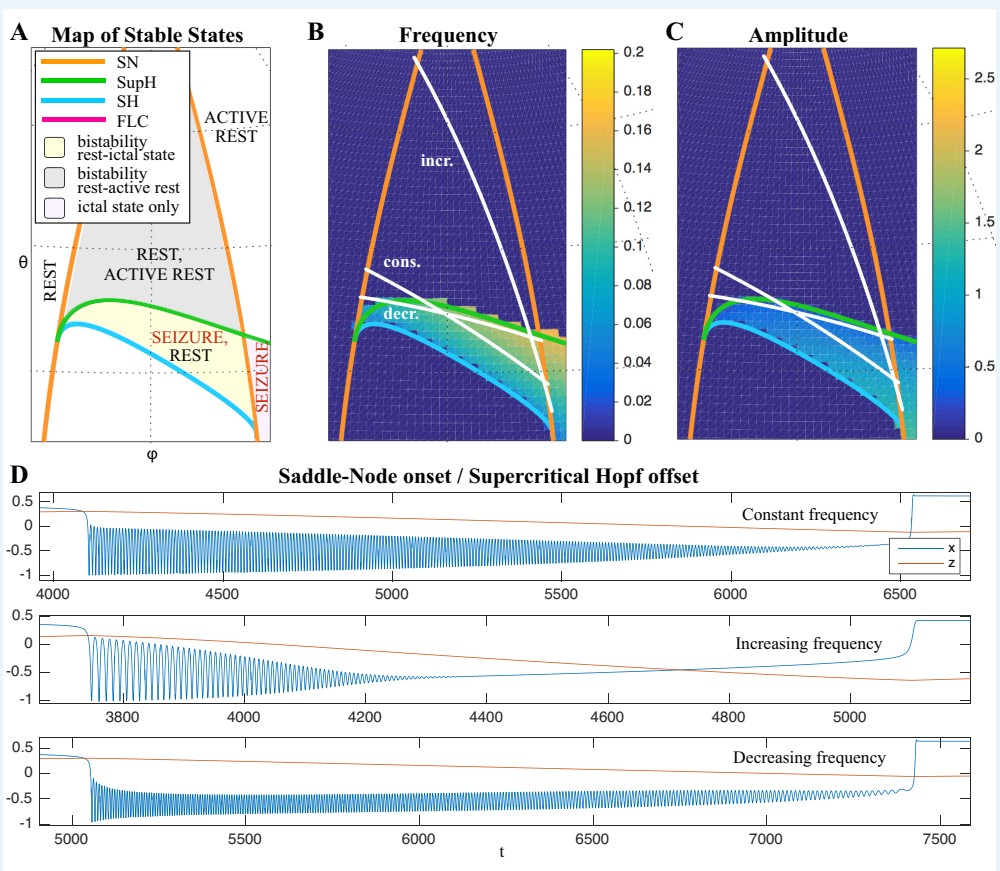

**Appendix 1—figure 19.** Frequency behavior of SupH offsets. The trajectory of seizure offset in SupH is influenced by the SH curve, allowing for constant, increasing and decreasing frequency trends. (**A**) Portion of the map in which the SN/SupH type can be found. (**B-C**) Frequency and amplitude trajectories of three different seizures showing constant, increasing, and decreasing frequency. (**D**) Simulations for the paths shown in B-C. Note the amplitude in each case is very characteristic of the SupH, despite the different frequency behavior.

Integration settings (same for the three simulations): Euler-Meruyama method; integration step 0.5 s; simulation length 20000 s; initial conditions were set to 0 for all variables; simulations without noise. Model parameter settings:

Constant frequency: $c = 0.00025$; $d^* = 0.3$; $R = 0.4$; $A_1 = (0.3101, -0.0217, 0.2517)$; $B_1 = (0.3426, 0.0772, 0.1915)$.

Increasing frequency: $c = 0.001$; $d^* = 0.3$; $R = 0.4$; $A_1 = (0.3216, 0.0625, 0.2294)$; $B_1 = (0.3479, 0.0790, -0.1810)$.

Decreasing frequency: $c = 0.0002$; $d^* = 0.3$; $R = 0.4$; $A_1 = (0.3102, -0.0437, 0.2488)$; $B_1 = (0.3249, 0.0713, 0.2221)$.

These two examples show some of the complex characteristics that can occur when a seizure's dynamics are in a region of the map that is close to two different bifurcations. This can cause unusual phenomena such as accelerating spikes at seizure offset, but also seizures that combine features from different bifurcations, such as FLC (**Appendix 1—figure 18**) or SupH (**Appendix 1—figure 19**) that show slowing down. This slowing down is in principle different from that of SH or SNIC because it does not follow a scaling law down to zero. The slowing down becomes more similar to that of SH-SNIC the closer the path gets to these bifurcation curves. Some extreme examples are shown in **Appendix 1—figures 18** and **19**, in which it would be ambiguous to classify based upon ISI of the timeseries. In such cases, only the presence of amplitude scaling (for SupH) would help disambiguate.

## VI. Projection of other epilepsy models onto the Taxonomy of Seizure Dynamics map

The model in *Saggio et al., 2017* and *Figure 4* is capable of navigating the entire projection with first order trajectories and provides minimal models for many other bursting types. For instance the Epileptor model maps to the region of the SN/SH type. In addition to the Epileptor, the map contains the bifurcation diagrams of other seizure onset and/or offset models used in the literature. For instance, *Sinha et al., 2017* used a model (only fast variables) with the same bifurcation diagram as the subH/FLC type. That type is also present in the lower portion of this map. The path of the SubH/FLC type is equivalent to several bistable and physiologically-inspired seizure models (*Suffczynski et al., 2005*; *Suffczynski et al., 2006*; *Marten et al., 2009*; *Kalitzin et al., 2011*; *Benjamin et al., 2012*; *Goodfellow et al., 2012*; *Meisel and Kuehn, 2012*; *Terry et al., 2012*; *Kalitzin et al., 2014*; *Hutchings et al., 2015*). Some of these models use noise induced transitions, others bifurcations (see Appendix III). Excitable models, such as those in *Wendling et al., 2002* and *Goodfellow et al., 2016*, can be obtained in the map when the system is close to the SNIC curve. There are also models used in the context of epilepsy that contain bigger portions of the map, such as the Jansen-Rit and the Wendling-Chauvel (*Touboul et al., 2011*) models. This versatility led other authors to state that the degenerate Bogdanov-Takens bifurcation (the basis of our map) is 'a good candidate for a simple qualitative model of a cortical mass' (*Touboul et al., 2011*).

