## [Decision Letter]

**Acceptance summary:**

There is currently no clear way to characterize and compare seizure dynamics. In this work the authors introduce an organizing principle that leads to the first objective Taxonomy of Seizure Dynamics (TSD) based on bifurcation theory, introducing the concept of a seizure "dynamotype". They present easily classifiable types as well as transparently presenting difficult cases.

This tool will be able to provide objective metrics for classification and thus aid in an understanding of seizure dynamics.

**Decision letter after peer review:**

Thank you for submitting your article "Epidynamics characterize and navigate the map of seizure dynamics" for consideration by *eLife*. Your article has been reviewed by three peer reviewers, and the evaluation has been overseen by a Reviewing Editor and Michael Frank as the Senior Editor. The following individuals involved in review of your submission have agreed to reveal their identity: Wim Van Drongelen (Reviewer #2); Taufik A Valiante (Reviewer #3).

The reviewers have discussed the reviews with one another and the Reviewing Editor has drafted this decision to help you prepare a revised submission. In recognition of the fact that revisions may take longer than the two months we typically allow, until the research enterprise restarts in full, we will give authors as much time as they need to submit revised manuscripts.

Summary:

In this work, the authors classify seizure onset and offset using mathematical theoretical 'tools' of nonlinear dynamics associated with bifurcations. They employ a large set of clinical data considering their classification leading to intriguing suggestions. Two main contributions are discerned. The first is the classification of the onset and offset of epileptic seizures in iEEG data using dynamical quantities associated with bifurcations in dynamical systems. This type of idea has been used in other contexts, e.g., simplified/large-scale climate models. The authors make a strong case, using analysis of real iEEG data, that their classification system is robust. Their analysis of the data leads to some intriguing results: that patients could exhibit more than one “type" of seizure onset/offset and that seizure classification did not correlate with pathology, but did correlate with age. The second contribution is to link the bifurcations they associated with the iEEG data to the unfolding of a particular codimension three bifurcation point. They then hypothesize that the seizures are bursting solutions caused by slow variation of a parameter in this unfolding and show how this hypothesis can account for some of their observations, such as a single patient having multiple seizure types. Both contributions could help increase our understanding of seizure dynamics.

While all of the reviewers thought that the work was interesting, several concerns were raised along with various points requiring clarification.

Essential revisions:

1) It was agreed by all that this paper was more appropriate as a 'tools' type rather than a 'research article' type paper. This is because it mainly revolves around using bifurcation theory as a 'tool' to classify seizure onset and offset. From this perspective then some of the concerns regarding results could potentially be less of an issue and/or presented differently.

Specifically, a principal problem is that a significant part of the data presented is inconclusive (the experimental support, e.g. in Appendix—figure 1). Therefore, one could say the epidynamics approach is neither falsified nor verified. The DC, amplitude and burst interval criteria don't seem to work that well in the examples shown. I fully agree with the authors that this may be due to noise and limitations in the recording procedures. Because of this, I find it difficult to assess how the charts with a relatively low % of no-consensus in Figure 2E were obtained. I would like a clarification. In addition, in the Appendix I would prefer to see cases that clearly illustrate the proposed dynamics while admitting that such validation is often not feasible.

2) Given the general nature of this journal, using "Epidynamics" was thought to be less than ideal as it quite close to 'epidemics' wording. Also, from the 'math model' perspective, this is not about the 'epileptor' model per se, but rather, similar from the consideration of bifurcation classification. The authors are urged to consider different wordings to avoid confusion and misinterpretations. In light of a revised presentation of the work, if presented as a potential 'clinical tool', then the authors could make clear how they obtain/develop their classification from the JMath Neurosci model with a range of timescales etc. as given in the extensive methods (i.e., not their previous epileptor model per se), then describing the clinical data in this 'tool' context that suggests its potential robustness and usage. This would also help avoid confusion about 'epileptor' and 'epidynamics' – really the similarity is about bifurcation 'quantities'.

3) The authors should clearly present how this 'clinical tool' could/should be viewed in light of other available techniques along with usage and future considerations. Is it yet another way to look at clinical data? There were some doubts as to whether this would/could be adopted as it would slow down clinical workflows, and since the primary question that clinicians ask (from a surgical perspective) is "where does the seizure start", with the 'when' guiding this effort, would argue there will be little uptake from a clinical perspective. The lack of a relationship to pathology and demographics also seems to further limit its clinical utility.

Given their attached future work, the authors may be viewing this as trying to associate cellular mechanisms to onset and offset dynamics? Neuromodulation? It would be helpful for the authors to articulate all of this in a revised version.

4) The work is predicated on previous seminal dynamical system theory work describing cellular bursting, and their 2017 paper. The methodology seems to accomplish relating key features of different bifurcations to clinical recordings, and they provide specific examples of predictions (class changes, and status epilepticus) using a dynamical system their derived in 2017 to display the various onset/offset bifurcations.

The predictions of a specific dynamical system they derived (2017, illustrated in Figure 4; which I assume is the same model presented in their previous paper) are presented as examples of a general theory to describe seizure onset/offset dynamics. I was wondering if the dynamical system that they developed in 2017, and that is applied here, is but one of many possible dynamical systems that could describe what is seen clinically. In other words, might this (Figure 4, and 2017 paper) be but one example of many other potential dynamical systems that could explain some but not all the aspects of seizure dynamics? The question is asked in light of a statement made in the Introduction, where it is stated that "Here we present the first mathematical theory accounting for the essential onset and offset dynamics of electrophysiological seizures based upon first principles." Is it a "theory" that is being presented or a formulation within an existing theory (dynamical systems theory) that is being presented and thus is one of many such formulations.

5) The seizure onset is defined to occur when there is an obvious change from the baseline/destabilization of the resting state. As they allude to not uncommonly there is a sentinel spike, that is seen clinically as well as in in vivo, and in vitro seizure models. Although it is mentioned that the phenomenon is a not a canonical feature of any of the onset bifurcations, it seems curious that a general theoretical framework for describing seizure onsets does not account for this. It is not to minimize the presented work, however, can the authors explain using their formulation how might their system generate this sentinel spike, or why it cannot be used to guide future work.

Along the same line, and again to not minimize this work, however inter-ictal spikes co-exist with seizures, and are used as an indirect marker for the epileptogenic zone. Such spikes are seen in iEEG recordings, in vivo and in vitro seizure models. What does Epidynamics suggest as an explanation for these phenomena that co-exist with seizures? I think this should be discussed.

---

## [Author Response]

Essential revisions:1) It was agreed by all that this paper was more appropriate as a 'tools' type rather than a 'research article' type paper. This is because it mainly revolves around using bifurcation theory as a 'tool' to classify seizure onset and offset. From this perspective then some of the concerns regarding results could potentially be less of an issue and/or presented differently.Specifically, a principal problem is that a significant part of the data presented is inconclusive (the experimental support, e.g. in Appendix—figure 1). Therefore, one could say the epidynamics approach is neither falsified nor verified. The DC, amplitude and burst interval criteria don't seem to work that well in the examples shown. I fully agree with the authors that this may be due to noise and limitations in the recording procedures. Because of this, I find it difficult to assess how the charts with a relatively low % of no-consensus in Figure 2E were obtained. I would like a clarification. In addition, in the Appendix I would prefer to see cases that clearly illustrate the proposed dynamics while admitting that such validation is often not feasible.

We agree that this will be suitable as a Tools type paper and editorial staff have updated the article accordingly.

The reviewers point out concerns here about the feasibility of experimental verification. While classifying as a “tools” paper does alleviate some concerns, it is clear we must explain the validation better. The short answer is that our presentation did not make a clear enough distinction between the examples that “clearly illustrate the proposed dynamics” and the examples that were meant to show how difficult cases were handled. It is also critical to establish the expectations of what this paper seeks to accomplish.

Our paper indeed has several examples that are quite clear. Figure 2B-D, Appendix—figure 1A, G, H, I are all non-ambiguous following the rules we set forth. Each of these is easily classifiable, and all 3 of our reviewers agreed in each case as described. However, in the interest of being transparent and preparing others for how to classify the inevitable “difficult cases”, we also included several challenging seizures. And we stress that we did not search through the data to find “perfect” seizures that fit each bifurcation type – we quite literally chose the first suitable seizure available from each patient, and presented those results. Despite this, we found a wide array of seizure types (i.e. “dynamotypes”, see our first response to comment #2), and some of them were challenging to classify. We have included a section of the Appendix (subsection “Challenges with classification”) to describe how to handle unusual and challenging patterns.

According to our rules, most patterns that do not fit with DC or trends in amplitude or ISI are “arbitrary” and are classified as SN/SubH for onset and FLC for offset. However, as we describe in “Analyzing arbitrary dynamics” in the main body and Appendix subsection “Unusual seizure dynamics”, these patterns could also be due to noisy fluctuations; our system classifies them as arbitrary. Overall, our goal in the Appendix was not to suggest that our classification is so comprehensive that all seizures would fit perfectly. On the contrary, our goal was to present robust data as a reality check, in order to allow others to use this tool when faced with the inevitable heterogeneity of patterns seen under real-world conditions. As described below, we are confident that the described methods are easy to learn to achieve reliable classification when that is possible. We tested that we could do just that by recruiting and quickly training a new author as one of our scorers.

As the reviewers point out, there will always be noise in a real system: ideal cases do not arise in nature and we do not expect to find perfectly symmetric cases. Human seizures are an excellent real-world scenario to demonstrate this problem. However, since the invention of EEG we have been trying to group seizures into convenient categories: “absence”, “temporal lobe epilepsy”, “myoclonic/clonic/tonic/tonic-clonic/atonic seizures”. However, even these standard categories are imperfect, despite being so universally regarded that they have survived the last two rounds of redefining the epilepsy nomenclature (Berg et al., 2010; Fisher et al., 2017). For instance, our dataset includes 8 patients that are cared for by the author of this response. Of these, 2 are “exceptions.” Both have focal epilepsy, but one has 2.5 Hz spike wave discharges and the other had tonic seizures with maintained awareness – both of these cases blatantly break the rules of standard clinical terminology. There is simply too much variability in nature to expect classification to be perfect.

It is thus not surprising, in a diverse collection of seizures, to find some that do not “fit” with a given classification system – even one based on empiric, subjective measurements like semiology. Trying to develop a new electrophysiological classification invites even more variability. As such, we feel it is improper simply to present the “perfect” examples, as any practical implementation of our methods will immediately run into gray areas. Recognizing there was a great deal of variability, we found it was necessary to create a flow chart, in which certain features would be given priority. For instance, if there is a DC shift, regardless of any other pattern, it is SN onset or SH offset. In the absence of that, if there is a trend in amplitude it is a SupH, and so on. Our presentation in Appendix—figure 1 follows this pattern, using the same order of progression described in Appendix—table 1 and Appendix subsections “Visual classification of seizure dynamics”, “Visual classification of seizure dynamics”, “Validation of classification methods” and “Challenges with classification” (DC shift first, then amplitude, then ISI, then arbitrary).

To see if this method worked, we first verified that these rules work in the simulated data, with great success. Then, we recruited a clinical epileptologist (M. Nakatani), who had no prior experience in these methods or exposure to our group, trained him for 30 minutes using these methods, then compared him with the two authors who developed the tool (DC, WS). He is one of the three reviewers used for the consensus measurements in Figure 2, and Appendix—tables 2, 3. Thus, although Appendix—figure 1 is not “perfect”, it is realistic. This is what actually happens when you apply this method to over 100 patients from 8 centers around the world – there are several “exceptions”. Nevertheless, we feel our Appendix does not detract from the power of our tool or suggest it fails – instead, we feel it provides some realism and practical guidance for clinical translation.

That realism and guidance, we believe, has been somewhat absent from past works. As an example of this, we also applied the published “Perucca” categorization to our data, which is based upon standard clinical descriptions of the waveforms (amplitude and frequency of the spikes) (Perucca et al., 2014). We found that, while there were several seizures that could easily be categorized into the 7 types, nearly half of them had some ambiguity (what if the amplitude/frequency is on the border? What if the pattern is not consistent? What if the pattern changes from one to another, what if none of the patterns really fit, etc… All of these happen very frequently). There was little guidance about how to handle such situations. We have approached the current paper with a higher degree of transparency in the hopes of providing better guidance and a necessary part of that is demonstrating how to approach difficult conditions. As a consequence, we present some data that do not look “perfect”, just like the data we expect others to find.

In order to clarify this, we have made the following changes to the document.

1) In each example in Appendix—figure 1, we include a “Classification” statement at the end describing the consensus of the reviewers, and how it was reached by following the pattern set forth in Appendix—table I. We are fully transparent about the concerns of the different scorers, showing how such situations can be handled in the future. This style lends to more of a training didactic style.

2) One important weakness is that our human data only had 3 SNIC onsets, and none of them are “perfect” examples – this was one of the primary concerns in this comment. However, as we point out in the paper, SNIC are less likely to occur by the theory as well (Appendix—figure 5). The SNIC example now has more discussion about how it is not completely straightforward. Most importantly, however, we now include a new figure containing SNIC onsets from a recently-published paper from rats (Crisp et al., 2020) (new Appendix—figure 1D). These seizures are much more demonstrative of SNIC.

3) In Appendix subsection “Examples of classifying clinical bifurcations”, we start with the following explanation, which instructs the reader what to expect from the primer figures in the Appendix:

“Due to the large degree of heterogeneity and noise in clinical recordings, it is common to have patterns that do not fit perfectly with the theoretical dynamotype. […] However, it is important to note that, as a whole, the reviewers were able to reach a high level of agreement (see Figure 2, Appendix—tables 2 and 3) – most seizures can be classified reliably by following these rules.”

4) We have changed the main text to read “To supplement the examples in Figure 2, as well as clarify how to approach several challenging scenarios, there is a primer with examples of all the different bifurcations in Appendix subsection ‘Examples of classifying clinical bifurcations’.” We also state that “Note however that the identification of bifurcations from empirical data is notoriously difficult and generally cannot unambiguously prove that a given bifurcation is present, although it allows assessment of self-consistency. Further investigation may use additional tools, such as perturbations of the system, to corroborate these results further.”

2) Given the general nature of this journal, using "Epidynamics" was thought to be less than ideal as it quite close to 'epidemics' wording. Also, from the 'math model' perspective, this is not about the 'epileptor' model per se, but rather, similar from the consideration of bifurcation classification. The authors are urged to consider different wordings to avoid confusion and misinterpretations.

We pondered a great deal over this comment and comment #4, which are highly interrelated. We feel our assessment of dynamics needs a “name”, and in response to these comments have developed a much more suitable name. We have changed the terminology by coining the term “dynamotype”. This terminology is more straightforward and allows a direct comparison with other clinical nomenclature: seizures can thus be described by their genotype, phenotype, and dynamotype. These three classifications may be related but are independent measures. The clinical terminology describes the phenotype, but until this time there has not been a scientifically-based assessment of underlying dynamics. We have altered the rest of the paper accordingly.

In addition, our map of seizure dynamics provides a taxonomy, which we now call the Taxonomy of Seizure Dynamics, TSD.

As described in comment #4, this new terminology leads to much more appropriate discussions of what this paper provides (as the Abstract now states “We introduce an organizing principle that leads to the first objective Taxonomy of Seizure Dynamics (TSD) based on bifurcation theory”), and how it fits into the clinical repertoire (phenotype, genotype, and dynamotype).

In light of a revised presentation of the work, if presented as a potential 'clinical tool', then the authors could make clear how they obtain/develop their classification from the JMath Neurosci model with a range of timescales etc. as given in the extensive methods (i.e., not their previous epileptor model per se), then describing the clinical data in this 'tool' context that suggests its potential robustness and usage. This would also help avoid confusion about 'epileptor' and 'epidynamics' – really the similarity is about bifurcation 'quantities'.

We now clarify this point, also in light of the request in comment #4. We now emphasize that the classification is fully based on dynamotype, the composite characteristic features of onset and offset bifurcations. This leads naturally to the unbiased classification along these two axes and a Taxonomy of Seizure Dynamics (TSD). Only then, as a second step, the link to canonical dynamic systems enters via the JMath Neurosci work and essentially breaks the symmetry between dynamotypes, introducing a classification hierarchy. This clarification and perspective are presented at various locations in the paper, in particular in the Introduction, Results and Discussion sections.

3) The authors should clearly present how this 'clinical tool' could/should be viewed in light of other available techniques along with usage and future considerations. Is it yet another way to look at clinical data? There were some doubts as to whether this would/could be adopted as it would slow down clinical workflows, and since the primary question that clinicians ask (from a surgical perspective) is "where does the seizure start", with the 'when' guiding this effort, would argue there will be little uptake from a clinical perspective. The lack of a relationship to pathology and demographics also seems to further limit its clinical utility.Given their attached future work, the authors may be viewing this as trying to associate cellular mechanisms to onset and offset dynamics? Neuromodulation? It would be helpful for the authors to articulate all of this in a revised version.

We agree that it is important to provide a better discussion of how this tool might be used. It is especially important to make it clear that this tool is not intended to replace clinical practice. What we have developed is a description of a seizure’s “dynamotype”. Our Introduction now explains that:

“In effect, that [2017 ILAE clinical] classification is based upon the epilepsy phenotype, the clinical symptoms that arise during a seizure. […] The organization of seizures along dynamotypes leads naturally to a Taxonomy of Seizure Dynamics (TSD) providing practical, objective metrics for classification.”

Thus, we believe this tool is orthogonal to the current methods – it addresses a critical aspect that has been relatively absent and therefore opens many new possibilities for clinical and research applications. The goal is to help clinicians focus on the most important dynamical aspects and provide a language for discussing dynamics.

Obviously, the questions of when and where a seizure starts will remain, and are likely to continue with the same methods. However, there are many other questions that constantly nag clinicians and researchers, which this tool is uniquely suited to address. As we now state in the Discussion:

“There is great clinical and research potential in characterizing a seizure’s dynamotype, as it provides a unique perspective on brain networks. […] This would supplement current visual descriptions, which typically are limited to amplitude and frequency. But there are many deeper applications of this tool as well.”

That paragraph is followed by several examples: our recent publication showing how dynamotype characterizes epileptogenesis, the ongoing EPINOV clinical trial in France, applications to network tools, use for control theory applications such as probing seizure threshold and tailoring stimulation, and personalized patient modeling. Thus, we feel this tool opens many new doors for research, several of which are already underway. This paper will be the introduction to these tools.

4) The work is predicated on previous seminal dynamical system theory work describing cellular bursting, and their 2017 paper. The methodology seems to accomplish relating key features of different bifurcations to clinical recordings, and they provide specific examples of predictions (class changes, and status epilepticus) using a dynamical system their derived in 2017 to display the various onset/offset bifurcations.The predictions of a specific dynamical system they derived (2017, illustrated in Figure 4; which I assume is the same model presented in their previous paper) are presented as examples of a general theory to describe seizure onset/offset dynamics. I was wondering if the dynamical system that they developed in 2017, and that is applied here, is but one of many possible dynamical systems that could describe what is seen clinically. In other words, might this (Figure 4, and 2017 paper) be but one example of many other potential dynamical systems that could explain some but not all the aspects of seizure dynamics? The question is asked in light of a statement made in the Introduction, where it is stated that "Here we present the first mathematical theory accounting for the essential onset and offset dynamics of electrophysiological seizures based upon first principles." Is it a "theory" that is being presented or a formulation within an existing theory (dynamical systems theory) that is being presented and thus one of many such formulations.

We thank the reviewers for insisting on this point, as we have also been struggling with it and consider the clarification of the language essential for communication, but also for correct positioning of our work. It is more than just a mathematical model, but not a theory either. The best perspective is probably obtained, when comparing it with the periodic table of chemical elements as we now discuss explicitly in the manuscript. Our work provides an organizing principle of seizure dynamics based on constructive methods, which links structural features (proton electron number, bifurcation types) to functional features (chemical properties, discharge behaviors). This perspective provides an intuitive feeling for the organization, and then a better understanding of the subsequent link to the 2017 paper. We write in particular the following in the Introduction:

“In this work, we introduce an organizing principle of seizure dynamics based on nonlinear dynamics and bifurcation theory. […] TSD is available for immediate transfer to clinical practice, providing a rational method of characterizing seizures and subsequently a better understanding of the underlying principles governing seizure generation and termination.”

5) The seizure onset is defined to occur when there is an obvious change from the baseline/destabilization of the resting state. As they allude to not uncommonly there is a sentinel spike, that is seen clinically as well as in in vivo, and in vitro seizure models. Although it is mentioned that the phenomenon is a not a canonical feature of any of the onset bifurcations, it seems curious that a general theoretical framework for describing seizure onsets does not account for this. It is not to minimize the presented work, however, can the authors explain using their formulation how might their system generate this sentinel spike, or why it cannot be used to guide future work.

Thank you for raising this issue. In our previous version of the paper, we mentioned that sentinel spikes are not a canonical feature of the seizure dynamics and so were not, and in fact should not be, included in the dynamotype (but were used to identify seizure onset times). However, we recognize that our previous treatment of this subject left several unanswered questions. We searched the literature for “sentinel spike”. The first mention in the literature we are aware of is from Cukiert et al. (2001). Many papers refer to “sentinel spikes” at seizure onset, in particular for seizures that are characterized by Low Voltage Fast (LVF) onset. To the best of our knowledge, each time they are mentioned, they are simply acknowledged and observed, but we could find no explanation of their etiology or clinical significance. To illustrate this, please see panel B from Figure 1 of Lévesque et al. (2016), a rigorous work which included microelectrode recordings and spike sorting; even that work did not address the mechanisms of the sentinel spikes that were observed. However, we note in this figure that the sentinel spike appears to have a morphology similar to the interictal spikes, and there is a possible return to baseline prior to LVF seizure – thus in this case we cannot be certain whether this particular spike was involved in seizure onset or not. Clearly, there is much ambiguity in sentinel spikes.

So, it appears the mechanisms and consequences of these spikes are still unknown. However, their frequent presence in LVF seizures is intriguing (though note that sentinel spikes were not restricted to, nor always present in, our LVF seizures). LVF mostly corresponds to SN onset seizures in our taxonomy, i.e. seizures that often start with a DC shift. When acquired under standard clinical protocols (usually with a high pass filter of at least 0.1 Hz, i.e. AC coupling), a fast DC shift will appear as a spike (e.g. our Figure 2C in the paper). Most recordings in humans and animal models (in vivo and in vitro) are done in AC. We illustrate this with an in vivo recording of an epileptiform discharge (Author response image 1).

**Author response image 1. sa2fig1:** Representative example of an epileptiform discharge recorded in the hippocampus in a freely moving animal (Tetanus Toxin injected in the hippocampus to create a focus). The upper trace is in AC, the lower one is in DC. Note that the first AC “spike”, which would be qualified as “isolated interictal spike” is actually a very large DC shift. The second “spike” is part of an epileptic discharge. Note that AC filtering artificially shortens the true duration of the event. The large DC shift will eventually recover (not shown).

Hence, there are many reasons that limit our ability to include them in our analysis.

1) Without performing DC recordings, we cannot be sure when this “sentinel spike” is actually just a DC shift, and when it is a true spike.

2) Even with high resolution recordings we have difficulty discerning when it is an integral part of some seizures.

3) The science has little understanding of what is going on.

4) From our theory, there is no invariant feature in the bifurcations to explain this phenomenon.

Thus, we feel uncomfortable discussing this issue more than already done, as it deserves a more thorough examination. In our in vivo recordings, most spikes are fast transient events (no DC shift), but many “big” spikes are in fact DC shifts, in particular before or at seizure onset.

To explain these limitations, we have amended the Appendix subsection “Spike analysis” as follows:

“Sentinel spikes. Some seizures began with a single high amplitude ‘sentinel spike,’ which typically was present in many electrodes, often far more than were involved in the subsequent seizure. […] Thus, in this work, we note their presence to determine seizure onset time, but at present do not include them within a framework of a canonical model of seizure dynamics.”

Along the same line, and again to not minimize this work, however inter-ictal spikes co-exist with seizures, and are used as an indirect marker for the epileptogenic zone. Such spikes are seen in iEEG recordings, in vivo and in vitro seizure models. What does Epidynamics suggest as an explanation for these phenomena that co-exist with seizures? I think this should be discussed.

The full version of the original Epileptor model (Jirsa et al., 2014) includes interictal spikes. By definition, they are not part of the dynamotype, but may obviously depend on it. In the Epileptor model they were thus treated phenomenologically. But to extract the basic features of seizure onset and offset dynamics, they are not necessary. As requested, we now have added a relevant paragraph in the Discussion.

“One limitation of previous clinical descriptions of seizures is that it has been unclear which dynamical features are relevant. […] Our approach is more generic, but can be refined with inclusion of a second set of differential equations to generate spikes (Jirsa et al., 2014).”